

# Image Classification of Marine-Terminating Outlet Glaciers using Deep Learning Methods

Melanie Marochov, Chris R. Stokes, Patrice E. Carbonneau

Department of Geography, Durham University, Durham, DH1 3LE, UK

*Correspondence*: Melanie Marochov (melanie.marochov@durham.ac.uk) and Patrice E. Carbonneau
(patrice.carbonneau@durham.ac.uk)

**Abstract.** A wealth of research has focused on elucidating the key controls on mass loss from the Greenland and Antarctic ice sheets in response to climate forcing, specifically in relation to the drivers of marine-terminating outlet glacier change. Despite the burgeoning availability of medium resolution satellite data, the manual methods traditionally used to monitor change in satellite imagery of marine-terminating outlet glaciers are time-consuming and can be subjective, especially where mélange exists at the terminus. Recent advances in deep learning applied to image processing have created a new frontier in the field of automated delineation of glacier termini. However, at this stage, there remains a paucity of research on the use of deep learning for pixel-level semantic image classification of outlet glacier environments. In this contribution, we apply and test a two-phase deep learning approach based on a well-established convolutional neural network (CNN) called VGG16 for automated classification of Sentinel-2 satellite images. The novel workflow, termed CNN-Supervised Classification (CSC), was originally developed for fluvial settings but is adapted here to produce multi-class outputs for test imagery of glacial environments containing marine-terminating outlet glaciers in eastern Greenland. Results show mean F1 scores up to 95% for in-sample test imagery and 93% for out-of-sample test imagery, establishing a state-of-the-art in classification of marine-terminating glacial environments with significant improvements over traditional pixel-based methods such as band ratio techniques. This demonstrates the transferability and robustness of the deep learning workflow for automated classification despite the complex and seasonally variable characteristics of the imagery. Future research could focus on the integration of deep learning classification workflows with platforms such as Google Earth Engine, to more efficiently classify imagery and produce datasets for a range of glacial applications without the need for substantial prior experience in coding or deep learning.

## 1 Introduction

Quantifying glacier change (e.g. volume, area, geometry, surface hydrology, and terminus position) from remote sensing data is essential to improve our understanding on the impacts of climate change on glaciers (Vaughan et al., 2013; Hill et al., 2017). In many glaciated areas, well-established semi-automated techniques such as image band ratio methods are used to extract



glacier outlines for this purpose and to create glacier inventories (Paul et al., 2016). These methods accurately classify areas of debris-free ice in contrast to surrounding topography and are widely used in studies of mountain glaciers and ice caps (e.g. Bolch et al., 2010; Frey et al., 2012; Guo et al., 2015; Rastner et al., 2012; Stokes et al., 2018). However, these approaches are less effective for accurately mapping more complex glaciers and glaciated landscapes. For example, highly debris-covered mountain glaciers are more difficult to classify using band ratios, and often require labour intensive manual adjustments and

the integration of high resolution digital elevation models (Paul et al., 2016; Herreid and Pellicciotti, 2020). This has resulted in the development of several alternative automated techniques, using methods such as object based image analysis and decision tree algorithms (e.g. Racoviteanu and Williams, 2012; Robson et al., 2015). Band ratio techniques also struggle to accurately map the termini of marine-terminating outlet glaciers such as those surrounding Greenland and Antarctica. This is largely due to the presence of seasonally variable areas of a spectrally similar mélange containing sea-ice and icebergs near

their termini (e.g. Amundson et al., 2020). Additionally, to extract boundaries between classes using band ratio methods requires the establishment of a threshold value which varies locally depending on differences in image properties (Walter, 2004) and thus makes the method less transferrable to different data sources or geographical regions.

Whilst manual digitisation remains the most common technique used to map marine-terminating outlet glacier termini (e.g.
Miles et al., 2016, 2018; Carr et al., 2017; Wood et al., 2018; Brough et al., 2019; Cook et al., 2019; King et al., 2020), its labour intense and time-consuming nature often results in studies which are limited to small spatial areas, and only analyse seasonal glacier dynamics at the scale of individual glaciers (Seale et al., 2011). In contrast, where studies encapsulate larger numbers of glaciers over increased spatial areas, terminus position monitoring is often limited to inter-annual to decadal scales (e.g. Moon and Joughin, 2008), removing the opportunity to understand seasonal changes and drivers. The importance of

processes occurring at marine-terminating outlet glaciers on a range of timescales (Amundson et al., 2010; Juan et al., 2010; Chauché et al., 2014; Carroll et al., 2016; Bunce et al., 2018; Catania et al., 2018, 2020; King et al., 2018; Bevan et al., 2019; Sutherland et al., 2019; Tuckett et al., 2019) and the drawbacks of manual digitisation highlight the growing need for a method to efficiently quantify outlet glacier change in an era of increasingly available satellite data. This in turn could allow the incorporation of a more detailed understanding of marine-terminating outlet glacier dynamics and interactions in models used

to project future sea-level changes (Csatho et al., 2014).

To confront this challenge, some automated techniques for extracting outlet glacier termini have been developed, exemplified in a small number of studies which delineate the boundaries of marine-terminating glaciers and ice shelves at the margins of the Greenland (Krieger and Floricioiu, 2017; Seale et al., 2011; Sohn and Jezek, 1999) and Antarctic ice sheets (Liu and Jezek,

2004; Yu et al., 2019). These methods generally rely on tools from the fields of image processing and computer vision, namely semantic segmentation, and edge detection. Semantic segmentation is a term used interchangeably with pixel-level semantic classification and refers to the process of dividing an image into its constituent parts based on groups of pixels of a given class, assigning each pixel a semantic label (Liu et al., 2019). Throughout the remainder of this study, we refer to this generally as



classification. The technique was used by Liu and Jezek (2004) to partition Synthetic Aperture Radar (SAR) imagery into two

major semantic classes (ice/land and water). Following substantial post-processing, they applied an edge detection algorithm to the classified image to extract the boundary between ice/land and water around Antarctica. Edge detection identifies areas in an image with abrupt changes in pixel brightness, therefore providing a useful tool to detect boundaries from satellite imagery (Chen and Hong Yang, 1995). In further work, Seale et al. (2011) applied an edge detection algorithm to Moderate Resolution Imaging Spectroradiometer (MODIS) satellite imagery of Greenland to detect glacier terminus positions with a

similar level of accuracy to manual digitisation. Despite adequate levels of accuracy, edge detection techniques require substantial pre- and post-processing, and have since only been used for terminus delineation in a few studies (e.g. Joughin et al., 2008; Christoffersen et al., 2012). Meanwhile, traditional statistical classification techniques (e.g. maximum likelihood) are not considered robust when it comes to extracting glacier termini where there is little contrast between glacier ice and a spectrally similar mélange of sea-ice and icebergs or even water containing icebergs (Baumhoer et al., 2019). Moreover,

classification techniques which rely solely on individual pixel values often miss contextual, class representative shapes and textures due to the resolution of satellite imagery, meaning that for land cover classification, pixel-based approaches rarely produce satisfactory levels of accuracy (Blaschke et al., 2000) and commonly result in noisy classifications (Li et al., 2014). Therefore, edge detection and traditional pixel-based classification methods have not yet overcome the widespread use of manual digitisation for marine-terminating outlet glaciers.


More recently, deep learning methods have been developed to extract terminus outlines and overcome the drawbacks of manual delineation, semi-automated edge detection techniques, and traditional pixel-based classifiers (Baumhoer et al., 2019; Mohajerani et al., 2019; Zhang et al., 2019). Each of these studies used a modified U-Net Fully Convolutional Neural Network (FCN) architecture (Ronneberger et al., 2015) and various post-processing techniques to extract the boundaries between 1) ice

shelves and ocean in Antarctica (Baumhoer et al., 2019), and 2) marine-terminating outlet glaciers and mélange in Greenland (Mohajerani et al., 2019; Zhang et al., 2019). In Greenland, the workflows used by Mohajerani et al. (2019) and Zhang et al. (2019) produced termini outlines with a mean deviation from manually delineated 'true' outlines of 96 m and 104 m, respectively. For Antarctic coastlines, Baumhoer et al. (2019) produced ice/ocean boundaries with a mean deviation from true fronts of 78 m for imagery of CNN training sites and 108 m for test sites previously unseen in model training. In terms of

classification accuracy, Baumhoer et al. (2019) produced classifications with mean F1 scores of 89 to 90% for training sites and 90 to 91% for test sites, which are the most accurate for image classifications of complex marine-terminating glacial settings to date. These deep learning methods have so far relied on a binary classification of input images. For example, Baumhoer et al. (2019) used only two classes (land ice and ocean), as did Zhang et al. (2019) who classified the input image into ice mélange regions and non-ice mélange regions (including both glacier ice and bedrock). While these methods are

incredibly useful for extracting glacier terminus outlines and quantifying fluctuations over time, they perhaps overlook the ability of deep learning methods to create highly accurate image classification outputs which contain more than two classes (i.e. not just ice and no-ice areas). Moreover, deep learning has been used successfully in other disciplines to classify entire



landscapes or image scenes to a high level of accuracy (Sharma et al., 2017; Carbonneau et al., 2020a). Despite this, image classification of entire marine-terminating outlet glacier environments has not yet been tested using deep learning.


The use of deep learning in glaciology is still in its infancy, but given the abundance of available satellite imagery, it could be a significant aid in the automation of image processing of glacial settings. Image classification using deep learning techniques has the clear potential to not only reduce the labour-intensive nature of manual methods but facilitate automated analysis in numerous research areas. Aside from terminus delineation, a method which quickly produces accurate multi-class image

classifications of complex and seasonally variable outlet glacier environments could provide an efficient way to further elucidate processes and interactions controlling outlet glacier behaviour at high temporal resolution (e.g. calving events, subglacial plumes, and supra-glacial lakes). The compatibility of deep learning image classification methods with platforms such as Google Earth Engine (Gorelick et al., 2017) and its integration with Geographic Information Systems (GIS) software could also improve the efficiency of such analysis and remove the need for prior expertise in deep learning and coding.


The aim of this paper is to adapt a deep learning method developed in fluvial settings to classify airborne imagery and test it on satellite imagery of marine-terminating outlet glaciers. The workflow is composed of two deep learning phases (Carbonneau et al. 2020a). We first modify and train a well-established CNN called VGG16 (Simonyan and Zisserman, 2015) using labelled image tiles from 13 seasonally variable images of Helheim Glacier, south east Greenland (Fig. 1). In the first phase of the

workflow, this transferable, pre-trained model is applied to an unseen image tile from an outlet glacier environment. The resulting class predictions are then used as training data specific to the unseen input image. In phase two, a second deep learning model uses the class predictions of the phase one CNN to determine a final pixel-level classification. While the methods developed here are primarily tested on Greenland outlet glaciers, they are also applicable to mapping outlet glaciers anywhere in the world, including Antarctica. We assess the sensitivity of the workflow to different band combinations, training

techniques, and model parameters for fine-tuning and transferability. Our objective is to establish and evaluate a workflow for multi-class image classification of glacial settings which can be accessed and used rapidly without having specialised knowledge of deep learning or the need for time-consuming generation of new training data. Furthermore, we aspire to exceed the current state-of-the-art and advance accuracy levels (F1 scores >90%) for pixel-level image classification of glacial environments which contain complex marine-terminating outlet glaciers.











**Figure 1: Location of outlet glacier environments used for training and testing the deep learning workflow. (a) Sentinel-2 tile of Helheim Glacier (acquired 13/09/2019) used for testing the workflow (in-sample), with inset which shows the specific area used to create training data. (b) Sentinel-2 tile of Scoresby Sund area (acquired 01/08/2019) used for testing the workflow (out-of-sample). (c) Model training area (acquired 07/08/2019). Note the substantial ice mélange and active plume at the terminus of Helheim Glacier.**



## 2 Deep Learning and Convolutional Neural Networks (CNNs)

Deep learning is a type of machine learning in which a computer learns complex patterns from raw data by building a hierarchy
of simpler patterns (Goodfellow et al., 2016). While the field of deep learning has been evolving since the 1940s (Goodfellow
et al., 2016), the discipline has experienced significant advances over the past few decades alongside computer vision. This
has resulted from the increasing availability and size of training datasets, and the improvement of computer hardware and
software (LeCun et al., 2015). Numerous fields have helped shape the development of contemporary deep learning, including
contributions from neuroscience, engineering, and fundamental mathematical principles such as probability theory (see
Goodfellow et al., 2016 for a detailed review). Several of the earliest designs of deep learning architectures were inspired by,
and attempted to replicate, learning procedures in the mammalian brain, whereby layers of computational 'neurons' interact
to acquire knowledge from an input (Goodfellow et al., 2016). For example, Fukushima (1980) developed a neural network
for pattern recognition in images called the neocognitron. The model was based on the organisation of neurons used for visual
perception, elucidated by early studies of the visual system in cats (Hubel and Wiesel, 1962). It was designed to correspond to
the ventral stream of the visual cortex which processes a retinal image using a hierarchy of cells from the eye to the primary
visual cortex (V1), visual areas V2 and V4, and the inferotemporal (IT) cortex (Hubel and Wiesel, 1962; Serre, 2013). Neurons
in each progressive level of the hierarchy can identify increasingly complex features ranging from simple edges in the V1
visual area to complex combinations composing entire patterns and objects in the IT visual area (Felleman and Van Essen,
1991). Alongside this, neurons in higher stages of the hierarchy are shown to be increasingly tolerant to small changes in the
scale and position of input images (Serre, 2013). This increase in image processing and neuron invariance represented by
progressive layers in the visual hierarchy was also a key inspiration for the convolutional and pooling layers in the more recent
CNN (LeCun et al., 1989, 1998).

CNNs are deep learning models specifically designed to process multiple 2D arrays of data such as multiple image bands
(LeCun et al., 2015). They differ from conventional classification algorithms based solely on the spectral properties of
individual pixels by detecting the contextual information of images such as texture, in the same way a human operator would.
CNNs are usually arranged in a series of layers containing convolutional, non-linearity, and pooling functions (LeCun et al.,
2015). The input data is converted into an array of features (called a feature map) in each convolutional stage using a locally
weighted sum which represents an array of parameters adjusted by the model learning algorithm (Goodfellow et al., 2016).
Initial convolutional layers learn low-level features such as lines and edges which compose the high-level features extracted
by deeper convolutional layers, allowing the model to extract textures and shapes representative of image classes (Cheng et
al., 2017). The output passes through a non-linear activation function such as the rectified linear unit (ReLU) and then goes
through a pooling layer to introduce some invariance to the features, meaning the model can detect features with small
variations such as differences in orientation (Goodfellow et al., 2016). There are typically several of these stages in a CNN,



creating a hierarchy similar to that of the mammalian visual system, allowing the model to learn features from an image and output a prediction of class for each pixel. As a result of this, one of the main benefits of CNNs is that they remove the need for prior feature extraction for image classification (Längkvist et al., 2016). The CNN we use for image classification falls into the category of supervised learning (Goodfellow et al., 2016). This means the CNN is trained using labelled pixels and tested based on its ability to predict the class of pixels in unseen imagery. The ability of a model to accurately predict the class of

pixels in an unseen image is called generalisation (Goodfellow et al., 2016) and determines the transferability of the model.

CNNs were popularised in 2012 when Krizhevsky et al. (2012) won the ImageNet Large Scale Visual Recognition Challenge (ILSVRC) with a CNN called AlexNet. They have since been applied to a broad range of disciplines, improving tasks in object detection (Zhao et al., 2019), speech recognition (Abdel-Hamid et al., 2014), and numerous medical imaging applications

(Lundervold and Lundervold, 2019). They are also increasingly being used for a variety of remote sensing applications (Buscombe and Ritchie, 2018), including classification of fluvial scenes (Carbonneau et al., 2020a), land-use classification (e.g. Luus et al., 2015), and automated detection of geological features on Mars (Palafox et al., 2017). In glaciology, CNNs have achieved success in mapping debris-covered land-terminating glaciers (Xie et al., 2020), rock glaciers (Robson et al., 2020), supraglacial lakes (Yuan et al., 2020) and snow cover (Nijhawan et al., 2019). The application of deep learning models

in workflows for automated delineation of marine-terminating glacier termini has also been effective, resulting in accuracy comparable to conventional manual methods (Baumhoer et al., 2019; Mohajerani et al., 2019; Zhang et al., 2019).

## 3 Methods

### 3.1 Study Areas

#### 3.1.1 Training Area: Helheim Glacier, SE Greenland

The area chosen to train the phase one CNN in the deep learning workflow spans 68.8 x 37.2 km (Fig. 1c) and includes Helheim Glacier (66.4° N, 38.8° W), a major outlet of the south-eastern Greenland Ice Sheet (GrIS). Helheim is one of the five largest outlet glaciers of the GrIS by ice discharge (Howat et al., 2011; Enderlin et al., 2014) and has flow speeds of 5-11 km a$^{-1}$ (Bevan et al., 2012). The glacier has a 48,140 km$^2$ drainage basin (Rignot and Kanagaratnam, 2006) equivalent to ~4% of the ice sheet's total area (Straneo et al., 2016), from which several tributaries converge into a ~6 km wide terminus. There is an

extensive area of ice mélange (a mixture of sea-ice and icebergs) adjacent to the terminus where it enters Sermilik Fjord and is influenced by ocean currents (Straneo et al., 2016) (Fig. 1c). Inspection of available satellite imagery reveals that the area of mélange varies seasonally with monthly variations in extension and composition (Andresen et al., 2012, 2013). For example, our observations from February through to April 2019 show that the area of mélange was relatively small and consisted primarily of sea-ice, with fewer large icebergs in comparison to later months. Fjord waters were also dominated by sea-ice in

various stages of development with few icebergs. From May through to August 2019, the mélange area expanded to cover a



larger proportion of the fjord surface and its composition became dominated by icebergs, reflecting a change to iceberg-dominant fjord waters and a reduction in sea-ice. A gap in the mélange at the glacier terminus appeared at the beginning of July and persisted until mid-August, suggesting the presence of an active meltwater-fed glacial plume as previously observed (Straneo et al., 2011).


The glacier, fjord, and surrounding landscape provide an ideal test area for the deep learning workflow because it contains a number of diverse elements that vary over short spatial and temporal scales and are typical of other complex outlet glacier settings. These characteristics include; 1) seasonal variations in the degree of surface meltwater ponding on the glacier and ice mélange; 2) weekly to monthly changes in the extent and composition of mélange; 3) short-lived, meltwater-fed glacial plumes

which result in polynyas adjacent to the terminus; 4) sea-ice in varying stages of formation; 5) varying volumes and sizes of icebergs in fjord waters and; 6) seasonal variations in snow cover on both bedrock and ice. The resulting spectral variations over multiple satellite images in addition to potential variations resulting from changes in illumination and weather, pose a considerable challenge to image classification. However, capturing these characteristics at the scale of an entire outlet glacier image scene is important for a more efficient and integrated understanding of how numerous glacial processes interact.

**3.1.2 Test Areas: Helheim Glacier and Scoresby Sund, SE Greenland**

The deep learning workflow was trained at Helheim Glacier and then tested on two areas (Fig. 1a, b) using: 1) a previously unseen Sentinel-2 tile of Helheim Glacier and the surrounding landscape, acquired on 13/09/2019 (in-sample), and; 2) a Sentinel-2 image of the glacial landscape in the area of Scoresby Sund, ~600 km north of Helheim, which features several smaller outlet glaciers and was acquired on 01/08/2019 (out-of-sample). This area was chosen as an ideal test site because it

encompasses all the classes used in model training (including mélange which is not always present at glacier termini). Both unseen Sentinel-2 tiles used for testing were divided into nine smaller image tiles spanning 3000x3000 pixels, resulting in 18 test images for processing by the deep learning workflow.

**3.2 Imagery**

Remote sensing studies which apply deep learning to image classification usually use high resolution (sub-metre) imagery

(Sharma et al., 2017) and typically require large datasets (Krizhevsky et al., 2012). Acquiring high resolution imagery of outlet glacier landscapes can be expensive and challenging, especially over large spatial areas. Therefore, the abundance of widely available medium resolution satellite imagery (10 - 60 m), often used for remote sensing applications in glaciology, provides an ideal data source for training and testing the deep learning workflow. Here we use Sentinel-2 bands 2 (blue), 3 (green), 4 (red), and 8 (Near Infrared (NIR)) at 10 m spatial resolution to train and test our approach. The red, green, and blue bands were

chosen because they are commonly used in image classification with deep learning architectures such as VGG16, making existing, pre-trained, models easily transferable for the purpose of this study. The NIR band was chosen due to its common



use in remote sensing of glacial environments, for example in band ratios to automatically identify glacier outlines (e.g. Alifu et al., 2015).

Examination of available Sentinel-2 imagery showing the seasonal change of the glacial landscape throughout the year resulted in the establishment of seven semantic classes, including: 1) open water, 2) iceberg water, 3) mélange, 4) glacier ice, 5) snow on ice, 6) snow on rock, and 7) bare bedrock (see detailed criteria for each in Table 1). To best encompass the seasonally variable landscape characteristics and collect sufficient training data to represent intra-class variation in all seven classes, 13 cloud-free Sentinel-2 images taken between February and October 2019 were acquired (Table S1 in the Supplement). Level-

2A images were downloaded from Copernicus Open Access Hub (available at: https://scihub.copernicus.eu/dhus/#/home, last accessed: 20/07/20). The atmospherically corrected red, green, blue and NIR bands were combined into composite four band images and cropped to the training area (Fig. 1c). Two Sentinel-2 tiles of the unseen Helheim (Fig. 1a) and Scoresby Sund (Fig. 1b) study areas were also acquired, and the corresponding composite band images were created.










| Example Image of Class | Class Number and Label | Class Description | Total number of Tiles | | |
|---|---|---|---|---|---|
| | | | 50x50 (total: 354,668) | 75x75 (total: 319,292) | 100x100 (total: 293,720) |
| | 1. Open Water | Open water with no icebergs | 14,312 | 12,024 | 10,520 |
| | 2. Iceberg Water | Water with varying amounts of icebergs or disintegrated mélange | 48,668 | 44,084 | 41,212 |
| | 3. Mélange | Mixture of sea-ice, and icebergs of varying sizes | 25,540 | 23,396 | 21,192 |
| | 4. Glacier Ice | Glacier ice, with seasonally variable surface meltwater | 84,356 | 77,584 | 71,040 |
| | 5. Snow on Ice | Snow/ice with a smooth appearance | 88,412 | 79,540 | 77,004 |
| | 6. Snow on Rock | Bedrock with varying amounts of snow cover | 63,180 | 55,052 | 47,588 |
| | 7. Bedrock | Bedrock with no snow cover | 30,300 | 27,612 | 25,164 |

**Table 1: Example image samples and descriptions of each of the seven semantic classes used to train and validate the phase one convolutional neural networks in the deep learning workflow. Total number of tiles refers to the total number of tiles used for training and validation in each of the three datasets used to test model sensitivity to tile size after the tiling process described in Fig. 5. Note that the open water, mélange, and bedrock classes have the smallest representation of all classes, despite the aim of producing equally represented class samples.**




## 3.3 Classification Workflow, Model Architectures and Training

### 3.3.1 CNN-Supervised Classification

The classification workflow used here is termed CNN-Supervised Classification (CSC), and was originally developed and tested on airborne imagery (<10 cm resolution) of fluvial scenes (Carbonneau et al., 2020a). CSC is a novel two-phase workflow (Fig. 2) which uses a pre-trained CNN to replace the human operator's role in labelling training areas for the final pixel-level classification. In the first phase of the workflow, a pre-trained CNN is used to predict the classes of a tiled input image. The image tiles are then reassembled to create a class raster which is used as training data for the second model in phase two of the workflow. In the second phase, the reassembled class raster and image features are vectorised and used to train a second model specific to the input image. The predictions of the second model result in a final classified image output.

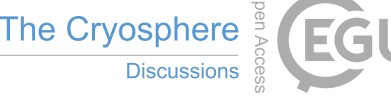

**Pre-Processing**

Cloud-free Sentinel-2 Images downloaded

Images cropped to training area and labelled according class to produce Class raster

Image and Class raster tiling and allocation to training and validation data folders (see Fig. 5)

Classes:

| 1. Open Water | 5. Snow on Ice |
| 2. Iceberg Water | 6. Snow on Rock |
| 3. Mélange | 7. Bedrock |
| 4. Glacier Ice | |

Tile sizes: 50, 75, 100 pixels

**CNN models trained and saved**

**Class Prediction**

Unseen Sentinel-2 Images tiled and 4D tensors prepared

Trained CNN loaded

**Phase 1**

Run CNN model → Reassemble tiles to produce CNN-predicted class raster

**Phase 2**

Median Filter 1x1 → CNN class raster and image features vectorised

MLP/cCNN model trained

**Final predictions and classified image**

**Figure 2: Image classification workflow showing pre-processing steps, convolutional neural network training and 2-phase final classification steps.**





### 3.3.2 Phase 1: Model Architecture and Training

For the base architecture of the pre-trained CNN used in phase one we adapt a well-established CNN called VGG16 (Simonyan

and Zisserman, 2015) which outperformed the state-of-the-art performance of AlexNet in the ILSVRC 2014. The VGG model

we use consists of five stacks of 13 2D convolutional layers which have filters with a 3x3 pixel kernel size (Fig. 3). The filter

spatially convolves over the input image to create a feature map, using the filter weights. The dimensions of the output filters

increase from 64 in the first stack of convolutional layers to 512 in the last (Fig. 3). All the convolutional layers use ReLU

activation and are interspersed with five max-pooling layers. The convolutional and pooling stacks are followed by three fully

connected (dense) layers (i.e. a normal fine-tuned neural network) without shared weights, typical of CNN architectures. This

section allows the features learned by the CNN to be allocated to a class by a final Softmax layer with the same number of

units as classes. The input image tile size for the first convolutional layer in the original VGG16 model architecture was fixed

as a 224x224x3 RGB image. However, here we test the impact of tile size by using three datasets with different tile sizes of

50x50, 75x75, and 100x100 pixels. Thus, we adjust the input image size, so it matches our three tile sizes (Fig. 3 shows an

example of a input tile size of 100), and adjust the number of input channels depending on the number of image bands used

for training (i.e. three or four).

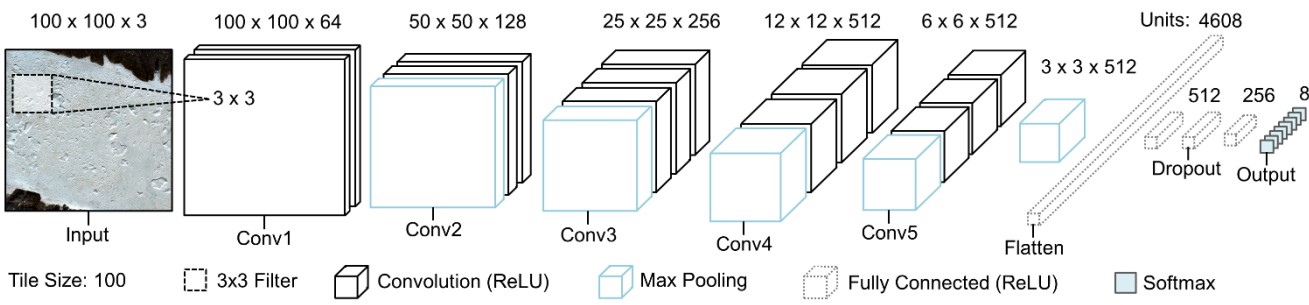

**Figure 3: Architecture of phase one convolutional neural network, adapted for three tile size datasets from the original VGG16 model architecture (Simonyan and Zisserman, 2015). Diagram shows an example using a tile size of 100 pixels. There are five stacks**

**of 2D convolutional layers which extract features from input tiles using a 3 x 3 filter. The convolutional stacks are followed by a fully connected neural network and Softmax activation for final class predictions used as localised training data for phase two models.**

We tested three approaches for training the phase one CNN using our three image tile datasets to test the sensitivity of each

approach to tile size, resulting in a total of nine trained CNNs. The three approaches of model training were as follows: 1) we

used only three image bands (RGB); 2) we used the NIR band in addition to the three RGB bands (RGB+NIR), and; 3) we

used three image bands (RGB) in combination with transfer learning (RGB+TL). The transfer learning approach trains the

model using pre-existing weights from the ImageNet database which contains over 14 million labelled images (Deng et al.,

2009). Only the weights in the final layers of the CNN are re-trained specifically to classify glacial scenes, making it quicker

to train than standard full CNN architectures (Buscombe and Ritchie, 2018). Transfer learning has been shown to decrease



training time and reduce the volume of data needed to produce similar levels of accuracy to non-transfer learning techniques (Kunze et al., 2017). As a result, with a tile size of 100 the transfer learning model had 9,572,616 trainable parameters of a total 17,207,880 trainable parameters if the VGG16 model was trained without transfer learning, and all weights were adjusted. For each of the nine models, training hyperparameters were kept constant, with training occurring over 15 epochs, with a batch size of 50 images and a learning rate of 0.0001. In all models, we used $L^2$ regularization to reduce overfitting which occurs when a model is unable to generalize between training and test data (Goodfellow et al., 2016). We also used Adam gradient-based optimisation in all model training (Kingma and Ba, 2017). Following training of the phase one CNNs, they were saved for application on unseen images in phase two without further training.

### 3.3.3 Phase 2: Model Architectures and Training

To classify airborne imagery of fluvial scenes using the CSC workflow, Carbonneau et al. (2020a) applied a pixel-based approach using a multilayer perceptron (MLP) in the second phase of the workflow, achieving high levels of accuracy (90-99%). We propose that applying pixel-based techniques to coarser resolution imagery such as Sentinel-2 data may be less effective compared to applying the workflow to high resolution imagery. We therefore adopt a patch-based approach which uses a small window of pixels to determine the class of a central pixel, as in Sharma et al. (2017). This approach is based on the idea that a pixel in remotely sensed imagery is spatially dependent and likely to be similar to those around it (Berberoglu et al., 2000). Sharma et al. (2017) use a patch size of 5x5 pixels for patch-based classification of medium resolution Landsat 8 imagery. This use of a region instead of a single pixel allows for the construction of a small CNN (dubbed 'compact CNN' or cCNN: Samarth et al., 2019) with a single convolutional layer that assigns a class to the central pixel according to the properties of the region (Carbonneau et al., 2020b). It therefore combines spatial and spectral information. Here we test both pixel- and patch-based approaches using an MLP and cCNN in the second phase of the workflow (the architectures and application of which are detailed in the following sections 3.3.3.1 and 3.3.3.2). Specifically, four patch sizes of 1x1 (pixel-based), 3x3, 7x7, and 15x15 pixels were tested. In combination with the phase one CNNs using different tile sizes, this resulted in the testing of 36 model workflows overall which were subsequently tested on in-sample and out-of-sample test images.

### 3.3.3.1 Multilayer Perceptron

For the pixel-based classification in phase two we use an MLP (Fig 4a.). An MLP is a typical deep learning model (also commonly known as an artificial neural network (ANN)) which consists of three (or more) interconnected layers (Rumelhart et al., 1986; Berberoglu et al., 2000). The first and final layers of an MLP are called the input and output layers, respectively. The layers in between are 'hidden layers' used to apply weights to the input data, which is then fed forward to units in other hidden layers (Atkinson and Tatnall, 1997). The MLP used here has five layers consisting of four fully connected (dense) layers and one batch normalisation layer (Fig. 4a). The first dense layer has the same number of input dimensions as image bands and 64 output filters. This is followed by a batch normalization layer which helps to reduce overfitting in a similar way to dropout layers, by adjusting the activations in the network to add noise. This is followed by two more dense layers with 32





and 16 filters, respectively. The final output layer in the network is a dense layer with Softmax activation and eight output filters, to match the number of output classes. For both the MLP and cCNN, model training hyperparameters were kept constant (150 epochs, learning rate of 0.001, and subsamples size of 100,000). All the layers use ReLU activation except the output layer which uses Softmax activation. Since the MLP is pixel-based, the number of parameters is smaller compared to the patch-based model, with 3,128 trainable parameters.

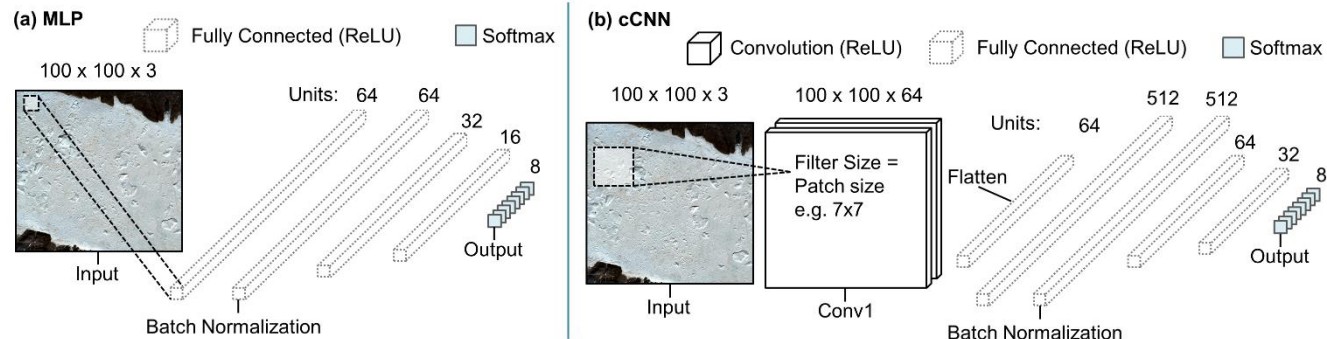

**Figure 4: Architecture of phase two models. (a) Shows the Multilayer Perceptron used for the pixel-based classification of new input images. (b) Shows the compact convolutional neural network used for patch-based classification of new input images. The size of the filter in the cCNN changes according to the patch size being tested. For example, as shown in (b) the filter size is 7 x 7 for testing a patch size of 7 pixels.**

### 3.3.3.2 Compact Convolutional Neural Network

For the patch-based classification in phase two we use a cCNN (Fig. 4b). We refer to this model architecture as a compact CNN (cf. Samarth et al., 2019) because it only contains one convolutional layer and is much smaller than conventional CNNs (Fig. 4b). This model is comprised of a 2D convolutional input layer which extracts features from the input image using a small window of pixels called a filter. The input layer has 64 filters with a kernel (window) size which is modified dependant on patch size (i.e. for testing a patch size of 7x7 pixels, the kernel size is 7) (exemplified in Fig. 4b). As with the phase one CNN, the input shape is a four-dimensional (4D) tensor determined by the patch size and number of input image bands (channels). This is followed by a flatten layer which converts the inputs into a one-dimensional feature vector to be fed into the following four fully connected (dense) layers. The first dense layer has 512 filters and is followed by a batch normalisation layer. The following three dense layers have 64, 32, and 8 filters, respectively. As with the MLP, all the layers use ReLU activation except the output layer. As with all the models used in the workflow, the final layer comprises the same number of units as output classes and results in a vector of probability scores used to predict class. The cCNN had 71,272 trainable parameters with a patch size of 3, 78,952 trainable parameters with a patch size of 7, and 112,744 trainable parameters with a patch size of 15.





### 3.3.4 Training and Validation Data Preparation

The CNNs used in phase one of the workflow were trained using image tiles which represent image subsamples of each individual class. These tiles were processed by the model in the form of 4D tensors consisting of multiple image bands (consistent with conventional data formatting designed for training CNNs for multiband image classification). To create training and validation data for the model, the composite images were manually labelled according to the seven training classes using QGIS 3.2 digitising tools. Vector polygons labelled by class number were rasterised to produce a class raster with the

same geometry as the input image. Both the input image and class raster were then tiled using a specified size (height and width in pixels) and stride (number of pixels the window moves before extracting another tile) (Fig. 5). Three different tile sizes were used to test model sensitivity and its ability to identify landscape features at the scale of the 10 m resolution imagery. This resulted in three datasets containing tile sizes of 50x50, 75x75, and 100x100 pixels (Table 1). A stride of 35 pixels was used to allow overlap between tiles, and any tiles occupied by less than 95% pure class were rejected, removing tiles containing

mixed classes. The image tiles were then rotated in increments of 90° to augment the dataset and saved to separate class folders.

Data augmentation is a common step for bolstering training datasets in deep learning, and usually entails slightly altering existing data to increase the number of training samples (Chollet, 2017). In addition to data augmentation, tile rotation allows the model to learn classes which may appear at different orientations in unseen images, for example accounting for different

glacier flow directions, providing the potential for increased workflow transferability. Each tile was normalised by 16384 (a maximum integer value drawn from satellite imagery) to reduce bit depth to a scale from 0 to 255. This adjusts the range of pixel values to make them compatible with RGB imagery for processing by the CNN. The tiles were divided into training and validation datasets whereby 95% of tiles were randomly allocated to a training data folder and the remaining 5% were allocated to a validation data folder (Fig. 5). It is common when training deep learning models for image classification applications to

have an 80/20% split of training and validation data (Carbonneau et al., 2020a). However, here a 95/5% split is appropriate as the 'in-sample' data we used to test the workflow is a new satellite image of the training area and surrounding landscape, previously unseen by the model during training, making it a more stringent test. Overall, this resulted in three datasets containing 354,768 tiles of 50x50 pixels, 319,292 tiles of 75x75 pixels, and 293,720 tiles of 100x100 pixels for training and validating the phase one CNNs (Table 1). These datasets were extracted from only 13 cropped images of Helheim Glacier and

are much larger compared to those used in previous work to train and validate CNNs for glacier boundary delineation. For example, Mohajerani et al. (2019) used only 123 tiles of 152x240 pixels obtained from three different glacier study sites. Baumhoer et al. (2019) opted for larger tile sizes and used a dataset of 19,576 tiles of 780x780 pixels derived from 38 scenes from four study sites. Finally, Zhang et al. (2019) used 36,414 tiles with a larger size of 960x720 pixels using 75 images from one glacier.







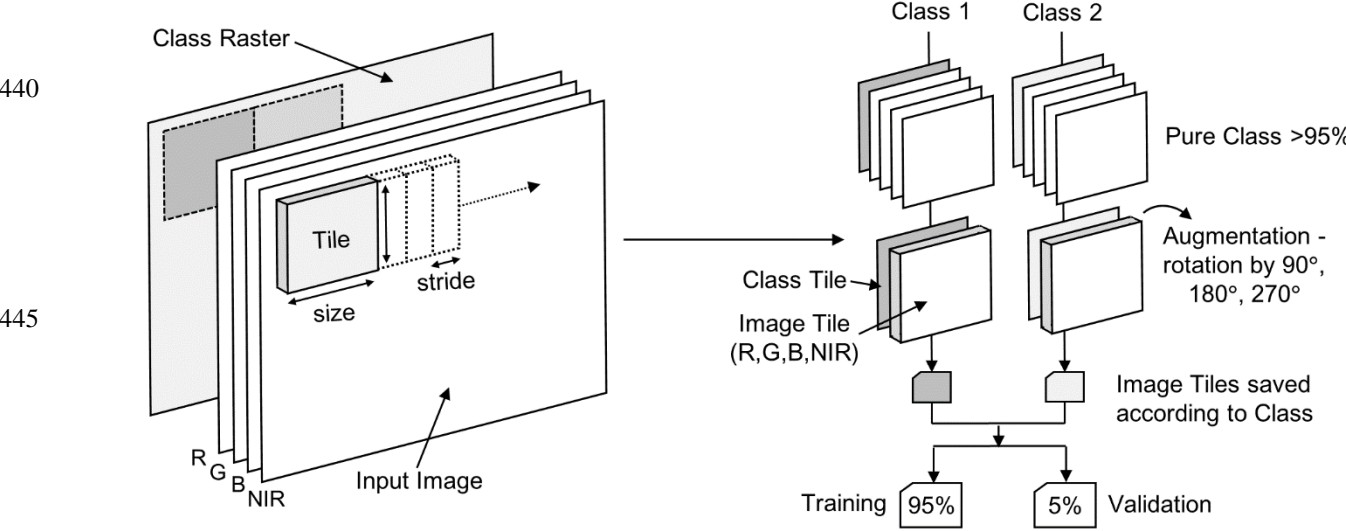

**Figure 5: Conceptual diagram of tiling process used to create training and validation data. A specified tile size (of 50, 75 or 100 pixels) and stride (of 35 pixels) are used to extract tiles from the class raster and image bands. These tiles are filtered and augmented and saved to individual class folders using a 95/5 % split for training and validation data.**

### 3.4 Sensitivity Analysis: Training Epochs

Since CNNs are sensitive to the number of epochs used in training, we applied the epoch tuning method used by Chollet (2017) with 95% of our data used to train the model and 5% used for validation. The term epochs refers to the iterations over training data in the CNN (Chollet, 2017). Each epoch is used to adjust weights and improve accuracy in the CNN based on training loss. In all models we used categorical cross entropy as the loss function. We ran the VGG16 models for 25 epochs and saved

training accuracy, training loss, validation accuracy, and validation loss for each individual epoch. Similarly, we ran the MLP and cCNN models for 500 epochs and saved the same values. These were plotted against number of epochs (Fig. S1 and S2 in Supplement). The number of epochs used to train the final set of models was then determined by the point of divergence between training and validation data. Where a gap between training and validation data appears, the model begins to overfit and its ability to generalise is reduced. The epoch tuning graph of the VGG16 model (Fig. S1) begins to diverge slightly after

15 epochs, so the model was trained for 15 epochs for optimal accuracy and training time. The epoch tuning graphs for the phase two models revealed that the optimum number of training epochs was 150 (Fig. S2).

### 3.5 Model Performance

Model performance is often measured by classification accuracy (the number of correct predictions divided by the total number of predictions). However, some models require more robust measures of accuracy which also take into account confusion





between predicted classes (Goodfellow et al., 2016; Carbonneau et al., 2020a). We use an F1 score as the primary performance metric for the models used in both phases of the classification workflow. The F1 score is defined as the harmonic mean between precision ($p$) and recall ($r$):

$$F1 = \frac{2pr}{p + r}$$

*(1)*

where precision finds the proportion of positive predictions that are actually correct by dividing the number of true positives by the sum of both true (correct) positives and false (incorrect) positives. Recall finds the proportion of positive predictions that were identified correctly by dividing the number of true positives by the sum of true positives and false negatives (misidentified positives). Thus, the inclusion of recall provides a metric which represents confusion between class predictions and takes into account class imbalance (Carbonneau et al., 2020a). F1 scores range from 0 to 1 with 1 being equivalent to

100% accuracy. Carbonneau et al. (2020a) used classification results from 862 images to compare F1 and accuracy. They found that they are closely correlated (accuracy = 1.03F1 +4.1% with an $R^2$ of 0.96), with F1 and accuracy converging at 100%. We plotted F1 scores against patch and tile sizes to show workflow sensitivity for each of our three training approaches, as well as confusion matrices to show agreement between predicted classes and manually delineated validation data in the final classification outputs (see Figs. S5 to S28 in Supplement).


We also use Cohen's Kappa as a secondary performance metric which is a coefficient of agreement (Cohen, 1960). This compares the agreement between the model class predictions and manually determined classes (validation data). Cohen's Kappa accounts for the chance occurrence of true positives in class predictions (i.e. correctly guessing the class). It is a useful complement to metrics such as accuracy and F1 because it better reflects the performance of models with class imbalance. It

removes the problem of overshadowing in prediction performance for a smaller class by that of a larger class. Cohen's Kappa is a normalised statistic, so it ranges from 0-1. A set of arbitrary thresholds were determined by Landis and Koch (1977) to interpret the agreement statistic (Table 2).

| Cohen's Kappa Statistic | Strength of Agreement |
|:---:|:---:|
| <0.0 | Poor |
| 0.0 – 0.2 | Slight |
| 0.21 – 0.4 | Fair |
| 0.41 – 0.6 | Moderate |
| 0.61 – 0.8 | Substantial |
| 0.81 – 1.0 | Almost Perfect |



**Table 2: Arbitrary thresholds used to interpret Cohen's Kappa measure of agreement (Landis and Koch, 1977).**





### 3.6 Comparison to Traditional Mapping Techniques

For a comparison of effectiveness between the CSC workflow, and pixel-based techniques such as band ratio methods, we also classified a test image tile of Helheim Glacier using a band ratio technique. To create the band ratio image, we divided the Sentinel-2 band 4 (red) by band 11 (Shortwave Infrared) (Paul et al., 2016). We used a series of thresholds to classify the resulting band ratio image into three classes including glacier ice, snow on ice and bedrock. We were unable to classify the band ratio image using all seven classes utilised in the CSC workflow. This is because the band ratio method did not detect changes between all the different classes such as mélange, iceberg water and open water. For comparison to our CSC classifications, we produced an overall F1 score for the resulting band ratio classification using the same validation labels used to produce F1 scores for the CSC classification.

### 4 Results

### 4.1 CNN-Supervised Classification

### 4.1.1 Performance of Phase 1 Convolutional Neural Networks and Tile Size Sensitivity

The performance of the phase one VGG16 models in classifying unseen Sentinel-2 image tiles of the Helheim and Scoresby Sund study areas are shown in Fig. 6. With the exception of the transfer learning model (RGB+TL) in the Scoresby Sund study area, all models produced accurate classifications (F1 Scores ≥ 88%). The best performing model on the Helheim study area was the RGB transfer learning model (RGB+TL) with a tile size of 50 pixels. The model predictions produced a classification with an overall F1 score of 93% (Fig. 6a) and Kappa value of 0.9 (Fig. S3). This indicates that the model class predictions are highly accurate and have almost perfect agreement with manually delineated validation data (see Table 2). The highest performing models for the Scoresby Sund study area were the RGB models which scored slightly lower F1 scores of 90% irrespective of tile size (Fig. 6b). This shows that the model produces slightly improved classification performance on in-sample data compared to out-of-sample data. However, the RGB model performance on the Scoresby Sund image remains high and indicates that the phase one model is transferable to outlet glaciers not used in training.

Overall, the performance of non-transfer learning models does not appear to be greatly sensitive to tile size, with RGB and RGB+NIR models resulting in F1 scores ranging from 90 to 92% for in-sample (Helheim) data and 88 to 90% for out-of-sample (Scoresby) data. However, the transfer learning models were greatly impacted by tile size for both test areas, with tile sizes of 75 and 100 pixels producing lower F1 and Kappa scores compared to models trained with a tile size of 50 pixels (Fig. 6a and Fig. S3). The transfer learning models also performed substantially worse on out-of-sample data (Fig. 6b). The addition of the NIR band in both study areas did not appear to improve classification results. In summary, while the best performing phase one CNN for in-sample data used transfer learning, the transfer learning approach was highly sensitive to tile size and did not perform well on out-of-sample data, suggesting it is less transferable compared to non-transfer learning approaches of

model training. Additionally, both models trained using RGB and RGB+NIR tiles were only slightly sensitive to tile size, but the addition of the NIR band did not improve model performance, suggesting that the RGB models are the most transferable while providing high levels of classification accuracy.

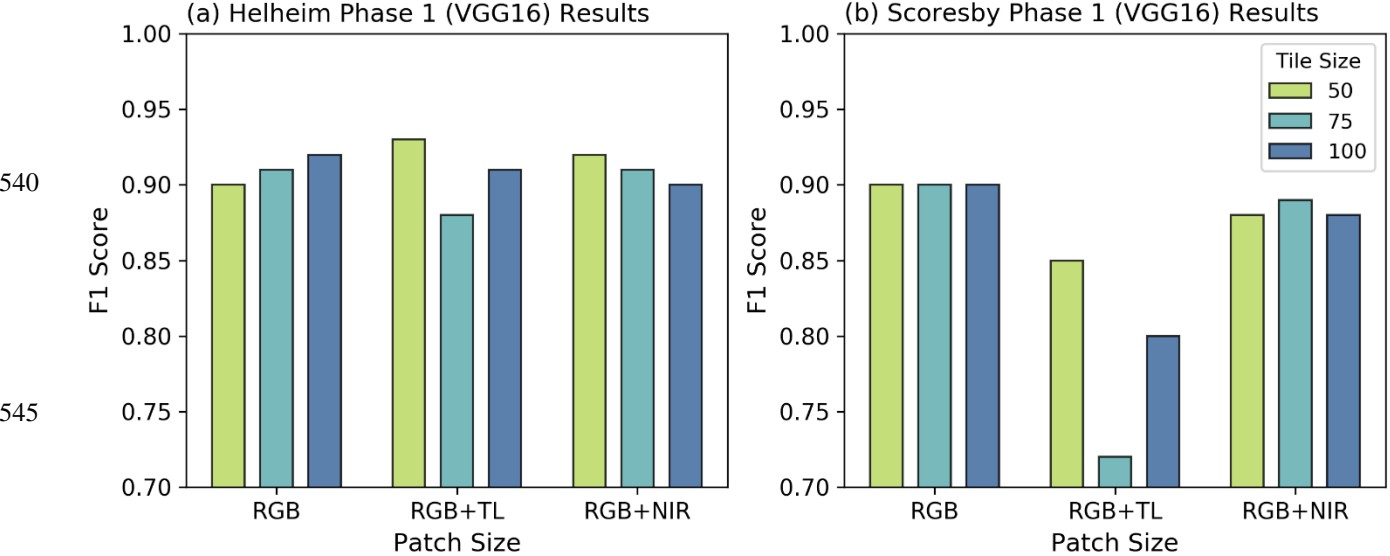

**Figure 6: The F1 scores of the phase one (VGG16) model classifications used to produce training data for phase 2 of the CSC workflow. Showing results for (a) the Helheim test area (in-sample) and (b) the Scoresby Sund test area (out-of-sample). Note the low sensitivity of RGB and RGB+NIR models to tile size (with a range in F1 scores of 2 % for both (a) and (b)). Note also the high sensitivity of transfer learning approaches to tile size and lower transferability to out-of-sample data compared to non-transfer learning approaches.**

### 4.1.2 Performance of Phase 2 Models and Patch Size Sensitivity

#### 4.1.2.1 Helheim (In-sample)

Figure 7 shows the overall F1 scores of the CSC (CNN + MLP/cCNN) results, demonstrating the impact of patch size. In general, the results of applying CSC to the Helheim study area showed a clear sensitivity to patch size with a patch size of 1 pixel yielding lower F1 scores and Kappa values than larger patch sizes in all models. Larger patch sizes of 3, 7, and 15 pixels either produced F1 scores consistent with phase one CNN outputs or improved upon classification performance by 1 to 2%. A patch size of 7 pixels yielded the best results in all models with the highest F1 scores of 92% in the RGB+NIR model (Fig. 7e), 93% in the RGB model (Fig. 7a), and 95% in the RGB transfer learning model (Fig. 7c).







**Figure 7:** The F1 scores of the phase two classifications following the CSC workflow for the Helheim test image (a, c, e) and Scoresby test image (b, d, f). Note in some cases phase two results outperform phase one results. One prominent exception is the pixel-based approach for in-sample data. The patch-based approach performs well for in-sample data, with a patch size of 7 creating optimal results. The pixel-based approach performs better on out-of-sample data compared to in-sample data.





Specifically, the CSC results of the RGB models yielded F1 scores from 82 to 93% (Fig. 2a) and Kappa values of 0.75 to 0.89

(Fig. S4). RGB models with tile sizes of 75 and 100 pixels scored highest and had correspondingly high Kappa scores (≥0.8: see Fig. S4). In terms of patch size, the RGB models using a cCNN patch size of 7 improved on the results of the phase one CNNs by 1%. RGB models using a cCNN patch size of 3 and 15 also performed well, either producing the same F1 score as phase one CNNs or improving classification results (Fig. 7a). The CSC results of the RGB transfer learning models yielded F1 scores from 84 to 95% (Fig. 7c) and Kappa values of 0.85 to 0.92 (Fig. S4). RGB+TL models with a tile size of 50 were

highest performing with F1 scores of 94 to 95% for patch sizes of 3 to 15 pixels (Fig. 7c). As with the RGB models, the use of a cCNN with a patch size of 7 was the best, consistently improving on phase one results by 2%. The RGB+NIR models had F1 scores ranging from 85 to 92% (Fig. 7e) and Kappa values of 0.77 to 0.88 (Fig. S4). The results of phase one RGB+NIR models with a tile size of 50 were not improved by the addition of a patch-based cCNN. However, RGB+NIR models with tiles sizes of 75 and 100 and a cCNN patch size of 3 and 7 were consistent with or improved upon phase one classification

results. As with the pixel-based approach, the phase two model which used a patch size of 15 did not improve phase one RGB+NIR classification results. Overall, this suggests that the pixel-based CSC workflow is outperformed by the patch-based CSC workflow for in-sample classification, with a patch size of 7 pixels producing the optimal results. It also suggests that with optimal patch size, phase one model classifications are improved upon by phase two model results.

Figure 8 shows the in-sample CSC outputs for the best performing phase one models using RGB (Fig. 8c), RGB with transfer learning (Fig. 8d), and RGB+NIR (Fig. 8e) training approaches. All models in the figure used a cCNN patch size of 7 pixels and are applied to a 3000x3000 pixel image tile of Helheim glacier (Tile 5 of 9 extracted from the test image). The RGB model produced an F1 score of 94% (Fig. 8c), while the RGB model with transfer learning (Fig. 8d) and the RGB+NIR model (Fig. 8e) both produced F1 scores of 97%. Visual comparison between the results suggests only small variations in classification

outputs, corresponding to small variations in F1 scores (on the scale of 1 to 3%). Confusion matrices which show the agreement between model-predicted classes and manually delineated classes for the CSC outputs can be found in supplementary materials (Figs. S5 to S28). Taken together, these results indicate that for in-sample data the patch-based (CNN + cCNN) CSC workflow produces the best results. Specifically, the best performing model used a phase two cCNN with a patch size of 7 pixels.

**4.1.2.2 Scoresby Sund (Out-of-sample)**

In contrast to the Helheim study area, the performance of CSC in out-of-sample data was high for the pixel-based approach, in most cases with identical or improved F1 scores compared to the larger patch-sizes (Fig. 7). Models using smaller patch sizes (1, 3, and 7) performed slightly (1 to 3%) better than those with a patch size of 15 (except for RGB and RGB+NIR models with a tile size of 50 pixels and the RGB+NIR model with a tile size of 100 pixels). It is important to note that for each of the

635 nine individual models tested on out-of-sample data, F1 scores only varied by up to 3% for the four different patch sizes. The MLP and cCNN outputs also notably outperformed phase one classification results in most models, by up to 4% (Fig. 7).







**Figure 8: Best performing CSC results for tile 5 of 9 from the Helheim study area (07/08/2019). (a) RGB input image (composite Sentinel-2 bands 4, 3, and 2). (b) Validation raster composed of manually digitised 'ground truth' polygons. Showing workflow outputs using (c) the RGB model (tile size: 100 pixels, patch size: 7 pixels), (d) the RGB model with transfer learning (tile size: 50 pixels, patch size: 7 pixels), and; (e) the RGB+NIR model (tile size: 50 pixels, patch size: 7 pixels). Note all models produce highly accurate classification outputs with small variations between outputs and minimal noise.**





For the Scoresby Sund area the RGB models were the highest performing overall (Fig. 7b), with F1 scores ranging from 90 to 93%. Kappa values also showed highest agreement for the RGB models, with a value of 0.9 for most RGB models with tile sizes of 75 and 100 (Fig. S4). In terms of patch size, the RGB model performed best with patch sizes of 1, 3, and 7. While the transfer learning approach improved model performance for the Helheim study area, as with the phase one CNN, its performance in the Scoresby Sund area (out-of-sample) was substantially worse, with F1 scores ranging from 74 to 89% (Fig. 7). The transfer learning approach was also more sensitive to tile size than other models, with a tile size of 50 pixels yielding the highest F1 scores of 88 to 89%, 75 pixels yielding 74 to 75%, and 100 pixels yielding 81 to 84%, mirroring phase one model results. However, it showed very little sensitivity to patch size. There was no variation in F1 score between patch sizes of 1, 3, and 7 for models trained on tile sizes of 50 and 75, and a patch size of 15 produced F1 scores 1% lower than smaller patch sizes. Despite this, the addition of the phase two model improved phase one transfer learning classification results consistently for the out-of-sample data. RGB+NIR models had F1 scores ranging from 88 to 92% (Fig. 7f) and Kappa values from 0.81 to 0.88 (Fig. S4), with a tile size of 75 yielding the best results. As with all models tested on out-of-sample data, the RGB+NIR models showed limited sensitivity to phase two patch size, but the pixel based approach (patch size: 1) was consistently 1 to 2% better than the patch based approach. Overall, these results show that out-of-sample data is less sensitive to patch size compared to in-sample data, with the pixel-based approach performing substantially better in out-of-sample data. The results also suggest that phase two model predictions are highly dependent on the quality of classification outputs resulting from phase one predictions which are subsequently used as localised training data.

Figure 9 shows a visual comparison of the out-of-sample CSC outputs for the best performing RGB, RGB with transfer learning, and RGB+NIR models when used on a 3000x3000 pixel image tile extracted from the Scoresby Sund area (Tile 8 of 9 extracted from the test image). Figure 9c shows the output of the RGB model, with an F1 score of 97%. The transfer learning model produced an F1 score of 89% (Fig. 9d) while the RGB+NIR model produced an F1 score of 94% (Fig. 9e). In the case of the example tile, most confusion appears to occur between open water and iceberg water classes (Fig. 9). In general, confusion matrices for the models show that the models with the lowest performance experience confusion between one or more classes (e.g. Figs. S18-28). For example, the application of the transfer learning model with a tile size of 75 to out-of-sample data resulted in confusion between open water and iceberg water classes, as well as confusion between snow on rock and glacier ice classes (Figs. S21-24). Since phase one results are used to train phase two models, high amounts of class confusion in phase one models can be transmitted to phase two results. However, some class confusion in phase one is overcome in phase two, as indicated by the consistent improvement of phase two results over phase one classifications.

705

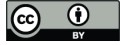

**Figure 9: Best performing CSC results for tile 8 of 9 extracted from the Scoresby Sund study area (01/08/2019). (a) shows the RGB input image (composite Sentinel-2 bands 4, 3, and 2); and (b) shows the validation raster composed of manually digitised 'ground truth' polygons. Showing workflow outputs using: (c) the RGB model (tile size: 100 pixels, patch size: 3 pixels), (d) the RGB model with transfer learning (tile size: 50 pixels, patch size: 1 pixel), and; (e) the RGB+NIR model (tile size: 75 pixels, patch size: 1 pixel). Note that most class confusion occurs between open water and iceberg water classes.**





## 4.2 Comparison of CNN-Supervised Classification to other Traditional Classification Methods

Figure 10 shows a visual comparison between a traditional band ratio technique (as described in Paul et al. (2016)) and the result of the CSC workflow using the best performing model on tile 5 of the 9 tiles extracted from the test image of Helheim. CSC successfully identifies areas of mélange, glacier ice and iceberg rich fjord waters as different classes (Fig. 10b). The band

ratio method allows clear identification of rock, and land terminating ice margins. However, the technique struggles to distinguish boundaries between glacier ice, mélange, and iceberg water (Fig. 10c). In the example shown, the abundant spectral variation and noise makes using a series of thresholds to extract margins in a mélange-filled fjord almost impossible. This is reflected by an F1 score of 53% for the band ratio technique which is substantially outperformed by CSC with a corresponding F1 of 97%. This comparison and the transferability of the CSC workflow suggests it is more robust than traditional techniques

and does not rely on the requirement of identifying threshold values to extract classes.







**Figure 10: Comparison of methods used on tile 5 of 9 (a) extracted from the Helheim study area (07/08/2019), including (b) Validation labels used to create F1 scores, (c) the CSC classification, and; (d) a band ratio classification using Sentinel-2 bands 4 (red) and 11 (SWIR). Note that only three classes could be extracted from the band ratio image due to significant noise and no contrast between glacier ice, mélange, and iceberg water classes.**



## 5 Discussion

### 5.1 Classification Workflow Performance and Comparison to Previous Work

The results reported here demonstrate state-of-the-art multi-class satellite image classification of complex outlet glacier image scenes using deep learning. The CSC workflow adapted for glacial settings in Greenland produced mean F1 scores up to 95% for in-sample test imagery and 93% for out-of-sample test imagery, with corresponding Kappa values of 0.92 and 0.9, respectively. Our method creates multi-class outputs in contrast to the binary classification outputs used by Baumhoer et al. (2019) and Zhang et al. (2019) for automated delineation of marine-terminating ice fronts. Despite this difference in output classification type, mean F1 scores of classifications by Baumhoer et al. (2019) were 89 to 90% for in-sample training sites and 90 to 91% for out-of-sample test sites, suggesting our workflow advances the state-of-the-art in image classification of complex marine-terminating glacial environments using deep learning. In terms of other glacial settings, for debris-covered land-terminating glaciers Xie et al. (2020) also produced binary classifications, with F1 scores up to 94% using a CNN trained and tested on image scenes of glaciers in the Karakoram region. Xie et al. (2020) also tested a transfer learning approach using the model initially trained for the Karakoram region, with weights adjusted based on training data from glaciers in Nepal, with resulting F1 scores of up to 90%. In addition to advancing the state-of-the-art for marine-terminating glacial settings, the multiclass outputs of our workflow widen the scope of image classification for a variety of research applications, beyond just automated delineation of calving fronts from binary classification outputs. Moreover, the ability of our deep learning workflow to classify images previously unseen by the model for both training and testing areas to a similarly high level of accuracy suggests good generalisation, and highlights the transferability of CSC to other marine-terminating outlet glacier environments in Greenland and elsewhere.

To evaluate our workflow, we tested three different approaches of training the phase one CNNs using two different band combinations and a transfer learning technique. We found that the addition of the NIR band did not significantly improve classification accuracy. Further testing of alternative band combinations or the addition of different satellite data types (e.g. SAR data) may be advantageous. However, we find that using RGB bands produces satisfactory results and adding additional image bands is likely to increase processing time without necessarily improving on the overall accuracy, as also suggested by Xie et al. (2020). In terms of transfer learning, the technique has been applied successfully in previous image classification studies which use remotely sensed satellite imagery, allowing reduced training times for smaller datasets (Hu et al., 2015; Pires de Lima and Marfurt, 2020). These studies highlighted that CNNs pre-trained on ImageNet data may be transferable to remote sensing imagery by fine-tuning the last layers in the CNN for dataset specific feature extraction, regardless of disparities in input image properties (e.g. angle of acquisition). However, Pires de Lima and Marfurt, (2020) recognise that, in contrast to training all the layers of a CNN on remote sensing data, the difference between ImageNet data and remote sensing data in some cases has resulted in transfer learning techniques overfitting and reducing the ability of models to learn. We found that



while transfer learning performed exceptionally well for in-sample data, its performance degraded substantially when applied to out-of-sample test imagery, suggesting reduced transferability. The strong performance of the transfer learning approach on in-sample data supports findings that it is a powerful deep learning tool (Xie et al., 2020). However, we suggest that the high-

level features representative of diverse, seasonally variable image elements and classes are not as successfully detected using transfer learning in comparison to full CNN training, indicating that transfer learning techniques would require further efforts to fine-tune for improved transferability to marine-terminating outlet glacier environments.

Our results build on the work of deep learning-based classification methods for glacier delineation (Baumhoer et al., 2019;

Mohajerani et al., 2019; Zhang et al., 2019; Xie et al., 2020), with several key innovations and variations of note. Firstly, the volume, type, and number of input channels of training data used in this workflow differs from those of previous work. In terms of training data volume, we use fewer training images (i.e. 13) compared to the number of training images used by Baumhoer et al. (2019), Mohajerani et al. (2019), and Zhang et al. (2019) (i.e. 38, 123, and 75, respectively) but produced larger volumes of smaller tiles to train the phase one models. Goodfellow et al. (2016) note that, as a general rule, each class

should contain at least 5,000 samples to reach satisfactory performance, but models can reach and exceed human-level performance when trained on at least 10 million samples. With this in mind, the number of labelled samples produced by manually labelled training images and data augmentation in the datasets used here (< 360,000) makes them relatively small. However, in comparison to pre-trained models such as VGG16 which were trained on the ImageNet database using over 1000 classes, our adapted VGG16 architecture only uses seven classes, and therefore can be trained sufficiently with 'only' a few

100 thousand samples. This suggests that relatively few images are needed to produce highly accurate image classifications using our workflow, reducing the time required for initial creation of manually labelled training data. Furthermore, our workflow does not require the same pre-processing steps such as manually rotating images so that glacier flow directions are consistent or cropping input images to a specified buffer width encompassing glacier calving fronts as in Mohajerani et al. (2019).


In relation to the type of data used to train the deep learning models, Baumhoer et al. (2019) and Zhang et al. (2019) used Sentinel-1 SAR data. Specifically, Baumhoer et al. (2019) used 4D inputs, incorporating different SAR polarisations with the addition of a digital elevation model (DEM) to train the FCN. In contrast, Mohajerani et al. (2019) used Landsat 5, 7 and 8 imagery for training, in particular using the 'green' band from Landsat 5 data and 'panchromatic' band from Landsat 7 and 8

data. Here we used Sentinel-2 optical data which is generally easier to process in comparison to SAR data and requires less specialised knowledge for pre-processing. For example, common pre-processing steps to implement SAR data include noise removal, radiometric calibration, and geocoding correction (in addition to training area cropping and tiling for model training) (Baumhoer et al., 2019; Zhang et al., 2019). Using L2A Sentinel-2 imagery eliminates the need to incorporate DEM data and removes SAR pre-processing steps from the deep learning workflow, allowing cloud-free imagery to be downloaded, cropped,

and tiled more quickly.





In terms of input dimensions, Zhang et al. (2019) and Mohajerani et al. (2019) used one-dimensional (1D) training inputs while Baumhoer et al. (2019) used 4D inputs. The input channels of the CSC workflow were multi-dimensional (i.e. 3 or 4 input bands) analogous with those of Baumhoer et al. (2019) but with different input data types (i.e. multispectral data vs. SAR and

DEM data). At the opposite end of the spectrum, Xie et al. (2020) used 17 input channels, incorporating all 11 Landsat 8 bands, a DEM, and five layers derived from the DEM. Xie et al. (2020) note that using fewer input channels in experimental CNN training resulted in lower levels of accuracy. However, despite the large difference in input dimensionality between our CNN and that of Xie et al. (2020), resultant F1 scores show that the use of only three Sentinel-2 bands produces classification with similar and higher levels of accuracy. However, it is important to note that we applied the CSC workflow to a markedly

different glacial landscape compared to Xie et al. (2020). Nevertheless, our results show that using only three input Sentinel-2 bands is sufficient for producing highly accurate classifications in scenes containing complex marine-terminating glaciers. Additionally, since the best results of this study were achieved using only three RGB bands, this suggests that our approach mimics human visual performance, as humans do not require 17 input bands to classify a landscape.

Likewise, further variations in comparison to previous work are apparent in the model architectures, number of models used, and training approaches tested for classification. All previous deep learning classification methods for marine-terminating glacial environments (Baumhoer et al., 2019; Mohajerani et al., 2019; Zhang et al., 2019) use the U-Net architecture (Ronneberger et al., 2015). Whilst U-Net architectures have reached state-of-the-art performance in computer vision tasks, their application in complex natural landscapes is not necessarily optimal given the intrinsic assumptions of U-Net models.

For example, U-Net architectures perform exceedingly well at delineating people in imagery (Xie et al., 2018; Wang and Bai, 2019). In such cases, skin colour and clothing colour must not be considered as identifying features. However, in Earth Observation (EO) images of natural landscapes, there is a much stronger correlation between colour and landform. Furthermore, the U-Net architecture will learn shapes that have a limited variability of both form and scale. For example, people have similar dimensions in imagery used in self-driving vehicles and their location in the image is limited to a horizontal

zone across the field of view. In contrast, natural landforms can vary in scales over several orders of magnitude and be located anywhere in an image. We therefore argue that more evidence is needed before we consider the use of U-Net architectures as the *de facto* algorithm for glacial landscape classification. Moreover, our results show that a deep learning approach based on a combination of local spectral and spatial properties determined by a compact CNN architecture has exceeded the results derived from U-Net architectures.


In contrast to previous work, we also assessed the workflow performance in comparison to a traditional band ratio method for classification of an image containing a marine-terminating glacier (Fig. 10). The results show that CSC is better at identifying classes which are spectrally similar such as mélange, glacier ice and iceberg water. This suggests that the method outperforms traditional pixel-based classification techniques, similar to findings from classification of fluvial image scenes (Carbonneau et



al., 2020a). CSC is also more transferable because it can be used on new images without further training and does not require additional steps to determine an optimal threshold value for outlining class boundaries. Moreover, the method has the ability to pick out textures and patterns in the same way a human operator would, irrespective of variations in illumination, weather conditions or seasonal changes to the landscape and individual classes. This also highlights the benefit and transferability of CNNs over purely pixel-based techniques for classifying complex image scenes with substantial seasonal variations.

## 5.2 Tiles Size Sensitivity

The impact of tile size (height and width of image samples used for training and validation) on model performance was also evaluated. Careful choice of tile size is important, as for models to learn the features which represent diverse image elements, class representative features need to fit within an individual tile. It is also important to consider that the number of tiles produced to compose a training dataset changes based on tile size. With the same source imagery, a large number of small tiles can be produced compared to fewer larger tiles (e.g. Table 1). Thus, the selection of tile size is dependent on the desired

information content of a tile and the number of tiles needed to sufficiently train a CNN. It was for this reason that tiles sizes of 50, 75 and 100 pixels were tested. We found that non-transfer learning phase one CNNs were not substantially sensitive to tile size, with models trained on all three tile sizes producing F1 scores within a 2% range for both in-sample and out-of-sample test data. Following the full CSC workflow, the RGB models (without transfer learning) trained on larger tile sizes produced

slightly better classifications with tile sizes of 100 and 75 outperforming tile sizes of 50 pixels by up to 3%. This suggests that using fewer larger tiles (e.g. size of 100 pixels) slightly improves RGB model performance, specifically for the scale of features in outlet glacier landscapes in Greenland. In contrast, the transfer learning phase one CNNs had increased sensitivity to tile size, producing F1 scores with a range of 5% for in-sample test data and 13% for out-of-sample test data. This was mirrored in the CSC classifications, with large differences in F1 scores depending on tile size, whereby the smallest tile size of 50 pixels

produced the best results, but model performance deteriorated with tiles sizes of 75 and 100 pixels. It is interesting to note that the transfer learning technique benefited from using a larger number of smaller tiles compared to the preferred smaller number of large tiles for the fully trained CNN. These results suggest that, for classification of outlet glacier landscapes, fully trained CNNs are more invariant to tile size for both in- and out-of-sample data, whereas transfer learning models produce a larger variability of F1 scores for different tile sizes, especially when applied to out-of-sample data. This supports our assertion that

the most transferable CSC workflow for outlet glacier image classification uses a phase one CNN with all weights trained using RGB bands and larger tile sizes (of 75 or 100 pixels).

## 5.3 Patch Size Sensitivity

In addition to testing the influence of tile size during training for phase one CNNs, we tested the sensitivity of phase two model performance using pixel- and patch-based techniques applying four patch sizes of 1x1 (pixel-based), 3x3, 7x7, and 15x15

pixels (patch-based). The reason for testing pixel- and patch-based techniques is due to our use of medium resolution imagery which tends to have spectral variations across images, making it difficult to distinguish class from the spectral characteristics





of a pixel alone (Maggiori et al., 2016). We proposed that adopting a patch-based technique which includes contextual information surrounding a pixel would aid classification of complex and seasonally variable outlet glacier landscapes, as it has in other applications (Sharma et al., 2017) and found that this was true, especially for in-sample test imagery. The best
performing patch size may also vary depending on the type of medium resolution satellite imagery (Sharma et al., 2017) making it an important testable parameter. When CSC was applied to in-sample test data, the workflow performance was clearly sensitive to patch size, with the pixel-based approach producing classifications with lower F1 scores compared to the patch-based technique. We found that the optimal patch-size for in-sample test data was 7x7 pixels. The in-sample results support the hypothesis that pixel-based approaches do not perform as well on medium-resolution imagery compared to the
patch-based approach. This also validates similar findings that patch-based CNNs outperform standard pixel-based neural networks and CNNs (Sharma et al., 2017). In contrast, for out-of-sample data, the pixel-based approach performed substantially better than for the in-sample test data, and smaller patch sizes of 1, 3, and 7 generally outperformed a larger patch size of 15. However, in general CSC results for out-of-sample data were less sensitive to patch size, producing a range of F1 scores that varied by only 3 % between all four patch sizes (per the three individual models). We therefore suggest that it
would be beneficial to test a range of patch sizes before applying the workflow to a new dataset.

## 5.4 Limitations, Transferability, and Implications for Future Work

The performance of the CSC workflow is dependent on the success of the pre-trained CNNs to identify image-specific training areas in phase one. The performance of the phase one models can be impacted by the size and class representation of training data. We note that the data used to train the phase one CNNs were extracted from only one outlet glacier in Greenland, and
that producing a larger training dataset from a wider array of imagery from outlet glacier settings may be beneficial to increase model performance and transferability in future work. The lack of a benchmark dataset specifically for outlet glacier settings means that the application of deep learning in this field initially relies on labour-intensive manually labelling training data. Despite this limitation, our deep learning workflow produces highly accurate classifications for both images of the glacier used in training and of different locations not 'seen' by the model. Furthermore, once the phase one models are trained and weights
are saved, no further training is required to apply the workflow to other marine-terminating outlet glaciers.

In addition to the size of the training dataset, class imbalance can impact model performance. Marine-terminating outlet glacier environments have classes which are naturally less abundant in imagery. For example, there are smaller areas of mélange compared to glacier ice or snow covered ice in most full satellite scenes, so despite efforts to balance the training dataset,
certain classes such as open water (without icebergs), mélange, and bedrock were represented by a smaller number of training samples compared to more prominent classes (Table 1). This can lead to confusion between classes as was found in some of our experiments. In models with lower performance, confusion occurred between open water and iceberg water classes, as well as between bedrock and snow on bedrock classes. Furthermore, class imbalance may be problematic for classifying large images which contain only small areas of a single class. For example, when applying CSC to a large image (e.g. an entire



Sentinel-2 tile) of an outlet glacier setting, if only a small area of mélange was classified following application of the phase one model, only a small proportion of image data can be used for image-specific training in phase two. Consequently, this may result in increased class confusion if the phase two model removes the class from the output classification, if including it will increase loss. However, in the case of the smaller 3000x3000 pixel test tiles used in this study, confusion between classes in output classifications was minimal for most of the 36 models tested.


We also note that there are numerous hyperparameters (e.g. learning rate, batch size, etc.) that could be tested and tuned for improved workflow implementation. Variations in such parameters are likely to impact model performance but require significant time to test and substantially increase the dimensions of model outputs (Carbonneau et al., 2020a). In addition, we only tested one model architecture for the phase one pre-trained models. VGG16 has a relatively simple architecture but the

state-of-the-art for image classification is constantly evolving, with consistent and rapid development of new CNN architectures. As a result, there are a myriad of variations in CNN characteristics that could be tested in future work, such as CNN depth and filter sizes. Moreover, other well-established pre-trained model architectures such as GoogLeNet (Szegedy et al., 2014), and NASNet (Zoph et al., 2018) have also been successfully applied to remote sensing applications (Ostankovich and Afanasyev, 2018; Carbonneau et al., 2020a) and could be explored for use in the CSC workflow for outlet glacier image

classification. Thus, there are several avenues for expanding the use of deep learning for image classification of marine-terminating outlet glacier landscapes in future work. Nevertheless, the implications of this study suggest that our adapted CSC workflow is transferable and capable of maintaining a high level of performance on other unseen outlet glaciers in Greenland and likely other glaciated regions such as Antarctica.

Further integration of the workflow with GIS platforms could provide an efficient tool for processing large amounts of imagery at high temporal resolution. In addition, since the workflow was implemented in Python 3.7, it is compatible with Google Earth Engine (Gorelick et al., 2017), a cloud-based geospatial platform for processing and analysing large-scale datasets (Tamiminia et al., 2020). GEE allows processing of Landsat and Sentinel-2 imagery without the need to download large volumes of data and has been used effectively in glacial applications such as automated mapping of glacial lakes (Shugar et

al., 2020). Therefore, there is scope to implement CSC within the GEE platform and build on existing tools for automated glacier margin extraction (e.g. Lea, 2018) and classification without the need for expertise in coding or glaciology. With such integration, classification outputs could be rapidly produced and used to efficiently generate vector datasets from boundaries between classes, for wide-ranging applications and analysis in outlet glacier landscapes.

## 6 Conclusion

We develop and evaluate a workflow for image classification of seasonally variable marine-terminating outlet glacier environments using deep learning. The development of deep learning methods for automated classification of outlet glaciers



is an important step towards monitoring important processes at high temporal and spatial resolution (e.g. changes in frontal position, calving events, plume development, supraglacial lake development and drainage). While still in its infancy in glacial settings, image classification using deep learning provides clear potential to reduce the labour-intensive nature of manual methods and facilitate automated analysis in an era of the burgeoning availability of satellite imagery. Our two-phase workflow, termed CNN-Supervised Classification, is adapted for classification of medium resolution Sentinel-2 imagery of outlet glaciers in south-east Greenland. In phase one, the application of a well-established, pre-trained CNN called VGG16 replicates the way a human operator would interpret an image, rapidly producing accurate training data without the requirement of time-consuming manual digitisation. In phase two, the workflow produces a pixel-level classification according to seven semantic classes characteristic of complex outlet glacier settings. Alongside an evaluation of various parameters and training methods on model performance, we apply and test the workflow on two new Sentinel-2 tiles of outlet glaciers, previously unseen by phase one CNNs during training.

Exemplified by resulting overall F1 scores of up to 95% for in-sample data and 93% for out-of-sample data, the workflow establishes a state-of-the-art in image classification for outlet glacier environments in Greenland. Additionally, when compared to traditional pixel-based techniques, the results of CSC clearly outperform those of image band ratio methods. These results demonstrate the transferability and robustness of our approach and, although the CSC workflow was applied and tested on outlet glaciers in Greenland, it may also be transferred to outlet glacier landscapes in other glaciated regions.

From a wider perspective, the results of this study strengthen the foothold of deep learning in the realm of automated processing of freely available medium resolution satellite imagery, especially building on the growing body of research using deep learning in glaciology (Baumhoer et al., 2019; Mohajerani et al., 2019; Zhang et al., 2019; Xie et al., 2020. The deep learning workflow presented here offers an efficient tool for glaciologists to analyse the dynamics of marine-terminating outlet glaciers, without significant prior experience in coding or deep learning.

**Code and Data Availability**: Sentinel-2 imagery is available from the Copernicus Open Access Hub (available at: https://scihub.copernicus.eu/dhus/#/home, last accessed: 20/07/20). The Python scripts for the full deep learning workflow and instructions on how to apply them are available at: http://doi.org/10.5281/zenodo.4081095 and can be cited as Carbonneau and Marochov (2020). The nine VGG16 models pre-trained on the Helheim area shown in Fig. 1c are available for download from this institutional repository: https://collections.durham.ac.uk/files/r2gh93gz51k#.X42cOdBKhPZ and can be cited as Marochov and Carbonneau (2020). The original code for the CSC workflow for classification of fluvial scenes is available at: https://github.com/geojames/CNN-Supervised-Classification.



**Supplement:** The supplement includes the full list of Sentinel-2 imagery used for training and testing the classification workflow (Table S1), epoch tuning graphs (Figs. S1 and S2), the Kappa values for both phase 1 results (Fig. S3) and phase 2 results (Fig. S4), and the confusion matrices for all 36 model workflows tested in this paper (Figs. S5 to S28).

**Author contributions:** PC developed the code with contributions and editing by MM. MM created training and validation data, implemented the code to perform image classifications and wrote the manuscript. CRS and PC supervised, discussed results and edited the manuscript.

**Competing Interests:** The authors declare no conflict of interest.

**Acknowledgements:** We acknowledge the European Union Copernicus program for providing Sentinel-2 data.

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
