# Peer review of "Image classification of marine-terminating outlet glaciers in Greenland using deep learning methods"

_The Cryosphere, 2020_

## Referee Comment (RC1) · Anonymous Referee #1 · 14 Dec 2020

General Comments:

This paper use a deep-learning-based workflow, termed CNN-Supervised Classification (CSC), to map glacial regions into seven classes using Sentinel-2 images. The method achieves reasonable results and shows its generalization ability. The author also shows significant improvements over traditional pixel-based methods such as band ratio techniques. There are some concerns regarding the explanation and technical details of the method, which are list below. Given this, I recommend this paper for publication after major revisions with attention to comments.

Specific comments:

I have two major concerns regarding the explanation and potential issues of the method

presented in this study.

1. The first concern I have relates to the superiority of the second phase model. The author mentioned that the second phase model (cCNN and MPL) is trained by the classification results of the phase one CNN model (Page 11, Line 310). And the author claimed that the second model outperforms the phase one CNN regarding the F1 scores. To me, the network cannot outperform the training label. For instance, Baumhoer et al. (2019) and Zhang et al. (2019) used the manual-prepared training labels to training the network, and the networks are eventually close to human-level performance but not exceed in terms of accuracy. Therefore, could the author provide more explanation of why the phase two model outperforms the phase one model?

2. The second concern relates to the method's performance on the edges of each class. Edges are important to glaciologists since that is where changes occur. The author only uses the pure tiles (Figure 5) to train the phase one model, which means the model might not have a promising performance on tiles with multiple classes (e.g., edges of the glacier or ice mélange). For phase two models, cCNN is for patch-based classification. Considering that a single patch could also contain multiple classes on edges and the phase two model is dependent on the phase one model, this method might have potential issues on the edges. It would be better if the author could quantify the method's performance on the edges or document such potential issues.

3. Page 8, Line 238: (1) How to get the variations of the surface meltwater on the glacier and ice mélange? They are not included in the seven classes. (2) It would be interesting to know how each characteristic can benefit the study of glaciology, for instance, the snow cover on bedrock.

4. Page 11, Line 311: It would be better if the author could provide more information about how to reassemble predicted classes to create a class raster. For instance, what is the stride size when predicting classes using a pre-trained CNN? How to deal with the overlap if there is any (when the stride size is smaller than the tile size)?

5. Figure 2: It would be better if the author could provide information about the median filter. What is the median filter for? Why is the median filter $1\times1$?

6. Figure 2: It would be better if the author could provide more details about vectorizing image features. For instance, how to deal with these impure patches (when the patch size is larger than 1)?

7. Page 14, Line 365: What is the stride size when using the second model to make the final classification? The stride size is important cause it influences the resolution (or size) of the final classified image.

8. Figure 8b: Could the author explain why they have unclassified regions? It seems that the edges of classes are usually unclassified (e.g., the black strip at the glacier front), which might also potentially influence the method's performance on the edges (See comment 2).

9. Page 24, Line 700: It would be better if the author could provide more a theoretical explanation about why some class confusion in phase one can be overcome in phase two (See comment 1)? Could the author provide a visual comparison between phase one and phase two classifications, like Figure 10 and Figure 11 in Carbonneau et al. (2020).

10. Page 30, Line 927: The studies based on U-Net (Baumhoer et al., 2019; Mohajerani et al., 2019; Zhang et al., 2019) focused on glacier boundaries, where this method might not generate promising results (See comments 2 and 8). Although this method could classify seven classes, I think it is not fair to conclude that this method has exceeded the U-Net based ones.

11. Page 31, Line937: The author only tests two images in summer. If the author test more images, it would be more convincing to conclude that the method could handle different illumination, weather conditions, or seasonal changes.

12. Page 34, Line 1054: I agree with the author that the combination of deep learning

methods, Google Earth Engine, and GIS software could remove the need for prior expertise in deep learning and coding (Page 33, Line 1025). But that is future work and not included in the current workflow. So I suggest removing this part from the conclusion.

Technical corrections:

Page 11, Line 311: I suppose it should be predicted classes but not image tiles that are reassembled.

Page 15, Line 396: I suppose it should be a 3D input (width, height, band).

Page 29, Line 891: Zhang et al. (2019) used TerraSAR-X images.

Page 30, Line 901: Zhang et al. (2019) and Mohajerani et al. (2019) used 2-D inputs (single-band images). Baumhoer et al. (2019) used 3-D inputs (width, height, band). The author also used 3-D input in this work (the band is just one dimension).

Reference

Baumhoer, C. A., Dietz, A. J., Kneisel, C. and Kuenzer, C.: Automated Extraction of Antarctic Glacier and Ice Shelf Fronts from Sentinel-1 Imagery Using Deep Learning, Remote Sensing, 11(21), 2529, doi:10.3390/rs11212529, 2019.

Carbonneau, P. E., Dugdale, S. J., Breckon, T. P., Dietrich, J. T., Fonstad, M. A., Miyamoto, H. and Woodget, A. S.: Adopting deep learning methods for airborne RGB fluvial scene classification, Remote Sensing of Environment, 251, 112107, doi:https://doi.org/10.1016/j.rse.2020.112107, 2020.

Mohajerani, Y., Wood, M., Velicogna, I. and Rignot, E.: Detection of Glacier Calving Margins with Convolutional Neural Networks: A Case Study, Remote Sensing, 11(1), 74, doi:10.3390/rs11010074, 2019.

Zhang, E., Liu, L. and Huang, L.: Automatically delineating the calving front of Jakobshavn Isbræ from multitemporal TerraSAR-X images: a deep learning approach, The

Cryosphere, 13(6), 1729–1741, doi:10.5194/tc-13-1729-2019, 2019.

---

## Referee Comment (RC2) · Anonymous Referee #2 · 18 Dec 2020

**General Comments**

This paper describes a two-phase deep learning approach for the image classification of Greenlandic marine-terminating outlet glaciers. Optical Sentinel-2 imagery acquired in 2019 over Helheim Glacier was used to train a VGG16 model generating training data for the multilayer perceptron/cCNN in phase two. The results were tested on two Sentinel-2 scenes over Helheim Glacier and Scoresby Sund for summer/autumn 2019. The novelty compared to previous studies is the classification of satellite images into seven different classes. Further results of this study include the performance testing on different tile sizes, transfer learning, and band combinations.

The manuscript is well written and explains the study approach in every detail. There are some concerns regarding the methodical approach and testing of the algorithm

which are explained below. Therefore, I recommend a revision of the manuscript before publication.

As outlined above, some major concerns exist regarding the following points:

- The validation labels include unclassified areas especially at the boundaries between two classes. Why was that done? What does this mean for the accuracy assessment? It seems the accuracy was only calculated over areas where a classified validation label exists. But this approach would miss out on the accuracy over regions with boundaries. Additionally, if no boundaries between classes exist in the training data I would expect that model predictions are inaccurate in those regions. Moreover, your accuracy assessment cannot account for that as validation labels include no data areas. Could you please explain how you handled class boundaries for training and validating the model?

- Testing was performed on only two images acquired temporally close to the training data. Training data was used for 07/08/2019, 01/09/2019, and 28/09/201 and tested for 13/09/2019 and 01/08/2019. This means between testing and training only one to two weeks elapsed. Hence, spectral properties of the images as well as the conditions at the glacier terminus were very similar in the training and test data and might overestimate the accuracies. To show that your approach is transferable in space and time I would recommend testing on a broader amount of images as it has been done by previous studies (Baumhoer et al., 2019; Cheng et al., 2020; Mohajerani et al., 2019; Zhang et al., 2019). I would recommend taking data from a previous/later year e.g. 2018 or 2020 not close to the training data for additional performance testing.

- What was the argument to create a two-phase deep learning model instead of using a fully convolutional network (FCN) architecture for semantic segmentation? It would be great if you could include some comments on that in your manuscript.

- Could you provide some information on the computational cost of the here presented two-phase model compared to semantic segmentation approaches?
**Specific & Technical Comments**

P2L60: For clarification, it would be great if you could give some more detail on the difference between semantic segmentation by an FCN and the pixel-based segmentation performed here. If I understood you correctly, your first CNN performs image classification, hence assigning one class to the entire image. The second cCNN performs a classification for each pixel. If patch size 1x1 is used, only the spectral properties of one-pixel are used for the classification. In the case of bigger patch sizes, also information on textural features of neighboring pixels can be used for the classification. But semantic segmentation by an FCN would also consider the spatial relationship between pixels of different classes which your approach does not.

P4L117: I think the spatial transferability is not yet proved by only testing on one outsample scene. For applications elsewhere in Greenland and Antarctica more spatially diverse training data would be required. Please mention that or show on a more diverse test image set the spatial transferability of your approach.

P9L266: How did you differentiate between snow on ice and snow on rock?

P11L311: Please describe the term "class raster" more precisely.

P12 Figure2: What does the 1x1 median filter do? Please describe.

P14L348: I would expect that the optimal hyperparameters (epochs, batch sizes, learning rate, etc.) for training are different depending on tile size. Did you experience that?

P15L396: What is the fourth dimension of your 4D tensor? Only three are listed.

P16L420: Please explain the normalization by 16384 in detail. Usually, the min/max values (normalization) or mean/standard deviation values (standardization) of the data set are used for scaling input images.

P16L429: It is not true that your dataset is larger than the previously used ones. The number of tiles might be higher as you use small single class tiles but the number of

TCD
images (13) is less than from Zhang et al. 2019 (75), Cheng et al. 2020 (20188), and Baumhoer et al. (38 scenes). You state this on P29L878.

P28L840: Be careful with comparing your F1-score directly with the one of Xie et al. (2020) and Baumhoer et al. (2019). Both studies used a more diverse set of test data. Moreover, Xie et al. calculated the accuracy also over the boundary between two classes and this is the area where errors occur. Additionally, Baumhoer et al. performed their accuracy analysis on a 1 km buffer at the calving margin to account for inaccuracies at the frontal area, where again, the inaccuracies occur. P28L849: I guess for future potential applications (e.g. glacier terminus tracking, snow line extraction, coastline mapping, etc.) especially the edges between classes are of major importance. Is it possible to get clear class boundaries from your classification result?

P29L898: You are right, that optical imagery is easier to pre-process but please also mention that SAR data has many advantages. Especially in polar regions, optical data availability is very limited due to cloud cover and polar night. SAR data overcomes those drawbacks and allows continuous time series with plenty of data.

P30L902: Be again careful with not confusing tensor size with the number of input channels.

P30L912: Maybe re-phrase or delete this sentence. Arguing by the number of bands whether a model mimics human visual performance is confusing.

P30L915: The paragraph comparing your classification approach to the U-Net architecture is slightly misleading. The U-Net allows semantic segmentation of images by delineating features. The U-Net learns shapes and forms but is not limited in variability unless the training dataset is restricted by too little data and missing augmentation. In natural landscape images, the challenge of color is often given by the fact that two classes (e.g. snow on ice and snow on rock) have similar spectral reflectance but a different texture and/or shape. That is why the U-Net is so powerful as it also considers the spatial context besides pixel values. To show that your approach exceeds the U-Net

TCD
architecture you would need to prove that it is as suitable for delineation on a larger test set. Therefore, I think it is problematic to conclude that the "compact CNN architecture has exceeded the results from the U-Net architecture". Your approach concentrates on the pixel-based classification of classes (and was tested for that) whereas the U-Net based approaches concentrated on the correct delineation between classes.

P33L1028: Again, I would be careful with class boundaries unless your approach was tested for it.

References

Baumhoer, C. A., Dietz, A. J., Kneisel, C. and Kuenzer, C.: Automated Extraction of Antarctic Glacier and Ice Shelf Fronts from Sentinel-1 Imagery Using Deep Learning, Remote Sens., 11(21), 2529, doi:10.3390/rs11212529, 2019.

Cheng, D., Hayes, W., Larour, E., Mohajerani, Y., Wood, M., Velicogna, I. and Rignot, E.: Calving Front Machine (CALFIN): Glacial Termini Dataset and Automated Deep Learning Extraction Method for Greenland, 1972–2019, Cryosphere Discuss., 2020, 1–17, doi:10.5194/tc-2020-231, 2020.

Mohajerani, Y., Wood, M., Velicogna, I. and Rignot, E.: Detection of Glacier Calving Margins with Convolutional Neural Networks: A Case Study, Remote Sens., 11(74), 1–13, doi:10.3390/rs11010074, 2019.

Xie, Z., Haritashya, U. K., Asari, V. K., Young, B. W., Bishop, M. P. and Kargel, J. S.: GlacierNet: A Deep-Learning Approach for Debris-Covered Glacier Mapping, IEEE Access, 8, 83495–83510, doi:10.1109/ACCESS.2020.2991187, 2020.

Zhang, E., Liu, L. and Huang, L.: Automatically delineating the calving front of Jakobshavn lsbræ from multitemporal TerraSAR-X images: a deep learning approach, The Cryosphere, 13(6), 1729–1741, doi:10.5194/tc-13-1729-2019, 2019.

---

## Referee Comment (RC3) · Anonymous Referee #3 · 22 Dec 2020

Marochov et al. develop a two-stage machine-learning pipeline to automatically segment glacier calving fronts into seven distinct classes. The initial phase of the pipeline uses a VGG16 convolutional architecture to label whole tiles as one of the seven classes (using a fully-connected layer at the end). Phase two uses the output of the initial labeling to perform pixel-level classification of the landscape into the seven classes. The authors explore a range of training regimes and find state-of-the-art performance for multi-class segmentation of glacier calving fronts. I believe this manuscript provides timely and suitable results for the community. However, there remain a number of major and minor issues that need to be addressed before the manuscript can be considered for publication:

**Major Comments:**

- There needs to be more justification as to why a two-stage pipeline is necessary. What happens if you directly start with a pixel-level classification of the features? It seems to me the point is that by using a pre-trained VGG16 to first classify the tiles and then using those classification as the training for the second phase, you are cutting down on the amount of required training labels to directly train on pixel-level classification. Is that true? And is this really the only reason? This needs to be communicated better.

- Statements regarding generalizability:

  – Lines 118-119: "they are also applicable to mapping outlet glaciers anywhere in the world, including Antarctica"

  – Lines 998-990: "once the phase one models are trained and weights are saved, no further training is required to apply the workflow to other marine-terminating outlet glaciers."

  – Lines 1018-1019: "our adapted CSC workflow is transferable and capable of maintaining a high level of performance on other unseen outlet glaciers in Greenland and likely other glaciated regions such as Antarctica."

  These statements need justification. You haven't shown how this pipeline performs in outside of Greenland, and delineating calving fronts in Antarctica can indeed be very different due to differences in the glacier sizes and mélange. In fact, Helheim glacier is a rather non-representative area given the shape of the calving front and the fjord. I recommend testing the pipeline in more out-of-sample areas including major fjords like Jakobshavn that are both important and have significant independent studies over time for context.

- Table 1: As mentioned in the caption, there is some class imbalance due to the geographical extent of the different features. You mention you tried to even out the imbalance in the selection as much as possible, but how is the remaining

imbalance dealt with and how is this affecting your results? You mention this in section 5.4, but the study will be significantly improved if the class imbalance is addressed. Can this be addressed in the construction of the loss function? Do you think it's not necessary or that the results won't be significantly affected to justify implementing a more nuanced loss function?

• Lines 348, 379-380: You mention the training hyper-parameters were kept constant for all nine model variations. But no justification is provided that the different models would require the same hyperparameters for optimal training. In fact, one would expect the different models to have different requirements. If this is the case, then comparing the model performances is not fair given that they have not all been optimally trained.

• Figure 4: In phase-two, instead of doing pixel-level classification for one patch at a time using an architecture with a fully-connected layer at the end, why not use a fully convolutional architecture that provides a different class for each pixel directly and convolves over the whole image, skipping the need for feeding in patches?

• Section 4.1: when reporting the performance of the pipeline with respect to manually classification, it is important to also report the uncertainty associated with manual classification at the pixel level. The exact boundaries of surfaces may differ from person to person, but this is not discussed in your performance evaluation.

**Minor Comments:**

Lines 89-92: The comparison between previous efforts and the conclusion that the Baumhoer study has the most accurate results seems unjustified. These studies are each on different geopgraphical areas and use different input data that have different resolutions. The delineation accuracy of the models in distance units (meters) is not

a fair comparison when the inputs have different resolutions. Instead, a pixel-based comparison seems more appropriate here. Also, it is a bit confusing why the statistics of the test data are reported for the first two studies, but both the test and training statistics are reported for the Baumhoer study (this confuses the reader in terms of which numbers should be compared).

Line 114: training size of 13 images. But how large are these tiles in terms of pixels? The number by itself doesn't convey any information.

Line 116: "resulting class predictions are then used as training data specific to the unseen input image". This sentence is hard to understand. I'm assuming you mean that the classification of the first stage is used as training in the second stage, even if regions where the first stage wasn't trained on. But it needs to be stated more clearly.

Line 311: "The image tiles are then reassembled to creates a class raster which is used as training data for the second model in phase two". Again this sentence is hard to understand until further on into the paper. If you explain what you mean by class raster earlier, it will be much easier to understand.

Line 325: "[. . .] typical of CNN architectures" This is not true. Not all CNN architectures have a fully connected layer. In fact that's what sets fully-convolutional networks like U-Net apart.

Line 341: "only the weights in the final layers of the NN are retrained". Which layers exactly are retrained? How far back does one have to go in the layers to adjust the pre-trained network?

Line 414-415: how does removing tiles with mixed classes (with a 95

Line 426: "new satellite image of the training area". Are you stating that the validation data used during training is never used as one of the actual training images, and therefore it is a more stringent test? But are you using the same "unseen" validation image in every epoch? This needs to be explained more clearly. If the validation loss

is being tracked for stopping the training, then it's in essence a part of your training and is different from a novel image during the testing stage that was never used in training as either training or validation data.

Line 428: Do the reported numbers of tiles include additions due to augmentation? Needs to be stated more clearly. If this is the case, then it may not be a fair comparison to the aforementioned studies in terms of comparing the volume of input data as they may not report the numbers after augmentation.

Lines 877-878: Comparing the number of training images to the previous studies here is not a fair comparison because the image sizes are different. In addition to the number of images, it is important to also mention how big (in terms of pixels and resolution) these images are. How do your 13 images compare to e.g. the 38 images of Baumhoer et al. or the 123 images of Mohajerani et al.?

Line 923: "Furthermore, the U-Net architecture will learn shapes that have a limited variability of both form and scale". This statement is not justified, and arguable false. Better justification and citation is needed for such a strong claim, especially given that U-Net has been successfully used in many contexts across many scientific fields from biomedical imaging to glaciology.

Line 955-957: "It is interesting to note that the transfer learning technique benefited from using a larger number of smaller tiles compared to the preferred smaller number of large tiles for the fully trained CNN." Is this because pre-trained networks are less versatile in learning more diverse and spatially connected features across a larger spatial domain? Maybe you can explore and explain this relationship better.

**Technical Comments:**

Liens 178-187: This is just a soft suggestion and feel free to ignore, but given the relatively long length of the manuscript and the scope of The Cryosphere, the context behind the biological inspiration of neural networks may be unnecessary here.

[Figure]

Line 192: "series of layers containing solutional, non-linearity, and pooling functions". Technically non-linearities are included as part of the convolutional layer given the activation function, whereas the pooling layer is a separate layer.

Line 199: In addition to differences in orientation, you can also mention pooling helps with translational invariance

Line 1052: Missing parenthesis at the end of list of references.

---

## Author Comment (AC1) · 12 Mar 2021

Dear Dr Bert Wouters,

We would like to thank the three referees for taking the time to review our manuscript and provide such constructive feedback, and we are pleased that they all want to see our work published. Indeed, we are delighted to have this opportunity to respond and we can confirm that we are able to address all the issues they raise.

In this response, we first provide a detailed point-by-point response to each of the three Reviewers, with their comments (*verbatim*) in blue and our response in black. As a result of their suggestions, we have undertaken some additional analysis. A summary of this analysis is found in the Appendix at the end of our response, to which we refer throughout.

We thank you for your editorial work on our manuscript and look forward to hearing from you in due course.

Mel Marochov

(on behalf of all authors)

**Reviewer 1:**

**General Comments:**

This paper use a deep-learning-based workflow, termed CNN-Supervised Classification (CSC), to map glacial regions into seven classes using Sentinel-2 images. The method achieves reasonable results and shows its generalization ability. The author also shows significant improvements over traditional pixel-based methods such as band ratio techniques. There are some concerns regarding the explanation and technical details of the method, which are list below. Given this, I recommend this paper for publication after major revisions with attention to comments. We thank the reviewer for their constructive review and encouragement.

**Specific comments:**

I have two major concerns regarding the explanation and potential issues of the method

**1**. The first concern I have relates to the superiority of the second phase model. The author mentioned that the second phase model (cCNN and MPL) is trained by the classification results of the phase one CNN model (Page 11, Line 310). And the author claimed that the second model outperforms the phase one CNN regarding the F1 scores. To me, the network

cannot outperform the training label. For instance, Baumhoer et al. (2019) and Zhang et al. (2019) used the manual-prepared training labels to training the network, and the networks are eventually close to human-level performance but not exceed in terms of accuracy. Therefore, could the author provide more explanation of why the phase two model outperforms the phase one model? This will be easy to clarify in the revised version. The key point is that these are two different models and they have different training data. We first train a VGG16 CNN on a set of tiles that have a size of 50x50, 75x75 or 100x100 pixels. In phase 1 of CSC, we apply this CNN. This will produce a single label for every tile (of 50x50, 75x75 or 100x100 pixels) in the target image. We then convert these tile labels to a full-size class raster by giving each pixel in the tile the same label value. This gives the pixelated results visible in the bottom left panels of Figures 8, 9, 10 and 15 in the Appendix. Then in phase 2 we use these CNN predictions to create a new model. Depending on the patch size parameter, this will be an MLP (patch size of 1) or a compact CNN (patch size of 3, 5, 7, or 15) (Please see Section 2) of the Appendix for updates to the cCNN). The second model will benefit from training data bespoke to the actual image it is trying to classify. This is because phase one model predictions help account for image heterogeneity and incorporate the specific illumination/weather conditions, acquisition angle and seasonal characteristics of the individual input image. In other words, one of the critical roles of the phase one CNN is to produce training data for phase two, which is locally specific to the image. Also, it has been found by Carbonneau et al. (2020) that neural networks are robust to noise. The first pixelated CNN-derived class raster will have very rough object boundaries that will straddle classes. This will generate error in the new training data. But the robustness of the MLP (which performed poorly here, but very well in Carbonneau et al., 2020) or the cCNN will overcome these errors and the phase 2 pixel-level classification will follow the edges much more closely.

2. The second concern relates to the method's performance on the edges of each class. Edges are important to glaciologists since that is where changes occur. The author only uses the pure tiles (Figure 5) to train the phase one model, which means the model might not have a promising performance on tiles with multiple classes (e.g., edges of the glacier or ice mélange). For phase two models, cCNN is for patch-based classification. Considering that a single patch could also contain multiple classes on edges and the phase two model is dependent on the phase one model, this method might have potential issues on the edges. It would be better if the author could quantify the method's performance on the edges or document such potential issues. We have extended validation areas to the edges. We have also quantified performance at the most important edge, the calving front, and included this in the final algorithm (see Appendix section 4 and the bottom right panels of Figures 8, 9, and 10). We will include this in the revised manuscript. For the benefit of the reviewers, we also

demonstrate that the valley walls do not show an edge dilation pattern (Appendix section 4 and Figure 14). We can therefore safely conclude that the edge detection performance of the method is controlled by pixel-level performance.

3. Page 8, Line 238: (1) How to get the variations of the surface meltwater on the glacier and ice mélange? They are not included in the seven classes. (2) It would be interesting to know how each characteristic can benefit the study of glaciology, for instance, the snow cover on bedrock. (1) This is a good point which is a limitation of CSC that will be clarified in the revised version. Given that the initial training requires tiles that are pure-class, CSC does not work well for classes that are only slightly bigger than the pixel dimensions. In the current version, we find that tiles of 50x50 are optimal (Appendix Section 3), but this means that we must be able to train our phase 1 CNN with tens of thousands of tiles of 50x50 that have pure classes. The VGG16 CNN we use can use smaller tiles, but the smallest is 32x32. Thus, the CSC workflow which has been designed here to produce overall landcover classes cannot directly detect these small features. That said, we note that our mélange, bedrock, snow on bedrock and glacier ice class are well classified and therefore a subsequent image processing approach, focussed on the class could be used. This could be as simple as unsupervised classification since the dark water pixels will be distinct from the ice pixels, or the darker rocks will be distinct from snow on bedrock. (2) We think there are many potential benefits of meaningful landcover classifications for glacial landscapes (such as those produced by CSC), and this will be easy to clarify in the revised manuscript.

4. Page 11, Line 311: It would be better if the author could provide more information about how to reassemble predicted classes to create a class raster. For instance, what is the stride size when predicting classes using a pre-trained CNN? How to deal with the overlap if there is any (when the stride size is smaller than the tile size)? When we apply an existing CNN to a new image for CNN-supervised classification, we use a stride of the tile size, so there is no overlap in tiles. The CNN produces labels for each tile. Then when we re-assemble the tiles into a class raster, all the pixels in the tile are given the same label value. This is why the lower left panels of Figures 8, 9, 10 and 15 in the Appendix have a pixelated appearance. Please note that this class raster has the same dimensions as the original images minus a border at the lower and right edges if the image size does not divide into the tile size. This will be explained more clearly in the revised manuscript.

**5. Figure 2**: It would be better if the author could provide information about the median filter. What is the median filter for? Why is the median filter 1×1? This filter has been removed in the revised analysis.

6. Figure 2: It would be better if the author could provide more details about vectorizing image features. For instance, how to deal with these impure patches (when the patch size is larger than 1)? We only vectorise in the case of the MLP. In the case of patches, we use a stride of 1. This means that successive patches will be overlapped, but our objective is to classify the pixel at the centre of the patch as a function of its neighbours.

**7.** Page 14, Line 365: What is the stride size when using the second model to make the final classification? The stride size is important cause it influences the resolution (or size) of the final classified image. In phase 2, the stride is 1. The phase 2 classifiers are all pixel level. For the MLP, we use all pixels, but for the cCNN we use a stride of 1. For each pixel, we extract a small tile with square dimensions of the patch size (3x3, 5x5, 7x7 or 15x15 pixels) from the image. Then we take the central pixel in the class raster and use this as the associated label. The only effective loss of resolution is at the image border where we lose (p-1)/2 pixels (p is the patch size). We will make sure this is clear in the revised manuscript.

8. Figure 8b: Could the author explain why they have unclassified regions? It seems that the edges of classes are usually unclassified (e.g., the black strip at the glacier front), which might also potentially influence the method's performance on the edges (See comment 2). During the creation of new validation data to test the workflow further, care was taken to reduce unclassified areas as much as possible (see the top right panel of Figure 8 in the Appendix for an example of our denser validation polygons). Collection of ground truth (validation) data relies on manual digitisation (i.e. drawing polygons in GIS and labelling them by class) which makes it difficult to avoid unclassified areas altogether. This will be discussed and made clear in the revised manuscript.

**9.** Page 24, Line 700: It would be better if the author could provide more a theoretical explanation about why some class confusion in phase one can be overcome in phase two (See comment 1)? Could the author provide a visual comparison between phase one and phase two classifications, like Figure 10 and Figure 11 in Carbonneau et al. (2020). The key effect is that neural networks are robust to noise in the training data (Carbonneau et al., 2020). When we re-assemble the phase 1 tiled outputs, we get a blocky class raster where each 50x50 (or 75x75 or 100x100) area has the same class. These blocks will obviously not follow the edges of the objects. However, when we use this as training data for phase 2 (MLP or cCNN), the majority of data is correct, and the final pixel-level models (especially the cCNN) can find object boundaries and even correct locations where the initial CNN was wrong. We have included Figures 8, 9, and 10 in the Appendix which show a visual comparison of phase one and phase two classifications and intend to make this clear in the revised manuscript with examples.

**10.** Page 30, Line 927: The studies based on U-Net (Baumhoer et al., 2019; Mohajerani et al., 2019; Zhang et al., 2019) focused on glacier boundaries, where this method might not generate promising results (See comments 2 and 8). Although this method could classify seven classes, I think it is not fair to conclude that this method has exceeded the U-Net based ones. See Appendix Sections 3 for re-analysed results which now include calving front detection errors, Section 4 for the performance of CSC on glacier boundaries and calving front detection, and Section 5 for a comparison of CSC and FCNs such as those used in previous work. We will address these issues in the revised version of the manuscript.

**11.** Page 31, Line937: The author only tests two images in summer. If the author test more images, it would be more convincing to conclude that the method could handle different illumination, weather conditions, or seasonal changes. Thank you for this suggestion. We have expanded the validation dataset to improve testing of spatial and temporal transferability. For details of the new validation dataset please see Section 1 of the Appendix. The full results which show how CSC handles these challenges will be detailed in the revised version and are shown in brief in Section 3 of the Appendix.

**12.** Page 34, Line 1054: I agree with the author that the combination of deep learning methods, Google Earth Engine, and GIS software could remove the need for prior expertise in deep learning and coding (Page 33, Line 1025). But that is future work and not included in the current workflow. So I suggest removing this part from the conclusion. It is not at all unusual for academic papers to include prospects for future research in the conclusion section of a paper and we would like to retain this, if possible.

**Technical corrections:**

**Page 11, Line 311**: I suppose it should be predicted classes but not image tiles that are reassembled. We will clarify this in the revised version of the manuscript.

*Page 15, Line 396: I suppose it should be a 3D input (width, height, band).* We will clarify this in the revised version. Individual training tiles are 3D [x, y, bands (channels)] but these tiles are stored as tensors in channels last format: [samples, x, y, channels].

*Page 29, Line 891*: *Zhang et al. (2019) used TerraSAR-X images.* This will be corrected in the revised version.

**Page 30, Line 901**: Zhang et al. (2019) and Mohajerani et al. (2019) used 2-D inputs (singleband images). Baumhoer et al. (2019) used 3-D inputs (width, height, band). The author also used 3-D input in this work (the band is just one dimension). We will clarify this.

**Reviewer 2:**

**General Comments:**

This paper describes a two-phase deep learning approach for the image classification of Greenlandic marine-terminating outlet glaciers. Optical Sentinel-2 imagery acquired in 2019 over Helheim Glacier was used to train a VGG16 model generating training data for the multilayer perceptron/cCNN in phase two. The results were tested on two Sentinel-2 scenes over Helheim Glacier and Scoresby Sund for summer/autumn 2019. The novelty compared to previous studies is the classification of satellite images into seven different classes. Further results of this study include the performance testing on different tile sizes, transfer learning, and band combinations. The manuscript is well written and explains the study approach in every detail. There are some concerns regarding the methodical approach and testing of the algorithm which are explained below. Therefore, I recommend a revision of the manuscript before publication. We thank the reviewer for their constructive review and are pleased to address their points below.

**As outlined above, some major concerns exist regarding the following points:**

- The validation labels include unclassified areas especially at the boundaries between two classes. Why was that done? What does this mean for the accuracy assessment? It seems the accuracy was only calculated over areas where a classified validation label exists. But this approach would miss out on the accuracy over regions with boundaries. Additionally, if no boundaries between classes exist in the training data I would expect that model predictions are inaccurate in those regions. Moreover, your accuracy assessment cannot account for that as validation labels include no data areas. Could you please explain how you handled class boundaries for training and validating the model? During the creation of new validation data to test the workflow further, care was taken to reduce unclassified areas as much as possible. Despite this, some unclassified areas were unavoidable as a result of the manual digitisation required to create the validation data. For more details on the new validation data please see Section 1 of the Appendix. Moreover, we added boundary detection analysis and calving front detection to show how CSC performs at the edges of classes. For details on this please see Section 3 and 4 of the Appendix. Figures 8 - 10 show example classification outputs for the new validation sites and include calving front detections with associated error. Figures 11-13 show the mean, median and modal error of calving front detections for our new validation data. This will be added to the revised version of the manuscript.

- Testing was performed on only two images acquired temporally close to the training data. Training data was used for 07/08/2019, 01/09/2019, and 28/09/201 and tested for 13/09/2019 and 01/08/2019. This means between testing and training only one to two weeks elapsed. Hence, spectral properties of the images as well as the conditions at the glacier terminus were very similar in the training and test data and might overestimate the accuracies. To show that your approach is transferable in space and time I would recommend testing on a broader amount of images as it has been done by previous studies (Baumhoer et al., 2019; Cheng et al., 2020; Mohajerani et al., 2019; Zhang et al., 2019). I would recommend taking data from a previous/later year e.g. 2018 or 2020 not close to the training data for additional performance testing. Please see the Appendix Section 1 for more details on our new validation dataset. In summary, the temporal range of validation imagery has been expanded to test the ability of CSC to classify seasonally variable imagery more adequately (see Figure 4). We have also included two new validation sites from central west Greenland (Jakobshavn Isbræ and Store Glacier), both of which have notably different shapes compared to the Helheim site (see Figures 2 and 3 in the Appendix) and different ice flow directions. The validation data for Jakobshavn and Store glaciers was acquired during 2020 (a year later than the training dataset).

- What was the argument to create a two-phase deep learning model instead of using a fully convolutional network (FCN) architecture for semantic segmentation? It would be great if you could include some comments on that in your manuscript. Please see Section 5 of the Appendix to find a summary comparing CSC to FCNs and our reasoning behind using a two-phase deep learning workflow. We will provide full details in the revised version of the manuscript.

- Could you provide some information on the computational cost of the here presented twophase model compared to semantic segmentation approaches? This is included in Section 5 of the Appendix which discusses FCNs. Our CSC approach has a much lower computational cost with much simpler pre-processing requirements. We will clarify this in the revised manuscript and include information about computational costs.

**Specific & Technical Comments**

**P2L60**: For clarification, it would be great if you could give some more detail on the difference between semantic segmentation by an FCN and the pixel-based segmentation performed here. If I understood you correctly, your first CNN performs image classification, hence assigning one class to the entire image. The second cCNN performs a classification for each pixel. If patch size 1x1 is used, only the spectral properties of one-pixel are used for the classification. In the case of bigger patch sizes, also information on textural features of

neighboring pixels can be used for the classification. But semantic segmentation by an FCN would also consider the spatial relationship between pixels of different classes which your approach does not. The first CNN classifies the image by assigning a class to each tile within the image (of 50x50, 75x75, or 100x100 pixels). Examples of this can be seen in Figures 8-10 and 15 in the Appendix, where the phase one CNN outputs are shown in the bottom left panels and have a pixelated appearance. We will clarify this in the revised version. We agree that the FCN has the advantage of considering inter-class relationships, but as highlighted in Section 5 of the Appendix, our 2-phase patch-based approach has other advantages in terms of processing and it produces high F1 scores.

**P4L117**: I think the spatial transferability is not yet proved by only testing on one outsample scene. For applications elsewhere in Greenland and Antarctica more spatially diverse training data would be required. Please mention that or show on a more diverse test image set the spatial transferability of your approach. Please see Section 1 of the Appendix. We have now expanded the validation data to two out-of-sample validation sites elsewhere in Greenland. We will clarify in the revised version that to classify images with different classes, such as glacial landscapes seen in Antarctica, training data should include samples of these classes, and model architectures should be modified to change/increase the number of output classes accordingly. The CSC workflow that has been trained on a single glacier in Greenland and designed to classify marine-terminating outlet glaciers in Greenland cannot be expected to correctly classify images with completely new classes.

**P9L266**: How did you differentiate between snow on ice and snow on rock? Class descriptions and example class images were shown in Table 1 of the original manuscript which will be modified and kept in the revised version. The 'Snow on Ice' class has a smooth appearance which contrasts the 'Snow on Rock' class where snow typically covers bedrock structures. The key difference in this case being the texture of the surface.

*P11L311: Please describe the term "class raster" more precisely.* A class raster is a 1-channel (1 band) raster where each pixel has a label corresponding to class. We will make sure this is clear in the revised manuscript.

**P12 Figure2**: What does the 1x1 median filter do? Please describe. This will be removed.

**P14L348**: I would expect that the optimal hyperparameters (epochs, batch sizes, learning rate, etc.) for training are different depending on tile size. Did you experience that? We did, which is why we have moved to the use of early stopping to control hyperparameters. We also use a fine tuning stage where a pre-trained CNN is trained again with a smaller number of samples (5000 per class as opposed to 30 000 in the initial stage), a lower learning rate of 10E-5 as opposed to 10E-4 in the initial stage and smaller batch sizes of just 10 (as opposed to 30). For

more details on this please see Section 2 of the Appendix. For the results of this approach see Section 3.3 of the Appendix. The full details of these methods will be explained in the revised manuscript.

**P15L396**: What is the fourth dimension of your 4D tensor? Only three are listed. We store our tensors in channels last format: [samples, X, Y, channels]. This will be clarified in the revised version of the manuscript.

**P16L420**: Please explain the normalization by 16384 in detail. Usually, the min/max values (normalization) or mean/standard deviation values (standardization) of the data set are used for scaling input images. As far as we are aware, the dominant normalisation procedure in computer vision is simply to divide by 255, which is the maximum of the RGB imagery commonly used. We have extended this to Sentinel-2 values which have a theoretical maximum of 16384. However, Sentinel-2 images are very un-saturated, so this maximum value is never reached. That said, in the new data analysis we are using a new GPU with a Turing architecture (see Appendix Section 5). This chip has so-called 'tensor cores' that are designed for tensor operations which use of 16-bit floating point data as opposed to 32-bit. Thus, we now normalise by 8192 in order to also use the integer part of the numbers and this has worked well for us. This will be explained clearly in the revised manuscript.

The method mentioned by the reviewer using unit scaling does work and would be inconvenient. Carbonneau et al. (2020) obtained very bad results when an image was subject to bespoke normalisation and concluded that CNN require that the same transform be applied to new images. Perhaps this was due to the fact that Carbonneau et al. (2020) used airborne imagery with a more variable radiometry when compared to atmospherically corrected satellite data. The scaling transform then needs to be saved and this is not as simple as just using a single factor.

**P16L429:** It is not true that your dataset is larger than the previously used ones. The number of tiles might be higher as you use small single class tiles but the number of images (13) is less than from Zhang et al. 2019 (75), Cheng et al. 2020 (20188), and Baumhoer et al. (38 scenes). You state this on P29L878. We will make our comparison to datasets in previous work clearer in the revised version. We agree with the reviewer that our initial description was confusing. If, in fact, we consider the number of satellite acquisitions used in training, our results actually required less training data and we now argue that this is an advantage of our method.

**P28L840**: Be careful with comparing your F1-score directly with the one of Xie et al. (2020) and Baumhoer et al. (2019). Both studies used a more diverse set of test data. Moreover, Xie et al. calculated the accuracy also over the boundary between two classes and this is the area

where errors occur. Additionally, Baumhoer et al. performed their accuracy analysis on a 1 km buffer at the calving margin to account for inaccuracies at the frontal area, where again, the inaccuracies occur. We will take care to compare to previous work within reason and with improved clarity. We have now improved our validation data and added calving front detection. See also Section 4 of the Appendix for more information about classification accuracy at class boundaries and Section 3 for examples of the outputs of this analysis.

**P28L849**: I guess for future potential applications (e.g. glacier terminus tracking, snow line extraction, coastline mapping, etc.) especially the edges between classes are of major importance. Is it possible to get clear class boundaries from your classification result? Yes. See Sections 3 and 4 of the Appendix.

**P29L898**: You are right, that optical imagery is easier to pre-process but please also mention that SAR data has many advantages. Especially in polar regions, optical data availability is very limited due to cloud cover and polar night. SAR data overcomes those drawbacks and allows continuous time series with plenty of data. We agree and appreciate the benefits of both optical and SAR data and will discuss both appropriately in the revised version.

**P30L902**: Be again careful with not confusing tensor size with the number of input channels. Thank you, this will be clarified in the revised version.

**P30L912**: Maybe re-phrase or delete this sentence. Arguing by the number of bands whether a model mimics human visual performance is confusing. Thank you for this suggestion, we will re-phrase/delete from the revised manuscript.

**P30L915**: The paragraph comparing your classification approach to the U-Net architecture is slightly misleading. The U-Net allows semantic segmentation of images by delineating features. The U-Net learns shapes and forms but is not limited in variability unless the training dataset is restricted by too little data and missing augmentation. In natural landscape images, the challenge of color is often given by the fact that two classes (e.g. snow on ice and snow on rock) have similar spectral reflectance but a different texture and/or shape. That is why the U-Net is so powerful as it also considers the spatial context besides pixel values. To show that your approach exceeds the U-Net architecture you would need to prove that it is as suitable for delineation on a larger test set. Therefore, I think it is problematic to conclude that the "compact CNN architecture has exceeded the results from the U-Net architecture". Your approach concentrates on the pixel-based classification of classes (and was tested for that) whereas the U-Net based approaches concentrated on the correct delineation between classes. We are going to change our discussion on FCNs in the revised version and base it on the arguments detailed in Section 5 of the Appendix.

**P33L1028**: Again, I would be careful with class boundaries unless your approach was tested for it. Again, see Section 3 and 4 of the Appendix. Our additional analysis shows how CSC performs at class boundaries and the results shown in Section 3 show the ability of CSC to detect calving fronts with good accuracy. We will of course add this to the revised version of the manuscript.

**Reviewer 3:**

**General Comments:**

Marochov et al. develop a two-stage machine-learning pipeline to automatically segment glacier calving fronts into seven distinct classes. The initial phase of the pipeline uses a VGG16 convolutional architecture to label whole tiles as one of the seven classes (using a fully-connected layer at the end). Phase two uses the output of the initial labeling to perform pixel-level classification of the landscape into the seven classes. The authors explore a range of training regimes and find state-of-the-art performance for multi-class segmentation of glacier calving fronts. I believe this manuscript provides timely and suitable results for the community. However, there remain a number of major and minor issues that need to be addressed before the manuscript can be considered for publication. Again, we thank this reviewer for taking the time to review our manuscript and their constructive comments that will help improve the revised manuscript.

**Major Comments:**

• There needs to be more justification as to why a two-stage pipeline is necessary. What happens if you directly start with a pixel-level classification of the features? It seems to me the point is that by using a pre-trained VGG16 to first classify the tiles and then using those classification as the training for the second phase, you are cutting down on the amount of required training labels to directly train on pixel-level classification. Is that true? And is this really the only reason? This needs to be communicated better. We will add more details and clarify this in the revised version of the manuscript. One of the critical roles of the phase one CNN is to produce training data for phase two which is locally specific to the input image, thus accounting for heterogeneity in individual images (which is beneficial when considering the wide-ranging seasonal differences seen in glacial landscape imagery). Please also see Section 5 of the Appendix where we discuss the advantages of CSC.

Statements regarding generalizability:

- Lines 118-119: "they are also applicable to mapping outlet glaciers anywhere in the world, including Antarctica"

- *Lines 998-990*: "once the phase one models are trained and weights are saved, no further training is required to apply the workflow to other marineterminating outlet glaciers."

– *Lines 1018-1019:* "our adapted CSC workflow is transferable and capable of maintaining a high level of performance on other unseen outlet glaciers in Greenland and likely other glaciated regions such as Antarctica."

These statements need justification. You haven't shown how this pipeline performs in outside of Greenland, and delineating calving fronts in Antarctica can indeed be very different due to differences in the glacier sizes and mélange.

In fact, Helheim glacier is a rather non-representative area given the shape of the calving front and the fjord. I recommend testing the pipeline in more out-ofsample areas including major fjords like Jakobshavn that are both important and have significant independent studies over time for context. We now have a wider validation set in terms of sites and seasons and we included Jakobshavn. See Section 1 of the Appendix for details of the increased validation dataset and Section 3 which shows some of the results for that imagery.

• **Table 1**: As mentioned in the caption, there is some class imbalance due to the geographical extent of the different features. You mention you tried to even out the imbalance in the selection as much as possible, but how is the remaining imbalance dealt with and how is this affecting your results? You mention this in section 5.4, but the study will be significantly improved if the class imbalance is addressed. Can this be addressed in the construction of the loss function? Do you think it's not necessary or that the results won't be significantly affected to justify implementing a more nuanced loss function? We balanced the training datasets for the production of the revised results, with each class now having 30 000 training samples. The details of this can be found in Section 2 of the Appendix and we will describe this in detail in the revised version of the manuscript.

• Lines 348, 379-380: You mention the training hyper-parameters were kept constant for all nine model variations. But no justification is provided that the different models would require the same hyperparameters for optimal training. In fact, one would expect the different models to have different requirements. If this is the case, then comparing the model performances is not fair given that they have not all been optimally trained. Please see Section 2 of the Appendix. We now use early stopping to adjust the training epochs of MLP and cCNN models.

Similarly, we use early stopping with a threshold callback to train the initial CNN. This will be described in full detail in the revised manuscript.

• **Figure 4**: In phase-two, instead of doing pixel-level classification for one patch at a time using an architecture with a fully-connected layer at the end, why not use a fully convolutional architecture that provides a different class for each pixel directly and convolves over the whole image, skipping the need for feeding in patches? Please refer to Section 5 of the Appendix which provides a summary of our reasoning behind using a 2-phase approach in comparison to the use of FCNs.

• Section 4.1: when reporting the performance of the pipeline with respect to manually classification, it is important to also report the uncertainty associated with manual classification at the pixel level. The exact boundaries of surfaces may differ from person to person, but this is not discussed in your performance evaluation. In our experience, unless you have a very large number of users, the results of such a test are wholly dependent on the experience your subjects have with imagery. Individual perception of details in a zoomed image can vary hugely depending on training, which renders this sort of test un-representative unless you put in the effort to do really large samples, which we've never seen in a remote sensing paper. Clearly, there is a 3-5 pixel error in manual digitising. This will be clarified in the manuscript.

**Minor Comments:**

Lines 89-92: The comparison between previous efforts and the conclusion that the Baumhoer study has the most accurate results seems unjustified. These studies are each on different geopgraphical areas and use different input data that have different resolutions. The delineation accuracy of the models in distance units (meters) is not a fair comparison when the inputs have different resolutions. Instead, a pixel-based comparison seems more appropriate here. Also, it is a bit confusing why the statistics of the test data are reported for the first two studies, but both the test and training statistics are reported for the Baumhoer study (this confuses the reader in terms of which numbers should be compared). Thank you for this suggestion, we agree that the application of models on different geographical regions is an important consideration when making comparisons. In the revised version of the manuscript, we will improve the clarity of comparisons and provide information on delineation accuracy with both pixel- and distance-based units as far as possible.

Line 114: training size of 13 images. But how large are these tiles in terms of pixels? The number by itself doesn't convey any information. This will be made clearer in the revised

version. We included the number of training tiles in Table 1 in the original manuscript and will modify this accordingly as well as providing more information about training images.

Line 116: "resulting class predictions are then used as training data specific to the unseen input image". This sentence is hard to understand. I'm assuming you mean that the classification of the first stage is used as training in the second stage, even if regions where the first stage wasn't trained on. But it needs to be stated more clearly. Yes, this will be clarified.

Line 311: "The image tiles are then reassembled to creates a class raster which is used as training data for the second model in phase two". Again this sentence is hard to understand until further on into the paper. If you explain what you mean by class raster earlier, it will be much easier to understand. We will clarify this accordingly.

Line 325: "[. . .] typical of CNN architectures" This is not true. Not all CNN architectures have a fully connected layer. In fact that's what sets fully-convolutional networks like U-Net apart. We were referring to the dense top which is a fully connected MLP and which is fairly standard. We can't recall seeing a CNN architecture where the outputs of convolution and/or max pooling go straight into the SoftMax layer.

Line 341: "only the weights in the final layers of the NN are retrained". Which layers exactly are retrained? How far back does one have to go in the layers to adjust the pre-trained network? We decided to discard transfer learning. We only needed 16-19 Sentinel-2 acquisitions and the transfer learning approach did not yield significant benefits. Given that the paper is a bit on the long side, we decided to remove all usage and reference to transfer learning.

Line 414-415: how does removing tiles with mixed classes (with a 95 It is not clear to us what the reviewer is asking here, but we can restate that, in the process of generating training samples for the VGG16 model, we only choose pure class tiles (defined as 95% of a single class). We do this because the method rests on the idea that internal spectral and textural properties can define a semantic class. As noted elsewhere in this response, this does have a disadvantage in that the raw training data must have large patches of pure class areas where a large sample of 100x100 pixel tiles can be mined. This means that this method is *not* well suited to classifying objects with spatial dimensions comparable or slightly larger than the spatial resolution of the image.

**Line 426**: "new satellite image of the training area". Are you stating that the validation data used during training is never used as one of the actual training images, and therefore it is a more stringent test? But are you using the same "unseen" validation image in every epoch?

This needs to be explained more clearly. If the validation loss is being tracked for stopping the training, then it's in essence a part of your training and is different from a novel image during the testing stage that was never used in training as either training or validation data. We are suggesting that the seen validation (not training) data is a stringent test because it is an image of the same site but acquired on a completely different date to any image used in training. We did not test CSC on any imagery that was used to train the phase one CNN. We define 'seen' or 'in-sample' validation data as an image from the same glacier site that was used to train the CNN (Helheim) but has not been used in training. We will clarify this in the revised version of the manuscript.

**Line 428**: Do the reported numbers of tiles include additions due to augmentation? Needs to be stated more clearly. If this is the case, then it may not be a fair comparison to the aforementioned studies in terms of comparing the volume of input data as they may not report the numbers after augmentation. This will be clarified, and we will make sure to compare volumes of training data with this in mind in the revised manuscript.

Lines 877-878: Comparing the number of training images to the previous studies here is not a fair comparison because the image sizes are different. In addition to the number of images, it is important to also mention how big (in terms of pixels and resolution) these images are. How do your 13 images compare to e.g. the 38 images of Baumhoer et al. or the 123 images of Mohajerani et al.? As stated above we will make sure that training data comparisons are clarified, indeed with reference to the size and resolution of training images. In fact, our results were obtained with fewer satellite acquisitions and we think this is an advantage.

Line 923: "Furthermore, the U-Net architecture will learn shapes that have a limited variability of both form and scale". This statement is not justified, and arguable false. Better justification and citation is needed for such a strong claim, especially given that U-Net has been successfully used in many contexts across many scientific fields from biomedical imaging to glaciology. See Section 5 of the Appendix. Statements in the old manuscript will be removed and the discussion on FCN will be based on the points detailed in this letter.

Line 955-957: "It is interesting to note that the transfer learning technique benefited from using a larger number of smaller tiles compared to the preferred smaller number of large tiles for the fully trained CNN." Is this because pre-trained networks are less versatile in learning more diverse and spatially connected features across a larger spatial domain? Maybe you can explore and explain this relationship better. As stated above, we felt that the transfer learning approach did not have enough impact to warrant pursuing. It's been dropped in the new analysis.

**Technical Comments:**

**Lines 178-187**: This is just a soft suggestion and feel free to ignore, but given the relatively long length of the manuscript and the scope of The Cryosphere, the context behind the biological inspiration of neural networks may be unnecessary here. We agree and intend to condense the section on CNNs.

**Line 192**: "series of layers containing solutional, non-linearity, and pooling functions". Technically non-linearities are included as part of the convolutional layer given the activation function, whereas the pooling layer is a separate layer. This is easy to correct.

*Line 199*: In addition to differences in orientation, you can also mention pooling helps with translational invariance. This is easy to correct.

Line 1052: Missing parenthesis at the end of list of references. This is easy to correct.

This will all be corrected in the revised version.

**APPENDIX: New analysis undertaken to address reviewer comments that are noted above**

**1. Re-acquired validation data with new sites and a wider seasonal range**

**1.1. New 'seen' validation site**

The 'seen' validation site has been changed to a smaller area (47.1 x 39.9 km, or 4711 x 3986 pixels) which includes Helheim Glacier, replacing the previous use of an entire Sentinel-2 tile (Figure 1). This allowed production of seasonal validation data for a larger number of images to improve testing of temporal transferability. It is also a better representation of a 'seen' study site in the traditional sense, due to its spatial extent within the phase one CNN training area.

---

## Author Response (AR1)

Dear Dr Bert Wouters,

We would like to thank the three referees for taking the time to review our manuscript and provide such constructive feedback, and we are pleased that they all want to see our work published. Indeed, we are delighted to have this opportunity to respond, and we can confirm that we have addressed all the issues they raise in the revised manuscript.

In this response, we first provide a detailed point-by-point response to each of the three Reviewers, with their comments (*verbatim*) in blue and our response in black with reference to specific sections of the revised manuscript. As a result of their suggestions, we have undertaken extensive new analysis. A summary of this analysis is found in the Appendix at the end of our response, to which we also refer throughout.

We thank you for your editorial work on our manuscript and look forward to hearing from you in due course.

Mel Marochov

(on behalf of all authors)

**Reviewer 1:**

**General Comments:**

*This paper use a deep-learning-based workflow, termed CNN-Supervised Classification (CSC), to map glacial regions into seven classes using Sentinel-2 images. The method achieves reasonable results and shows its generalization ability. The author also shows significant improvements over traditional pixel-based methods such as band ratio techniques. There are some concerns regarding the explanation and technical details of the method, which are list below. Given this, I recommend this paper for publication after major revisions with attention to comments.* We thank the reviewer for their constructive review and encouragement.

**Specific comments:**

*I have two major concerns regarding the explanation and potential issues of the method*

*1. The first concern I have relates to the superiority of the second phase model. The author mentioned that the second phase model (cCNN and MPL) is trained by the classification results of the phase one CNN model (Page 11, Line 310). And the author claimed that the*

*second model outperforms the phase one CNN regarding the F1 scores. To me, the network cannot outperform the training label. For instance, Baumhoer et al. (2019) and Zhang et al. (2019) used the manual-prepared training labels to training the network, and the networks are eventually close to human-level performance but not exceed in terms of accuracy. Therefore, could the author provide more explanation of why the phase two model outperforms the phase one model?* We have clarified this in section 2.1 of the revised manuscript and provided a new conceptual diagram of the CSC workflow (Figure 1 in revised manuscript). The key point is that these are two different models, and they have different training data. We first train a VGG16 CNN on a set of tiles that have a size of 50x50, 75x75 or 100x100 pixels. In phase 1 of CSC, we apply this CNN. This will produce a single label for every tile (of 50x50, 75x75 or 100x100 pixels) in the target image. We then convert these tile labels to a full-size class raster by giving each pixel in the tile the same label value. This gives the pixelated results visible in the bottom left panels of Figures 8, 9, 10 and 15 in the Appendix and in Figure 1 in the revised manuscript. Then in phase 2 we use these CNN predictions to create a new model. Depending on the patch size parameter, this will be an MLP (patch size of 1) or a compact CNN (patch size of 3, 5, 7, or 15) (Please see Section 2 of the Appendix for updates to the cCNN). The second model will benefit from training data bespoke to the actual image it is trying to classify. This is because phase one model predictions help account for image heterogeneity and incorporate the specific illumination/weather conditions, acquisition angle and seasonal characteristics of the individual input image. In other words, one of the critical roles of the phase one CNN is to produce training data for phase two, which is locally specific to the image. Also, it has been found by Carbonneau et al. (2020) that neural networks are robust to noise. The first pixelated CNN-derived class raster will have very rough object boundaries that will straddle classes. This will generate error in the new training data. But the robustness of the MLP (which performed poorly here, but very well in Carbonneau et al., 2020) or the cCNN will overcome these errors and the phase 2 pixel-level classification will follow the edges much more closely, thus improving the quality above that of the initial CNN tiled outcomes. This has all been explained more clearly in the revised manuscript.

***2.** The second concern relates to the method's performance on the edges of each class. Edges are important to glaciologists since that is where changes occur. The author only uses the pure tiles (Figure 5) to train the phase one model, which means the model might not have a promising performance on tiles with multiple classes (e.g., edges of the glacier or ice mélange). For phase two models, cCNN is for patch-based classification. Considering that a single patch could also contain multiple classes on edges and the phase two model is dependent on the phase one model, this method might have potential issues on the edges. It would be better if the author could quantify the method's performance on the edges or*

*document such potential issues.* We have extended validation label areas to the edges. We have also quantified performance at the most important edge, the calving front, and included this in the final algorithm (see Appendix section 4 and the bottom right panels of Figures 8, 9, and 10). We have now included these details in the revised manuscript. For the benefit of the reviewers, we also demonstrate that the valley walls do not show an edge dilation pattern (Appendix section 4 and Figure 14). We can therefore safely conclude that the edge detection performance of the method is controlled by pixel-level performance.

***3. Page 8, Line 238****: (1) How to get the variations of the surface meltwater on the glacier and ice mélange? They are not included in the seven classes. (2) It would be interesting to know how each characteristic can benefit the study of glaciology, for instance, the snow cover on bedrock.* (1) This is a good point which is a limitation of CSC that has been clarified in the revised version in section 4.1 of the discussion. Given that the initial training requires tiles that are pure-class, CSC does not work well for classes that are only slightly bigger than the pixel dimensions. In the current version, we find that tiles of 50x50 are optimal (Appendix Section 3), but this means that we must be able to train our phase 1 CNN with tens of thousands of tiles of 50x50 that have pure classes. The VGG16 CNN we use can use smaller tiles, but the smallest is 32x32. Thus, the CSC workflow which has been designed here to produce overall landcover classes cannot directly detect these small features. That said, we note that our mélange, bedrock, snow on bedrock and glacier ice class are well classified and therefore a subsequent image processing approach, focussed on the class could be used. This could be as simple as unsupervised classification since the dark water pixels will be distinct from the ice pixels. (2) We think there are many potential benefits of meaningful landcover classifications for glacial landscapes (such as those produced by CSC), which we have noted in the revised manuscript.

***4. Page 11, Line 311****: It would be better if the author could provide more information about how to reassemble predicted classes to create a class raster. For instance, what is the stride size when predicting classes using a pre-trained CNN? How to deal with the overlap if there is any (when the stride size is smaller than the tile size)?* When we apply an existing CNN to a new image for CNN-supervised classification, we use a stride of the tile size, so there is no overlap in tiles. The CNN produces labels for each tile. Then when we re-assemble the tiles into a class raster, all the pixels in the tile are given the same label value. This is why the lower left panels of Figures 8, 9, 10 and 15 in the Appendix have a pixelated appearance. Please note that this class raster has the same dimensions as the original images minus a border at the lower and right edges if the image size does not divide into the tile size. This has been explained in more detail in section 2.1 of the methods in the revised manuscript.

**5. Figure 2**: *It would be better if the author could provide information about the median filter. What is the median filter for? Why is the median filter 1×1?* This filter has been removed in the revised analysis.

**6. Figure 2**: *It would be better if the author could provide more details about vectorizing image features. For instance, how to deal with these impure patches (when the patch size is larger than 1)?* We only vectorise in the case of the MLP. In the case of patches, we use a stride of 1. This means that successive patches will be overlapped, but the patch-based cCNN is trained with the label of the pixel at the centre of the patch. We have clarified this in the manuscript and provided a new diagram of the phase two models (Figures 6 and 7) which show how the patches and pixels are used in the MLP and cCNNs.

**7. Page 14, Line 365**: *What is the stride size when using the second model to make the final classification? The stride size is important cause it influences the resolution (or size) of the final classified image.* In phase 2, the stride is 1. The phase 2 classifiers are all pixel level. For the MLP, we use all pixels, but for the cCNN we use a stride of 1. For each pixel, we extract a small tile with square dimensions of the patch size (3x3, 5x5, 7x7 or 15x15 pixels) from the image. Then we take the central pixel in the class raster and use this as the associated label. The only effective loss of data is at the image border where we lose $(p-1)/2$ pixels (p is the patch size). We have made sure this is clear in the revised manuscript in section 2.4.4.

**8. Figure 8b**: *Could the author explain why they have unclassified regions? It seems that the edges of classes are usually unclassified (e.g., the black strip at the glacier front), which might also potentially influence the method's performance on the edges (See comment 2).* During the creation of new test data to test the workflow further, care was taken to reduce unclassified areas as much as possible (see the top right panel of Figure 8 in the Appendix for an example of our denser validation polygons). Collection of ground truth (test) data relies on manual digitisation (i.e. drawing polygons in GIS and labelling them by class) which makes it difficult to avoid unclassified areas altogether. This has been discussed and made clear in the revised manuscript in section 2.5.

**9. Page 24, Line 700**: *It would be better if the author could provide more a theoretical explanation about why some class confusion in phase one can be overcome in phase two (See comment 1)? Could the author provide a visual comparison between phase one and phase two classifications, like Figure 10 and Figure 11 in Carbonneau et al. (2020).* The key effect is that neural networks are robust to noise in the training data (Carbonneau et al., 2020). When we re-assemble the phase 1 tiled outputs, we get a blocky class raster where each 50x50 (or 75x75 or 100x100) area has the same class (See Figure 1 in the revised manuscript). These blocks will obviously not follow the edges of the objects. However, when

we use this as training data for phase 2 (MLP or cCNN), the majority of data is correct, and the final pixel-level models (especially the cCNN) can find object boundaries and even correct locations where the initial CNN was wrong. We have included Figures 8, 9, and 10 in the Appendix which show a visual comparison of phase one and phase two classifications. This has been described in detail in section 2.1 of the revised manuscript and more results with both phase one and phase two outputs have been included (e.g., Figures 13, 15, 16, and 17 in the revised manuscript).

*10. Page 30, Line 927: The studies based on U-Net (Baumhoer et al., 2019; Mohajerani et al., 2019; Zhang et al., 2019) focused on glacier boundaries, where this method might not generate promising results (See comments 2 and 8). Although this method could classify seven classes, I think it is not fair to conclude that this method has exceeded the U-Net based ones.* See section 3 of the revised manuscript which show re-analysed results which now include calving front detection errors. In the appendix, Section 4 for shows performance of CSC on glacier boundaries and calving front detection, and Section 5 for a comparison of CSC and FCNs such as those used in previous work. We have addressed these issues in the revised version of the manuscript.

*11. Page 31, Line937: The author only tests two images in summer. If the author test more images, it would be more convincing to conclude that the method could handle different illumination, weather conditions, or seasonal changes.* Thank you for this suggestion. We have expanded the test dataset to improve testing of spatial and temporal transferability. For details of the new test dataset please see section 1 of the Appendix and sections 2.2.2 and 2.3 of the revised manuscript.

*12. Page 34, Line 1054: I agree with the author that the combination of deep learning methods, Google Earth Engine, and GIS software could remove the need for prior expertise in deep learning and coding (Page 33, Line 1025). But that is future work and not included in the current workflow. So I suggest removing this part from the conclusion.* It is not at all unusual for academic papers to include prospects for future research in the conclusion section of a paper and we would like to retain this, if possible.

*Technical corrections:*

*Page 11, Line 311: I suppose it should be predicted classes but not image tiles that are reassembled.* We have clarified this in the revised version of the manuscript.

*Page 15, Line 396**: I suppose it should be a 3D input (width, height, band).* We have clarified this in the revised version. Individual training tiles are 3D [x, y, bands (channels)] but these tiles are stored as tensors in channels last format: [samples, x, y, channels].

*Page 29, Line 891: Zhang et al. (2019) used TerraSAR-X images.* This has been corrected in the revised version.

*Page 30, Line 901: Zhang et al. (2019) and Mohajerani et al. (2019) used 2-D inputs (single-band images). Baumhoer et al. (2019) used 3-D inputs (width, height, band). The author also used 3-D input in this work (the band is just one dimension).* We have clarified the dimensions of our input tensors in the revised manuscript.

**Reviewer 2:**

**General Comments:**

*This paper describes a two-phase deep learning approach for the image classification of Greenlandic marine-terminating outlet glaciers. Optical Sentinel-2 imagery acquired in 2019 over Helheim Glacier was used to train a VGG16 model generating training data for the multilayer perceptron/cCNN in phase two. The results were tested on two Sentinel-2 scenes over Helheim Glacier and Scoresby Sund for summer/autumn 2019. The novelty compared to previous studies is the classification of satellite images into seven different classes. Further results of this study include the performance testing on different tile sizes, transfer learning, and band combinations. The manuscript is well written and explains the study approach in every detail. There are some concerns regarding the methodical approach and testing of the algorithm which are explained below. Therefore, I recommend a revision of the manuscript before publication.* We thank the reviewer for their constructive review and are pleased to address their points below.

*As outlined above, some major concerns exist regarding the following points:*

*- The validation labels include unclassified areas especially at the boundaries between two classes. Why was that done? What does this mean for the accuracy assessment? It seems the accuracy was only calculated over areas where a classified validation label exists. But this approach would miss out on the accuracy over regions with boundaries. Additionally, if no boundaries between classes exist in the training data I would expect that model predictions are inaccurate in those regions. Moreover, your accuracy assessment cannot account for that as validation labels include no data areas. Could you please explain how you handled class boundaries for training and validating the model?* During the creation of new test data care

was taken to reduce unclassified areas as much as possible. Despite this, some unclassified areas were unavoidable as a result of the manual digitisation required to create the validation labels for test data. We have discussed this in the revised manuscript (section 2.5). For more details on the new test data please see Section 1 of the Appendix. Moreover, we added boundary detection analysis and calving front detection to show how CSC performs at the edges of classes. For details on this please see Section 3 and 4 of the Appendix. Figures 8 - 10 show example classification outputs for the new test sites and include calving front detections with associated error. Figures 11-13 show the mean, median and modal error of calving front detections for our new test data. This has been added to the revised version of the manuscript.

*- Testing was performed on only two images acquired temporally close to the training data. Training data was used for 07/08/2019, 01/09/2019, and 28/09/201 and tested for 13/09/2019 and 01/08/2019. This means between testing and training only one to two weeks elapsed. Hence, spectral properties of the images as well as the conditions at the glacier terminus were very similar in the training and test data and might overestimate the accuracies. To show that your approach is transferable in space and time I would recommend testing on a broader amount of images as it has been done by previous studies (Baumhoer et al., 2019; Cheng et al., 2020; Mohajerani et al., 2019; Zhang et al., 2019). I would recommend taking data from a previous/later year e.g. 2018 or 2020 not close to the training data for additional performance testing.* Please see the Appendix Section 1 for more details on our new test dataset. In summary, the temporal range of test imagery has been expanded to test the ability of CSC to classify seasonally variable imagery more adequately (see Figure 4). We have also included two new test sites from central west Greenland (Jakobshavn Isbræ and Store Glacier), both of which have notably different shapes compared to the Helheim site (see Figures 2 and 3 in the Appendix) and different ice flow directions. The test data for Jakobshavn and Store glaciers was acquired during 2020 (a year later than the training dataset). We have included this in the revised version of the manuscript.

*- What was the argument to create a two-phase deep learning model instead of using a fully convolutional network (FCN) architecture for semantic segmentation? It would be great if you could include some comments on that in your manuscript.* Please see Section 5 of the Appendix to find a summary comparing CSC to FCNs and our reasoning behind using a two-phase deep learning workflow. We have provided full details in the revised version of the manuscript. Section 2.1 of the revised manuscript gives an overview of CSC which describes that CSC allows the creation of bespoke training data for each new test image.

*- Could you provide some information on the computational cost of the here presented two-phase model compared to semantic segmentation approaches?* This is included in Section 5 of the Appendix which discusses FCNs. Our CSC approach has a much lower computational cost with much simpler pre-processing requirements. We have clarified this in the revised manuscript and include information about computational costs.

*Specific & Technical Comments*

*P2L60: For clarification, it would be great if you could give some more detail on the difference between semantic segmentation by an FCN and the pixel-based segmentation performed here. If I understood you correctly, your first CNN performs image classification, hence assigning one class to the entire image. The second cCNN performs a classification for each pixel. If patch size 1x1 is used, only the spectral properties of one-pixel are used for the classification. In the case of bigger patch sizes, also information on textural features of neighboring pixels can be used for the classification. But semantic segmentation by an FCN would also consider the spatial relationship between pixels of different classes which your approach does not.* The first CNN classifies the image by assigning a class to each tile within the image (of 50x50, 75x75, or 100x100 pixels). Examples of this can be seen in Figures 8-10 and 15 in the Appendix, where the phase one CNN outputs are shown in the bottom left panels and have a pixelated appearance. We have clarified this in the revised version with a new figure (Figure 1 in manuscript). We agree that the FCN has the advantage of considering inter-class relationships, but as highlighted in Section 5 of the Appendix, our 2-phase patch-based approach has other advantages in terms of processing and it produces high F1 scores. These have been discussed in detail in Sections 4.4.1, 4.4.2, 4.4.3, and 4.4.4 in the revised manuscript.

*P4L117: I think the spatial transferability is not yet proved by only testing on one outsample scene. For applications elsewhere in Greenland and Antarctica more spatially diverse training data would be required. Please mention that or show on a more diverse test image set the spatial transferability of your approach.* Please see Section 1 of the Appendix. We have now expanded the test data to two out-of-sample sites elsewhere in Greenland. We included this in the revised version and note that to classify images with different classes, such as glacial landscapes seen in Antarctica, training data should include samples of these classes, and model architectures should be modified to change/increase the number of output classes accordingly. The CSC workflow that has been trained on a single glacier in Greenland and designed to classify marine-terminating outlet glaciers in Greenland cannot be expected to correctly classify images with completely new classes.

*P9L266: How did you differentiate between snow on ice and snow on rock?* Class descriptions and example class images were shown in Table 1 of the original manuscript which we have modified and kept in the revised version. The 'Snow on Ice' class has a smooth appearance which contrasts the 'Snow on Rock' class where snow typically covers bedrock structures. The key difference in this case being the texture of the surface. Example images of each class can now be found in Figure 2 of the revised manuscript.

*P11L311: Please describe the term "class raster" more precisely.* A class raster is a 1-channel (1 band) raster where each pixel has a label corresponding to class. We have made this clear in the revised manuscript.

*P12 Figure2: What does the 1x1 median filter do? Please describe.* This has been removed.

*P14L348: I would expect that the optimal hyperparameters (epochs, batch sizes, learning rate, etc.) for training are different depending on tile size. Did you experience that?* We did, which is why we have moved to the use of early stopping to control hyperparameters. We also use a fine tuning stage where a pre-trained CNN is trained again with a smaller number of samples (5,000 per class as opposed to 30,000 in the initial stage), a lower learning rate of 10E-5 as opposed to 10E-4 in the initial stage and smaller batch sizes of just 10 (as opposed to 30). For more details on this please see Section 2 of the Appendix. For the results of this approach see Section 3.3 of the Appendix. The full details of these methods are explained in the revised manuscript.

*P15L396: What is the fourth dimension of your 4D tensor? Only three are listed.* We store our tensors in channels last format: [samples, X, Y, channels]. This has been clarified in the revised version of the manuscript.

*P16L420: Please explain the normalization by 16384 in detail. Usually, the min/max values (normalization) or mean/standard deviation values (standardization) of the data set are used for scaling input images.* As far as we are aware, the dominant normalisation procedure in computer vision is simply to divide by 255, which is the maximum of the RGB imagery commonly used.  We have extended this to Sentinel-2 values which have a theoretical maximum of 16384. However, Sentinel-2 images are very un-saturated, so this maximum value is never reached.  That said, in the new data analysis we are using a new GPU with a Turing architecture (see Appendix Section 5). This chip has so-called 'tensor cores' that are designed for tensor operations which use of 16-bit floating point data as opposed to 32-bit. Thus, we now normalise by 8192 in order to also use the integer part of the numbers and this has worked well for us. This has been explained in the revised manuscript.

The method mentioned by the reviewer using unit scaling does work but we think it would be inconvenient when applied to new data. Carbonneau et al. (2020) obtained very bad results when an image was subject to bespoke normalisation and concluded that CNN require that the same transform be applied to new images. Perhaps this was due to the fact that Carbonneau et al. (2020) used airborne imagery with a more variable radiometry when compared to atmospherically corrected satellite data. Nevertheless, the scaling transform applied to the initial training data must be applied to all the new data and therefore the transform needs to be saved (e.g., with model persistence functions). This adds an extra step in the workflow.

*P16L429: It is not true that your dataset is larger than the previously used ones. The number of tiles might be higher as you use small single class tiles but the number of images (13) is less than from Zhang et al. 2019 (75), Cheng et al. 2020 (20188), and Baumhoer et al. (38 scenes). You state this on P29L878.* We have made our comparison to datasets in previous work clearer in the revised version. We agree with the reviewer that our initial description was confusing. If, in fact, we consider the number of satellite acquisitions used in training, our results actually required less training data and we now argue that this is an advantage of our method.

*P28L840: Be careful with comparing your F1-score directly with the one of Xie et al. (2020) and Baumhoer et al. (2019). Both studies used a more diverse set of test data. Moreover, Xie et al. calculated the accuracy also over the boundary between two classes and this is the area where errors occur. Additionally, Baumhoer et al. performed their accuracy analysis on a 1 km buffer at the calving margin to account for inaccuracies at the frontal area, where again, the inaccuracies occur.* We have taken care to compare to previous work within reason and with improved clarity. We have now improved our test data and added calving front detection. See also Section 4 of the Appendix for more information about classification accuracy at class boundaries and Section 3 for examples of the outputs of this analysis.

*P28L849: I guess for future potential applications (e.g. glacier terminus tracking, snow line extraction, coastline mapping, etc.) especially the edges between classes are of major importance. Is it possible to get clear class boundaries from your classification result?* Yes. See Sections 3 and 4 of the Appendix.

*P29L898: You are right, that optical imagery is easier to pre-process but please also mention that SAR data has many advantages. Especially in polar regions, optical data availability is very limited due to cloud cover and polar night. SAR data overcomes those drawbacks and allows continuous time series with plenty of data.* We agree and appreciate the benefits of

both optical and SAR data. We have noted in the revised manuscript that our image acquisition is limited by clouds and polar night.

*P30L902: Be again careful with not confusing tensor size with the number of input channels.* Thank you, this has been clarified in the revised version.

*P30L912: Maybe re-phrase or delete this sentence. Arguing by the number of bands whether a model mimics human visual performance is confusing.* Thank you for this suggestion, we have deleted this in the revised manuscript.

*P30L915: The paragraph comparing your classification approach to the U-Net architecture is slightly misleading. The U-Net allows semantic segmentation of images by delineating features. The U-Net learns shapes and forms but is not limited in variability unless the training dataset is restricted by too little data and missing augmentation. In natural landscape images, the challenge of color is often given by the fact that two classes (e.g. snow on ice and snow on rock) have similar spectral reflectance but a different texture and/or shape. That is why the U-Net is so powerful as it also considers the spatial context besides pixel values. To show that your approach exceeds the U-Net architecture you would need to prove that it is as suitable for delineation on a larger test set. Therefore, I think it is problematic to conclude that the "compact CNN architecture has exceeded the results from the U-Net architecture". Your approach concentrates on the pixel-based classification of classes (and was tested for that) whereas the U-Net based approaches concentrated on the correct delineation between classes.* We have changed our discussion on FCNs in the revised version and base it on the arguments detailed in Section 5 of the Appendix.

*P33L1028: Again, I would be careful with class boundaries unless your approach was tested for it.* Again, see Section 3 and 4 of the Appendix. Our additional analysis shows how CSC performs at class boundaries and the results shown in Section 3 show the ability of CSC to detect calving fronts with good accuracy. We have added this to the revised version of the manuscript which now includes calving front detection and associated error quantification.

*Reviewer 3:*

*General Comments:*

*Marochov et al. develop a two-stage machine-learning pipeline to automatically segment glacier calving fronts into seven distinct classes. The initial phase of the pipeline uses a VGG16 convolutional architecture to label whole tiles as one of the seven classes (using a*

*fully-connected layer at the end). Phase two uses the output of the initial labeling to perform pixel-level classification of the landscape into the seven classes. The authors explore a range of training regimes and find state-of-the-art performance for multi-class segmentation of glacier calving fronts. I believe this manuscript provides timely and suitable results for the community. However, there remain a number of major and minor issues that need to be addressed before the manuscript can be considered for publication.* Again, we thank this reviewer for taking the time to review our manuscript and their constructive comments that have helped improve the revised manuscript.

*Major Comments:*

• *There needs to be more justification as to why a two-stage pipeline is necessary. What happens if you directly start with a pixel-level classification of the features? It seems to me the point is that by using a pre-trained VGG16 to first classify the tiles and then using those classification as the training for the second phase, you are cutting down on the amount of required training labels to directly train on pixel-level classification. Is that true? And is this really the only reason? This needs to be communicated better.* We have clarified and explained the benefits of using a two-phase pipeline in the revised version of the manuscript (see sections 2.1 and the discussion in the revised manuscript). One of the critical roles of the phase one CNN is to produce training data for phase two which is locally specific to the input image, thus accounting for heterogeneity in individual images (which is beneficial when considering the wide-ranging seasonal differences seen in glacial landscape imagery). Please also see Section 5 of the Appendix where we discuss the advantages of CSC.

• *Statements regarding generalizability:*

– *Lines 118-119: "they are also applicable to mapping outlet glaciers anywhere in the world, including Antarctica"*

– *Lines 998-990: "once the phase one models are trained and weights are saved, no further training is required to apply the workflow to other marineterminating outlet glaciers."*

– *Lines 1018-1019: "our adapted CSC workflow is transferable and capable of maintaining a high level of performance on other unseen outlet glaciers in Greenland and likely other glaciated regions such as Antarctica."*

*These statements need justification. You haven't shown how this pipeline performs in outside of Greenland, and delineating calving fronts in Antarctica can indeed be very different due to differences in the glacier sizes and mélange.*

*In fact, Helheim glacier is a rather non-representative area given the shape of the calving front and the fjord. I recommend testing the pipeline in more out-ofsample areas including major fjords like Jakobshavn that are both important and have significant independent studies over time for context.* We now have a wider test set in terms of sites and seasons and we included Jakobshavn. See Section 1 of the Appendix for details of the increased test dataset and Section 3 which shows some of the results for that imagery. These details have been included in the revised version of the manuscript. We also note in the revised manuscript that suitable adaptations may be required to transfer the method to different glacial settings.

*• Table 1: As mentioned in the caption, there is some class imbalance due to the geographical extent of the different features. You mention you tried to even out the imbalance in the selection as much as possible, but how is the remaining imbalance dealt with and how is this affecting your results? You mention this in section 5.4, but the study will be significantly improved if the class imbalance is addressed. Can this be addressed in the construction of the loss function? Do you think it's not necessary or that the results won't be significantly affected to justify implementing a more nuanced loss function?* We balanced the training datasets for the production of the revised results, with each class now having 30,000 training samples. The details of this can be found in Section 2 of the Appendix and have been describe in detail in the revised version of the manuscript.

*• Lines 348, 379-380: You mention the training hyper-parameters were kept constant for all nine model variations. But no justification is provided that the different models would require the same hyperparameters for optimal training. In fact, one would expect the different models to have different requirements. If this is the case, then comparing the model performances is not fair given that they have not all been optimally trained.* Please see Section 2 of the Appendix. We now use early stopping to adjust the training epochs of MLP and cCNN models. Similarly, we use early stopping with a threshold callback to train the initial CNN. This has been described in full detail in the revised manuscript.

*• Figure 4: In phase-two, instead of doing pixel-level classification for one patch at a time using an architecture with a fully-connected layer at the end, why not use a fully convolutional architecture that provides a different class for each pixel directly and convolves over the whole image, skipping the need for feeding in patches?* Please refer to Section 5 of the Appendix which provides a summary of our reasoning behind using a 2-phase approach in comparison to the use of FCNs. This has been explained in detail in section 2.1 of the revised manuscript with a discussion on FCNs vs CSC in the discussion.

*• Section 4.1: when reporting the performance of the pipeline with respect to manually classification, it is important to also report the uncertainty associated with manual classification*

*at the pixel level. The exact boundaries of surfaces may differ from person to person, but this is not discussed in your performance evaluation.* In our experience, unless you have a very large number of users, the results of such a test are wholly dependent on the experience your subjects have with imagery. Individual perception of details in a zoomed image can vary hugely depending on training, which renders this sort of test un-representative unless you put in the effort to do really large samples, which we've never seen in a remote sensing paper. Clearly, there is a 3-5 pixel error in manual digitising.

*Minor Comments:*

*Lines 89-92: The comparison between previous efforts and the conclusion that the Baumhoer study has the most accurate results seems unjustified. These studies are each on different geopgraphical areas and use different input data that have different resolutions. The delineation accuracy of the models in distance units (meters) is not a fair comparison when the inputs have different resolutions. Instead, a pixel-based comparison seems more appropriate here. Also, it is a bit confusing why the statistics of the test data are reported for the first two studies, but both the test and training statistics are reported for the Baumhoer study (this confuses the reader in terms of which numbers should be compared).* Thank you for this suggestion, we agree that the application of models on different geographical regions is an important consideration when making comparisons. In the revised version of the manuscript, we have improved the clarity of comparisons and provide information on delineation accuracy with both pixel- and distance-based units as far as possible.

*Line 114: training size of 13 images. But how large are these tiles in terms of pixels? The number by itself doesn't convey any information.* This has been made clearer in the revised version (Section 2.2.1). The training area is 8871 x 3621 pixels.

*Line 116: "resulting class predictions are then used as training data specific to the unseen input image". This sentence is hard to understand. I'm assuming you mean that the classification of the first stage is used as training in the second stage, even if regions where the first stage wasn't trained on. But it needs to be stated more clearly.* Yes, this has been clarified.

*Line 311: "The image tiles are then reassembled to creates a class raster which is used as training data for the second model in phase two". Again this sentence is hard to understand until further on into the paper. If you explain what you mean by class raster earlier, it will be much easier to understand.* We have clarified this in section 2.1 of the revised manuscript.

*Line 325: "[. . .] typical of CNN architectures" This is not true. Not all CNN architectures have a fully connected layer. In fact that's what sets fully-convolutional networks like U-Net apart.* We were referring to the dense top which is a fully connected MLP and which is fairly standard. We can't recall seeing a CNN architecture where the outputs of convolution and/or max pooling go straight into the SoftMax layer.

*Line 341: "only the weights in the final layers of the NN are retrained". Which layers exactly are retrained? How far back does one have to go in the layers to adjust the pre-trained network?* We decided to discard transfer learning. We only needed 16-19 Sentinel-2 acquisitions and the transfer learning approach did not yield significant benefits. Given that the paper is a bit on the long side, we decided to remove all usage and reference to transfer learning.

*Line 414-415: how does removing tiles with mixed classes (with a 95* It is not clear to us what the reviewer is asking here, but we can restate that, in the process of generating training samples for the VGG16 model, we only choose pure class tiles (defined as 95% of a single class). We do this because the method rests on the idea that internal spectral and textural properties can define a semantic class. As noted elsewhere in this response, this does have a disadvantage in that the raw training data must have large patches of pure class areas where a large sample of 100x100 pixel tiles can be mined. This means that this method is *not* well suited to classifying objects with spatial dimensions comparable or slightly larger than the spatial resolution of the image.

*Line 426: "new satellite image of the training area". Are you stating that the validation data used during training is never used as one of the actual training images, and therefore it is a more stringent test? But are you using the same "unseen" validation image in every epoch? This needs to be explained more clearly. If the validation loss is being tracked for stopping the training, then it's in essence a part of your training and is different from a novel image during the testing stage that was never used in training as either training or validation data.* We are suggesting that the seen test (not training) data is a stringent test because it is an image of the same site but acquired on a completely different date to any image used in training. We did not test CSC on any imagery that was used to train the phase one CNN. We define 'seen' or 'in-sample' test data as an image from the same glacier site that was used to train the CNN (Helheim) but has not been used in training. We have clarified this in section 2.2.2 in the revised version of the manuscript.

*Line 428: Do the reported numbers of tiles include additions due to augmentation? Needs to be stated more clearly. If this is the case, then it may not be a fair comparison to the aforementioned studies in terms of comparing the volume of input data as they may not report*

*the numbers after augmentation.* We now report our training dataset of 30,000 samples per class following augmentation and make sure this is clear. We also now compare the number of satellite acquisitions used to train models in each study to avoid this confusion.

*Lines 877-878: Comparing the number of training images to the previous studies here is not a fair comparison because the image sizes are different. In addition to the number of images, it is important to also mention how big (in terms of pixels and resolution) these images are. How do your 13 images compare to e.g. the 38 images of Baumhoer et al. or the 123 images of Mohajerani et al.?* As stated above we have made sure that training data comparisons are clarified, indeed with reference to the size and resolution of training images. Our results were obtained with fewer satellite acquisitions and we think this is an advantage.

*Line 923: "Furthermore, the U-Net architecture will learn shapes that have a limited variability of both form and scale". This statement is not justified, and arguable false. Better justification and citation is needed for such a strong claim, especially given that U-Net has been successfully used in many contexts across many scientific fields from biomedical imaging to glaciology.* See Section 5 of the Appendix. In the revised manuscript the discussion on FCN is now based on the points detailed in this letter with reference to previous studies.

*Line 955-957: "It is interesting to note that the transfer learning technique benefited from using a larger number of smaller tiles compared to the preferred smaller number of large tiles for the fully trained CNN." Is this because pre-trained networks are less versatile in learning more diverse and spatially connected features across a larger spatial domain? Maybe you can explore and explain this relationship better.* As stated above, we felt that the transfer learning approach did not have enough impact to warrant pursuing. It's been dropped in the new analysis.

*Technical Comments:*

*Lines 178-187: This is just a soft suggestion and feel free to ignore, but given the relatively long length of the manuscript and the scope of The Cryosphere, the context behind the biological inspiration of neural networks may be unnecessary here.* We agree and have removed the section on CNNs.

*Line 192: "series of layers containing solutional, non-linearity, and pooling functions". Technically non-linearities are included as part of the convolutional layer given the activation function, whereas the pooling layer is a separate layer.* This has now been removed.

*Line 199: In addition to differences in orientation, you can also mention pooling helps with translational invariance.* This section has now been removed but thank you for the suggestion.

*Line 1052: Missing parenthesis at the end of list of references.* This has been corrected.

**APPENDIX: New analysis undertaken to address reviewer comments that are noted above**

**1. Re-acquired test data with new sites and a wider seasonal range**

**1.1.    New in-sample test site**

The in-sample test site has been changed to a smaller area (47.1 x 39.9 km, or 4711 x 3986 pixels) which includes Helheim Glacier, replacing the previous use of an entire Sentinel-2 tile (Figure 1). This allowed production of seasonal test data for a larger number of images to improve testing of temporal transferability. It is also a better representation of a 'seen' or in-sample study site in the traditional sense, due to its spatial extent within the phase one CNN

[Figure]

training area.

***Figure 1:*** *New test study sites, including (a) Helheim (acquired 18/06/2019), (b) Jakobshavn Isbræ (see Figure 2), and (c) Store glacier (see Figure 3).*

**1.2.    New out-of-sample test sites**

To test the spatial transferability of CSC more adequately, our out-of-sample test data has been expanded to include two new glacial landscapes, containing Jakobshavn Isbræ (Jakobshavn) and Store Glacier (Store) in central west (CW) Greenland (Figure 1). Both Jakobshavn and Store glaciers are major outlets of the CW Greenland Ice Sheet (GrIS) and replace the Sentinel-2 tile which previously represented the out-of-sample test site (Scoresby). Jakobshavn is the largest (by discharge volume) and fastest flowing outlet of the GrIS overall, despite periods of fluctuation (Bindschadler et al., 1989; Mouginot et al., 2019). Jakobshavn discharges 45% of the CW GrIS (Mouginot et al., 2019) and has been undergoing terminus retreat, thinning, and acceleration over the past few decades (Howat et al., 2007; Joughin et al., 2008). Comparatively, Store is responsible for 32% of discharge from the CW GrIS (Mouginot et al., 2019) but has remained relatively stable over the past several decades (Catania et al., 2018). Both glaciers have been subject to substantial independent study (e.g., Joughin et al., 2008; 2020; Cook et al., 2020).

**1.3. Characteristics of new out-of-sample test sites**

**1.3.1. Jakobshavn Isbræ**

At its current extent, Jakobshavn has notably different characteristics compared to Helheim in terms of glacier, terminus, and fjord shape (Figure 2). For example, the terminus region of Helheim is currently bound by fjord walls, whereas the retreated terminus position of Jakobshavn means it no longer occupies Ilulissat Fjord in a manner similar to Helheim. The terminus itself also has a substantially different shape composed of two distinct branches. Since the glacier and terminus have a particularly different shape to Helheim, we suggest that applying CSC to Jakobshavn imagery helps provide a more adequate test of spatial transferability.

Jakobshavn imagery also contains all the classes identified at the Helheim study area, including mélange which often makes terminus delineation challenging. Mélange has previously been observed to develop at the terminus of Jakobshavn and has been coincident with several periods of terminus advance (Joughin et al., 2008; 2020). Indeed, in the new test imagery acquired during 2020, there was a substantial area of mélange adjacent to the glacier terminus. Thus, the site provides a sufficient test area for CSC's ability to differentiate between the glacier terminus and mélange, where there is often little contrast. The Jakobshavn test site spans 35.7 x 23 km (3566 x 2265 pixels).

[Figure]

**Figure 2:** *The new out-of-sample test site: Jakobshavn, as shown by (b) in Figure 1. Showing test image acquired on 21/05/2020. Note the area of mélange occupying Ilulissat Fjord and two distinct branches of Jakobshavn terminus.*

**1.3.2. Store Glacier**

Similarly, the new Store test site has different fjord and glacier geometry to that of Helheim. In contrast to Jakobshavn, the terminus region of Store glacier extends between the walls of Ikerasak Fjord (Figure 3), and the new test imagery collected throughout 2020 shows variable fjord conditions with sporadic seasonal changes in mélange. Moreover, since both Jakobshavn and Store sites are on the west coast of Greenland, they also represent different ice flow directions compared to Helheim. The Store test site spans 27.8 x 20.9 km (or 2797 x 2089 pixels).

[Figure]

**Figure 3:** *The new out-of-sample test site: Store, located at (c) in Figure 1.*

Since the phase one CNN was trained using data solely from Helheim, we chose to only expand test sites within Greenland. Glacial landscapes elsewhere, for example in Antarctica, are often substantially different and likely contain a range of different classes beyond the seven classes identified from a single glacier in south east Greenland. We therefore suggest that to apply CSC on glacial landscapes which contain other classes, the diversity of training data should be increased, and the output classes modified to account for this. Even within Greenlandic marine-terminating glaciers, there are a range of different settings and characteristics which make classification challenging. Due to this, we have also employed and tested training techniques to improve spatial transferability, without significant manual labour (discussed in Section 2).

**1.4. Wider seasonal range**

Additionally, to further test the temporal transferability of CSC and the method's ability to handle seasonal changes in imagery, we have expanded the temporal range of test data for both the Helheim study site, and the new out-of-sample test sites (Figure 4). We aimed to acquire imagery with varying seasonal conditions, spanning the available timespan of optical Sentinel-2 imagery (i.e. obtained between February and October). In total, this resulted in 9 in-sample test images, and 18 out-of-sample test images.

[Figure]

**Figure 4:** *New imagery used to test CSC.*

**1.5. Training and test image format**

Due to the complete revision of test imagery and since the CSC workflow can handle a range of input image sizes, we removed the pre-processing step which tiles Sentinel-2 images into 3000x3000 pixel PNGs in favour of using raw GeoTIFFs as inputs. This also allows the creation of an output GeoTIFF which retains the spatial information of the input image and means output classifications can be manipulated in GIS software.

Reflecting this change, the format of phase one CNN training data was also changed from PNG to TIFF format. In brief, we start with sub-images of Helheim extracted from 13 Sentinel-2 acquisitions and we stack bands 4,3,2,8 in order to get an RGB+NIR image. Then, we run a script that will extract tiles of 100x100 using digitised training areas. This script is run with a high-overlap using a stride factor of 20. Once a tile for a given class is extracted, it is augmented by 3 successive rotations of 90 degrees and each resulting tile is saved to disk as a .tif image in 16 bit integer format. This resulted in a dataset upwards of 1 million tiles with a large imbalance that ranged from 50k tiles in class 1 to 900k tiles in class 4. So, we then drastically cut the tile population and randomly subsample 30k tiles from each class, thus resulting in a balanced training dataset. The final number, 30k per class, was chosen after trial

and error revealed that we could run all needed CNN models with all the tiles loaded in an available ram space of 64Gb with a 32Gb paging file. Finally, later in the work it was decided to implement a joint fine tuning of the models that added 5k samples per class drawn from one winter and one summer image for each of the 3 glaciers. These populations of 100x100 tiles in RGB+NIR can then be sliced as needed to create the tiles of 50x50 or 75x75 in either RGB or RGB+NIR.

The full methods of training data preparation are explained in detail in the revised version.

**2. Minor changes to network architectures and training approaches**

The revised version has 2 changes in training procedures and compact CNN (cCNN) architectures. In terms of network training, we now employ early stopping to inhibit overfitting. At the phase 1 CNN stage, we designed a custom callback that trains a network until the validation data (20% set aside with a train-test-split) reaches a desired target accuracy. These targets ranged from 92.5 to 99%. At the phase 2 stage where either an MLP or cCNN is applied, we used conventional early stopping with a patience parameter and a minimum improvement threshold. The minimum improvement was set as 0.5%. For the MLP, we found that training did not stabilise for at least 20 epochs and we set the patience to 20. This means that if training does not improve the validation accuracy by 0.05% after a period of 20 epochs, the training will stop. For the cCNN using patch sizes of 3, we used a patience of 15 and for patches of 7 and 15, a patience of 10.

Furthermore, when applying CSC to multiple sites, we came to a similar conclusion to Carbonneau et al. (2020) which found that model transferability was improved when the phase 1 CNN was trained with data from more than 1 site. We have therefore deployed a joint fine tuning training procedure where a CNN initially trained only on data from Helheim was trained with a small set of extra tiles using only two images (one from winter and one from summer) for all 3 glaciers. This fine tuning was done at a low learning rate of 10E-5 and with smaller batch sizes. This improved the final results. The rational for this is that if a glacier is identified for monitoring, the addition of 2 available scenes to produce data used to fine-tune an existing CNN is not an onerous task and can deliver significant improvements to the final results. For clarity, we will refer to CNN training without this extra level of fine-tuning as 'Single' training and CNN training with this added fine-tuning as 'Joint' training. We test the Joint training by applying it with tile sizes of 50x50 and RGB+NIR bands due to the good general performance of these parameters during Single training.

We have also slightly changed the architecture of the cCNN and now use a deepening series of convolution layers. As before, the cCNN trains to learn the class of a central pixel in a patch

as a function of a neighbourhood. But instead of using increasingly large filters that have the same size as the input image patch, we now use as many 3x3 filters as can be accommodated by the patch size without the recourse to padding.  Therefore, for 3x3 image patches, we use a single 2D convolution layer since the convolution of a 3x3 image with a 3x3 kernel returns a single scalar value. For the 5x5 image patch, we use two 2D convolution layers. The first convolution of the 5x5 image with a 3x3 kernel leaves a 3x3 image which is rendered to a scalar after a second 3x3 convolution. For the 7x7 image patch size, we use three 2D convolution layers. Finally, for the 15x15 patch size we use seven 2D convolution layers. In all cases, each convolution layer uses 32 filters and therefore passes 32 equivalent channels to the following layer, with the exception of the final layer which passes a set of 32 scalar predictors. These scalars are flattened and fed into a dense top which terminates in the usual softmax layer for class prediction.

**3.  Complete re-analysis of the data**

Given the revision of test data and the minor modifications to network architectures and training approaches, a complete re-analysis of the new classification results has been undertaken. In summary, average F1 scores reached 94% for Helheim, 97.3% for Jakobshavn, and 94.6% for Store. Details and example output classifications from these new results are shown in the following sections but are explored in greater detail in the revised version.

**3.1.    Phase 1 Tile Sizes and Bands**

From re-analysing the results based on our expanded test dataset, we see in phase one that for tile sizes of 50x50 and 75x75 pixels, the addition of the NIR band during training often substantially improved on overall F1 scores (Figure 5). For tile sizes of 100x100, the addition of the NIR band becomes less important as models trained with RGB bands produce better F1 scores than models trained on RGB+NIR tiles. We suggest a theoretical explanation for this is the trade-off between spatial and spectral data. In simple terms, larger tile sizes contain more spatial data in comparison to smaller tiles. This is explored in greater detail in the revised version.

[Figure]

*Figure 5:* The new results showing F1s from phase one, comparing all classes to the Glacier Ice class.

**3.2.    Phase 2 Pixel- vs Patch-based approaches**

The re-analysed results also show a more distinct improvement in classification performance going from pixel-based classification using an MLP to patch-based classification using a cCNN in phase two (Figure 6). This reinforces the proposal that per-pixel classification benefits from detecting the class of a pixel as a function of its neighbourhood rather than the individual pixel alone.

For the Helheim site using Single training, we see best overall results using RGB+NIR bands, with a tile size of 50x50 pixels (as in phase one: Figure 5) and patch sizes of 7x7 and 15x15. A patch size of 15 produced an average F1 score of 93.3%, while a patch size of 7 produced an average F1 of 92.9% (tile size:50, RGB+NIR bands).

Similarly, with Single training for Jakobshavn, we see best overall results using RGB+NIR bands, with a tile size of 50x50 pixels (as also seen in phase one: Figure 5). A patch size of 15 produces the highest average F1 of 95%, closely followed by a patch size of 5 with an overall F1 of 94.9%, and a patch size of 7 with an F1 of 94.5% (Figure 6).

Conversely, with Single training for Store, we see best overall results using RGB+NIR bands, and a tile size of 100x100 pixels with a patch size of 3 (F1: 91.4%) in phase 2 (Figure 6).

[Figure]

**Figure 6:** *F1 scores of final classifications after phase two of CSC in relation to tiles size, patch size and image bands.*

**3.3. Single vs Joint Phase 1 CNN Training**

Figure 7 shows a comparison of overall F1 scores between Single and Joint training for each site. To reiterate, Single training refers to the CNN trained on Helheim, and Joint training is a fine-tuned version of that CNN with the addition of a small number of training tiles from 2 site-specific images (one winter and one summer). This tunes CSC to the individual study site. As previously mentioned, we tested this Joint training approach using a tile size of 50 and RGB+NIR bands. The results show that the addition of extra training data from 2 site-specific images significantly improves F1 scores for the out-of-sample sites (Figure 7).

[Figure]

**Figure 7:** *Comparison of results using 'Single' and 'Joint' training for each site as a function of patch size and image bands.*

**3.3.1. Helheim**

For Helheim, the Joint approach increased the best average F1 score to 94% with a patch size of 15, this is only marginally (+0.7%) better than Single training (93.3%) but was largely expected since it is the in-sample site. An example of the Joint training output for the same image (a) seen in Figure 1 (acquired 18/06/2019) is shown in Figure 8.

[Figure]

*Figure 8:* *CSC results for image of Helheim (top left) acquired on 18/06/2019 ((a) in Figure 1) using Joint training. Note the increased density of validation labels (top right). The tiled class raster produced by the phase 1 CNN is shown in the bottom left which was subsequently used in training for a phase two cCNN with a patch size of 15x15 pixels. The F1 score for the final classification was 94.9% and calving front error (discussed in detail below) was an average of 50 metres (equivalent to 5 pixels).*

**3.3.2. Jakobshavn**

In comparison, for Jakobshavn the Joint approach increased the best average F1 score to 97.3% (an increase of 2.3% compared to Single training) with a patch size of 5. An example

of the classification output using Joint training is shown in Figure 9. The image used in this example classification is also shown in Figure 2.

[Figure]

*Figure 9:* CSC results for image of Jakobshavn (top left) acquired on 21/05/2020 as seen in Figure 2. Bottom left shows the phase one classification using Joint training and bottom right shows the final pixel-level phase two classification. The F1 score for the final classification was 98.3% and average calving front error was 36 m (3.6 pixels). The cCNN used in this example had a patch size of 5x5 pixels.

**3.3.3. Store**

For Store the Joint approach increased the best overall F1 score to 94.6% (an increase of 3.2% compared to Single training) when applied using cCNN with a patch size of 15. Figure 10 shows an example of the output classification for an image of Store acquired on 28/06/20 (a larger version of the image can be seen in Figure 3).

[Figure]

**Figure 10:** *CSC results using Joint training on an image of Store glacier (top left) acquired on 28/06/20 (for a larger version of input image see Figure 3). In this example, a phase two cCNN with a patch size of 7 produced the final pixel-level classification seen in the bottom right corner. The F1 score of the classification is 98.3% and the calving front error is 28.3 m (2.83 pixels). Note that in phase one predictions, some small areas of mélange were misclassified as glacier ice. But the phase two model is robust to noise, so these misclassified areas were*

*significantly reduced in phase two outputs. This explains the better F1 score in the phase two output compared to the phase one output.*

*3.4.    Summary*

Overall, these results can be summarised under the following key points:

- In reference to the influence of tile size and input bands during phase one, there is an important trade-off between spatial and spectral data. This is indicated by the fact that tile sizes of 50x50 pixels had improved performance with the addition of the NIR band, whereas tiles of 100x100 performed best without the addition of the NIR band. This relationship is explored in more detail in the revised manuscript.
- In reference to pixel- and patch- based methods in phase two, the patch-based method substantially outperforms the pixel-based method in terms of final classification quality and resulting F1 scores.
- In reference to the new training procedures tested (Single vs Joint), as might be expected, the Joint procedure shows a notable improvement over the Single training procedure, especially for the sites not used in training of the original phase one CNN. We suggest that the addition of training data from only two site-specific images for Joint training is not a substantial requirement and is worthwhile for the improvements we see in final classification outputs.

**4.    Calving front edge detection and valley margin edge dilation checks**

Reviewers 1 and 2 mentioned the issue of edges. From a theoretical perspective, the critical point is that the cCNN is a pixel-level classifier that uses an image patch to classify the central pixel in this patch. For example, when using a 5x5 image patch, the cCNN will try to classify the central pixel based on information from all 25 pixels in the patch. In training, we select the class of the central pixel of the patch as the label.  This is why the cCNN only uses odd-number patch sizes since these have a unique central pixel. This is clarified in the revised manuscript.

From a practical perspective, we have performed 2 new analyses to support our point.  First, we have implemented a calving front detection algorithm. Given that we have high quality pixel-level predictions, we were able to design a front detection procedure based on classic binary morphology operations. We start with the definition that the calving front will be the contact area between glacier ice pixels and 'ocean' pixels (open water, mélange or ice-berg water). Then we use morphologic active contours and other binary morphology operators to establish the calving front. The calving front validation consists of a single pixel-wide line

derived from manual digitisation in GIS. Full methods details are provided in the revised version and here we move directly to key results (Figures 11 to 13).

*Figure 11: Modal and median errors as a function of cCNN patch size and glacier for Single*

[Figure]

*CNN training.*

*Figure 12: Modal and median errors for the fine-tuned (Joint) model (RGB+NIR and tiles of 50).*

[Figure]

*Figure 13:* Full error distribution for the optimal parameter set of RGB+NIR, tiles of 50 and a cCNN patch size of 7. Data is for all glaciers combined.

Figure 11 and 12 show that the modal errors are very low. Crucially, these figures do not show a systematic increase of error with the patch size. In Figure 12, we see that the behaviour of the median error does not follow the same pattern for our different glaciers. Clearly if the cCNN patch caused some sort of edge bleeding effect, then calving front errors would be proportional to patch size. The variable patterns that we see in Figures 11 and 12 do not show this behaviour and instead they reflect the fact that it is pixel-level classification errors that dominate calving front errors. This calving front error evaluation has been fully included in the revised version and in fact our final algorithm assigns a new class to the calving front.

In Figure 13, we see the full distribution of errors for our optimal parameter set. Units are expressed in metres. Overall, the results are good with modal errors of 10m and medians to means ranging from 56 to 98 metres (5.6 to 9.8 pixels). We note a small tail of data where large errors can occur. First, we note that small classification errors of a few pixels (often caused by shadows at the front) can lead to errors in the 5-10 pixel range. Also, in all our test data 1 of the 24 images severely failed to detect the calving front (despite a high F1) which leads to the long tail of errors seen in Figure 13. This shows that CSC can successfully detect edges and whilst the data suggests that FCN approaches deliver slightly better performance

at these edges (error in pixels), the higher F1 scores of CSC make it a better choice for projects where the whole glacial landscape needs to be classified.

Additionally, and only for the purpose of review response, we have examined edge definition for the rock and snow on rock classes. Again, if the patch-based cCNN causes a loss of definition of edges, we would expect a dilation of objects as the patch size increases. We have therefore conducted some basic change detection on the rock classes. Using only our optimal parameter set with tiles of 50 and RGB+NIR imagery. We first isolate the rock classes into a binary image (1=rock or snow on rock and 0=other classes) and then successively difference the patch size of 3 from patch 5, patch size 3 from patch 7 and patch size 3 from patch size 15. Figure 14 presents a sample result from Helheim.

[Figure]

*Figure 14: Edge definition for rock and snow on rock classes.*

In all 3 cases, we see a complex pattern of differences (Figure 14). It is not the case that larger patch size classifications are dilated when compared to the patch size of 3. Had this been the case, Figure 14 would have been dominated by changes of –1 indicating that the patch size of 3 has a value of 0 where the map for the larger patch size (5, 7, 15) has a value of 1.

**5. Use of CSC vs FCN.**

All three reviewers queried the need to use a method that does not use fully convolutional networks (FCN) and we will therefore address the issue here. Hoeser et al. (2020) reviewed object detection in remote sensing and whilst they do conclude that FCN/U-net architectures are dominate, they still find about 30% of published work uses patch-based approaches which are akin to the second phase of the CSC method presented here. The advantage of CSC over the 1-stage patch-based method is that the initial phase 1 CNN provides transferability and delivers a bespoke training set for the pixel-level patch-based operator.

We also argue that CSC has certain practical advantages over U-nets in terms of data processing and computational loads. Our CSC method has low pre-processing requirements. In the revised version, we have removed the initial median filter. The test images were cropped to areas of ca. 2000-3000 by 2000-3000 pixels in order to have areas that are large yet still workable for detailed digitisation for testing. Then, the only pre-processing step we need is a normalisation by a constant factor. For the new revised version, we have used a GPU with a Turing architecture and have enabled the TensorFlow mixed precision training method that uses float16 data at input. We therefore decided to normalise our data by a constant 8192. This will lead to results that range from [0:2[ thus making use of the integer part of the float16 data. Once this is done, CSC has a low computational load. Training the initial VGG16 model can be done in under 1 hour using an I7 processor at 5.1Ghz, and an Nvidia RTX 2060 GPU. Then when we apply CSC to a sample image of 3000x3000 using optimal parameters of RGB+NIR, tile sizes of 50 for the phase 1 CNN and a patch size of 7, classification requires 4 minutes. We have also coded a low-memory usage pathway in the main script that classifies a large image row-by-row with a threshold to define 'large' set by the user. Using this, we can classify a stack consisting of full bands 4, 3, 2, and 8 for Sentinel 2 at native resolution (10960x10960 pixels each) in 12 minutes with a peak ram consumption of 11Gb. This makes CSC suitable for use in free cloud-based solutions such as Google Colaboratory.

In contrast, FCN architectures can be very demanding in terms of computer RAM and GPU RAM, especially when large images are used as inputs. We implemented the popular FCN8 based on VGG16 which has ca 130 million trainable parameters. We found that the largest dyadic image size that we could process was 512x512. This general problem has been resolved in different ways in the EO-facing literature. Baumhoer et al. 2019 used 90m SAR data as their base and using a smaller FCN with ca 7.8 million parameters, they used image tiles of 780x780 with 4 channels on a GTX 1080 GPU (8Gb vs 6Gb for the RTX2060). However, it is important to note that with 90m data, 780 pixels still covers 70km. If this were Sentinel 2 optical data, with a resolution of 10m, the sample tiles would only cover 7.8 km. In

contrast, the front of Jakobshavn used in this work has a width of ca 11km. In order to get around this sort of issue, downsampling is used. For example, Mohajerani et al. 2019 have an advanced pre-processing routine that involves a re-orientation and then a resampling of the scene to 200x300 pixels. In the end, the FCN they use only has 240x152 pixels in a single post-processed channel.

Here we note that our CSC approach requires fewer Sentinel-2 acquisitions for training. There is discussion in the reviews about the actual volume of input data and we agree comparisons can be confusing. Given that our basic phase 1 CNN training sample is no larger than 100x100 pixels, a very large number of samples can be extracted from a full Sentinel 2 tile with 10980x10980 pixels. In our initial training of the VGG16 model, we used sub-images of ca 3000x3000 pixels extracted from 13 Sentinel-2 acquisitions. In the joint-fine tuning step, we added data from 6 Sentinel-2 acquisitions (1 winter + 1 summer for each of the 3 glaciers). So, in total, this work uses data from 13-19 Sentinel-2 acquisitions. So, in fact, we argue that our results were obtained with less training data than those from comparator FCN-facing works.

This also illustrates an area where we argue CSC has advantages: In FCN architectures, the instance that must be classified must be well framed in the image. And often in the case of higher resolution images where such framing would lead to image sizes in excess of 1000x1000, downsampling must be used unless extremely powerfully GPU are available. Another important point is that the pre-processing methods used in FCN papers start with a user actually knowing where the feature of interest is and performing a suitable clip of the data. These things are not required in CSC. Our method can process entire tiles of Sentinel-2 data at native resolutions without the need for downsampling, selection and clipping of a known target area, or extensive pre-processing (Figure 15). In the paper, we have manually clipped study sites in order to produce digitised validation labels (digitisation of entire Sentinel-2 tiles to near pixel-levels of detail was beyond the timescale for this revision), but the CSC method is not sensitive to where the data clip boundaries fall, and it performs well even when an image boundary cuts a glacier in half. It also works well when the user does not have previous knowledge of the location of a feature of interest. Admittedly, in the case of glaciers, this is arguably not important because we already have high quality glacier inventories. But if we think of the wider scope of image classification in Earth Observation, there are many cases where a human user cannot be expected to know *a priori* the location of all features/class instances of interest in order to carry out the level of pre-processing required by FCN architectures. In these cases, the lower levels of pre-processing required by CSC are advantageous.

[Figure]

**Figure 15:** *Whole tile classification for a Sentinel 2 image of the Helheim region acquired on 13/09/2019.*

From a theoretical perspective, the idea that we unsuccessfully tried to convey in the initial manuscript is that FCN architectures can be strongly dependent on object shapes and less dependent on inner textures. In the final stages of the encoder part of the FCN architecture,

the simplified shape of the object will contribute to the weights learned in training (as will inter class relations). This means that an FCN must be trained to recognise specific shapes. A driverless car FCN only trained to recognise walking humans would still hit cyclists. An FCN trained only on data from Helheim could not be expected to perform well at the task of classifying Jakobshavn. We cannot find a published example where an FCN has been trained on a single site and displays transferability to very different glaciers. Rather what we see is that multiple sites must be included in FCN training in order to reach good transferability (e.g., Cheng et al., 2021 [pre-print]). However, in our results, even before the application of joint fine tuning, our phase 1 VGG16 CNN solely trained on data from Helheim successfully classified large areas of Jakobshavn leading to very high performance with final, phase 2 results in excess of 95% F1. This is because our method is driven by spectral and textural properties within the object whilst the downsampling often required in an FCN pipeline will remove local textures. FCN compensate this by making use of inter-class relations and we agree with the review comment that our method does not consider these inter-class relations. However, we counter this by arguing that on the terrestrial surface, there is a strong correlation between the ontology of a semantic class and it's colour and textural properties. This explains why a statistical learning algorithm such as maximum likelihood has been used with reasonable success by the EO community for nearly half a century. Furthermore, the learning of shapes, a strong point of FCN, is not so relevant in EO since many semantic classes have either variable shapes or no shapes at all. Good examples are forest and vegetated patches, water body shapes (including supraglacial lakes), rocky outcrop shapes, and sediment patches in rivers, etc.

Finally, we point to our empirical results. In the revised data, we have greatly extended the surface area of the test data making it reach to the edges of features. Nevertheless, F1 scores remain above 90% and often above 95% for all classes. They can be as high as 98% for the glacier ice class. CSC has delivered a state-of-the-art performance. Overall, when compared to FCN, we see lower training data volume requirements, simpler pre-processing, marginally better F1 scores and marginally poorer calving front detections. On balance, we think this shows that there is still a place in Earth Observation for patch-based classification methods.

**References**

Bindschadler, R. *Surface Topography of the Greenland Ice Sheet from Satellite Radar Altimetry* (National Aeronautics and Space Administration, Office of Management, Scientific and Technical Information Division, 1989).

Carbonneau, P. E., Dugdale, S. J., Breckon, T. P., Dietrich, J. T., Fonstad, M. A., Miyamoto, H. and Woodget, A. S.: Adopting deep learning methods for airborne RGB fluvial scene classification, *Remote Sensing of Environment*, 251, 112107, doi:https://doi.org/10.1016/j.rse.2020.112107, 2020.

Catania, G. A., Stearns, L. A., Sutherland, D. A., Fried, M. J., Bartholomaus, T. C., Morlighem, M., Shroyer, E., and Nash, J.: Geometric Controls on Tidewater Glacier Retreat in Central Western Greenland, *J. Geophys. Res.-Earth*, 123, 2024–2038, https://doi.org/10.1029/2017JF004499, 2018.

Cheng, D., Hayes, W., Larour, E., Mohajerani, Y., Wood, M., Velicogna, I., and Rignot, E.: Calving Front Machine (CALFIN): Glacial Termini Dataset and Automated Deep Learning Extraction Method for Greenland, 1972–2019, *The Cryosphere Discuss.* [preprint], https://doi.org/10.5194/tc-2020-231, in review, 2020.

Cook, S. J., Christoffersen, P., Todd, J., Slater, D., and Chauché, N.: Coupled modelling of subglacial hydrology and calving-front melting at Store Glacier, West Greenland , *The Cryosphere*, 14, 905–924, https://doi.org/10.5194/tc-14-905-2020, 2020.

Hoeser, T., Bachofer, F., Kuenzer, C.: Object Detection and Image Segmentation with Deep Learning on Earth Observation Data: A Review—Part II: Applications. *Remote Sens.* 12, 3053. https://doi.org/10.3390/rs12183053, 2020.

Howat, I., Joughin, I. & Scambos, T. Rapid changes in ice discharge from Greenland outlet glaciers. *Science* **315**, 1559–1561, 2007.

Joughin, I., Howat, I. M., Fahnestock, M., Smith, B., Krabill, W., Alley, R. B., Stern, H., and Truffer, M.: Continued evolution of Jakobshavn Isbrae following its rapid speedup, *J. Geophys. Res.-Earth*, 113, F04006, https://doi.org/10.1029/2008JF001023, 2008.

Joughin, I., Shean, D. E., Smith, B. E., and Floricioiu, D.: A decade of variability on Jakobshavn Isbræ: ocean temperatures pace speed through influence on mélange rigidity, *The Cryosphere*, 14, 211–227, https://doi.org/10.5194/tc-14-211-2020, 2020.

Mouginot, J., Rignot, E., Bjork, A. A., van den Broeke, M., Millan, R., Morlighem, M., Noël, B., Scheuchl, B., and Wood, M.: Forty-six years of Greenland Ice Sheet mass balance from 1972 to 2018, *P. Natl. Acad. Sci.* 116, 9239–9244, https://doi.org/10.1073/pnas.1904242116, 2019.

---

## Referee Report (RR1)

**Review on „Image Classification of Marine-Terminating Outlet Glaciers using Deep Learning Methods" by Marochov et al.**

Marochov et al. made a huge effort to revise their manuscript and to address the reviewer's comments in every detail. The manuscript improved particularly with regard to more accurate explanations on technical details, additional test data and the implementation of an approach to extract the calving front. Additionally, the edge classification problem was addressed accurately and is clear now. Nevertheless, a few major concerns remain which are outlined below.

1. The authors added two additional test sites and more test scenes covering a wider temporal variety. This really highlights the transferability of the developed approach. Could you please explain why you decided to remove the initial test site Scoresby Sund mentioned in the first version of the manuscript? As you already have the data it would be worth to include it as an additional test set.

2. The revised manuscript includes a lot of additional information on the developed approach and describes technical aspects in every detail. On the one hand, this is an advantage of the manuscript as the approach is very transparent. On the other hand, the manuscript has become rather long and focuses more on technical details. The authors have to be careful to not only provide methodical details on their approach but also to fit within the scope of The Cryosphere by addressing a broader cryospheric community.

    To make your manuscript more suitable for a wider cryospheric community I would recommend the following:

    a. Highlight the advantages of your approach for the cryospheric community. So far, the discussion is solely technical. But you could add one section discussing the future advantages of your approach for the analysis of the cryosphere (e.g. calving front detection, change detection of class distributions, snow cover changes between different years etc.)

    b. Highlight the great performance of your classification algorithm but try not to confuse the reader with too many details about performance differences. For example, consider to shift some of the plots to the supplementary material. You could just show the data for the optimal model configuration in the manuscript and keep the rest in the supplementary materials.

    c. Consider to shorten the text and densify the information on tile sizes, patches and different model configurations. For readers with no background in machine learning all those parameters might be confusing. Probably, a table including all those different parameters and the corresponding accuracies could help for a condensed and better overview.

    d. The authors put a lot of effort into comparing different model configurations (tile and patch size) which is highlighted in several figures and graphs. In my opinion, it would be worth to merge some of the figures to shorten the manuscript. Please see the suggestions in the technical corrections below.

**Technical corrections:**

Figure 1: This is a really nice figure and helps to understand your classification approach much better. Well done!

Figure 2: Nice idea to combine the class examples within this figure. Looks much nicer now.

Figure 6 & 7: Consider to merge those two figures into Figure 6a & 6b. It will be easier to see the differences between the cCNN and MPL approach.

Figures 11, 12, 13, 15, 16 & 17 demonstrate the classification results. The amount of figures might be a bit too heavy compared to the length of the paper. You could consider to merge the results into less figures or shift some results to the supplementary materials.

Figure 14: You could condense the information and use one plot showing all three glaciers in the same plot with the best model configuration (RGB+NIR, Single). The remaining part could be moved to the supplementary.

Figure 15d: It is interesting, that the model confuses mélange with glacier ice so heavily. Could you explain why this is the case?

L371: "trains to learn". Please re-phrase.

L486: What does "bergy" mean? Or just a typo.

L591: Why does the joint model provide higher classification accuracies but the single model higher accuracies for the calving front extraction? This seems to be contradictory.

L644: Here you mention that the developed approach might be suitable for lake mapping but earlier it is mentioned that the approach has difficulties with classes being smaller than the tile size. Are those lakes always large enough to be captured by your model? Additionally, the class "lake" is not included in the classification or is it defined as open water enclosed by glacier ice?

Supplement:

1. Why did you use additional Helheim scenes for the joint training method even though the single model was trained on Helheim anyways? (see scenes with * in Table S1)
2. Why do you provide only a confusion matrix for the single training but not the combined training approach? It would be interesting to see the performance differences to justify the necessity of a joint and single model approach. Moreover, I would assume that the most robust (spatially transferrable) model would be achieved by including several training areas from the beginning (instead of only Helheim) over different glaciers which would make the additional joint training unnecessary.

Please don't be discouraged by the length of my review. I know that a lot of work went into the revised version and the provided comments might require some further effort. Nevertheless, my comments are mostly suggestions and not a must hopefully helping to improve your manuscript.

---

## Referee Report (RR2)

**Comments:**

Most of the questions have been addressed carefully, and I appreciate the well-documented answers prepared by the author. Given this, I recommend this paper for publication.

However, I still have one concern regarding the response to my comment 9 about how the Phase 2 model can overcome the inaccurate boundaries in the Phase 1 model. I believe from Phase 1 to Phase 2, there is a huge improvement (Figure 13, 15, 16, and 17), and it deserves more detailed explanations. That is why I was looking for a theoretical explanation in my previous comment. However, the author only claims it is due to the robustness of the Phase 2 model. Again, it would be beneficial for me and other readers to know the mechanism behind the robustness.

My initial guess is that the robustness of the Phase 2 network is owing to the early stopping. The author mentioned that the training data are not all correct. Maybe the early stopping could prevent the Phase 2 network from being overfitted to the incorrect training information.

Also, I am curious about how much percent of the training data is incorrect. I suppose that the F1 scores for Phase 1 CNN tile classification are tile-based ones. If I am correct, what the pixel-based F1 score would be after converting the tile labels to a full-size class raster (e.g., Figure 16c & Figure 17c)?

---

## Author Response (AR2)

Dear Dr Bert Wouters,

Again, we would like to thank the three referees for taking further time to review our manuscript and provide valuable feedback which has helped to improve the paper. We are pleased to have this opportunity to respond, and we can confirm that all the issues they raise have been addressed.

In this response, we provide a detailed point-by-point response to each of the three Reviewers, with their comments (*verbatim*) in blue and our response in black with reference to specific lines/sections of the revised manuscript. As a result of their suggestions, we have undertaken further corrections, reduced the length of the paper and elaborated on potential applications of the method for the wider glaciological community. We shortened the manuscript by 3,271 words overall and length was reduced most by condensing the introduction for improve readability, focusing less on different parameter combinations (i.e., tile size, patch size, image bands), and by merging/moving figures to the supplement.

We thank you for your editorial work on our manuscript and look forward to hearing from you in due course.

Melanie Marochov

(on behalf of all authors)

**Reviewer 1**

*Comments:*

*Most of the questions have been addressed carefully, and I appreciate the well-documented answers prepared by the author. Given this, I recommend this paper for publication.* We thank the reviewer for their feedback and are delighted they would like to see our manuscript published.

*However, I still have one concern regarding the response to my comment 9 about how the Phase 2 model can overcome the inaccurate boundaries in the Phase 1 model. I believe from Phase 1 to Phase 2, there is a huge improvement (Figure 13, 15, 16, and 17), and it deserves more detailed explanations. That is why I was looking for a theoretical explanation in my previous comment. However, the author only claims it is due to the robustness of the Phase 2 model. Again, it would be beneficial for me and other readers to know the mechanism behind the robustness.*

*My initial guess is that the robustness of the Phase 2 network is owing to the early stopping. The author mentioned that the training data are not all correct. Maybe the early stopping could prevent the Phase 2 network from being overfitted to the incorrect training information.* In the first revision of the manuscript, we stated that the phase two models are robust to noise, and we try to further clarify the theory behind this in lines 115-121 of the revised manuscript. In general, it has been found that deep learning models can generalise to data well even if some training labels are incorrect due to the overall training process. This is because deep learning models are designed to minimise error and not overfit to training data (i.e., they are able to learn the overall trend in training data even if some labels are wrong). Thus, the phase two models can overcome some level of error in phase one predictions because *most* of the phase one predicted training labels are correct; they do not memorise the labels which are wrong but generalise to the overall trend. We have tried to clarify this in the revised manuscript.

*Also, I am curious about how much percent of the training data is incorrect. I suppose that the F1 scores for Phase 1 CNN tile classification are tile-based ones. If I am correct, what the pixel-based F1 score would be after converting the tile labels to a full-size class raster (e.g., Figure 16c & Figure 17c)?* F1 scores for both phase one and phase two are pixel-based and estimated from a sample of 10 million pixels which we state on line 371. Thus, phase one F1 scores give a representation of how correct phase two training labels are.

**Reviewer 2**

*Marochov et al. made a huge effort to revise their manuscript and to address the reviewer's comments in every detail. The manuscript improved particularly with regard to more accurate explanations on technical details, additional test data and the implementation of an approach to extract the calving front. Additionally, the edge classification problem was addressed accurately and is clear now. Nevertheless, a few major concerns remain which are outlined below.* We thank the reviewer for their constructive feedback which has helped to improve the revised manuscript.

*1. The authors added two additional test sites and more test scenes covering a wider temporal variety. This really highlights the transferability of the developed approach. Could you please explain why you decided to remove the initial test site Scoresby Sund mentioned in the first version of the manuscript? As you already have the data it would be worth to include it as an additional test set.* We decided to remove the Scoresby Sund site because although we had the original test data, the data for the revised test sites had to be completely re-digitised since reviewers rightly suggested that too much blank space existed in the original validation labels. As such, including the Scoresby site would also have prompted a complete re-digitisation of

the validation data for an increased number of acquisitions to provide a seasonal test of transferability as was done for the Helheim site and new out-of-sample sites. While it would have been ideal to include more sites, we were limited by the time it takes to manually delineate validation labels within the constraints of review deadlines (especially since the original Scoresby site spanned an entire Sentinel-2 tile).

*2. The revised manuscript includes a lot of additional information on the developed approach and describes technical aspects in every detail. On the one hand, this is an advantage of the manuscript as the approach is very transparent. On the other hand, the manuscript has become rather long and focuses more on technical details. The authors have to be careful to not only provide methodical details on their approach but also to fit within the scope of The Cryosphere by addressing a broader cryospheric community. To make your manuscript more suitable for a wider cryospheric community I would recommend the following:*

*a. Highlight the advantages of your approach for the cryospheric community. So far, the discussion is solely technical. But you could add one section discussing the future advantages of your approach for the analysis of the cryosphere (e.g. calving front detection, change detection of class distributions, snow cover changes between different years etc.)* The first revision of the manuscript had a paragraph explaining the further potential applications of CSC within the cryosphere which we have tried to strengthen in the second revision (Section **4.2 CSC performance and wider application**). We have also added a small case study (Figure 10) as requested by the editor, which demonstrates that CSC outputs can be used to create time series data, in this case including calving front change and mélange area variation throughout the test imagery collected for the Helheim site during 2019.

*b. Highlight the great performance of your classification algorithm but try not to confuse the reader with too many details about performance differences. For example, consider to shift some of the plots to the supplementary material. You could just show the data for the optimal model configuration in the manuscript and keep the rest in the supplementary materials.* We have followed your suggestions by condensing the amount of technical detail in relation to parameter testing and moving it to a small section titled **2.6 Optimal performance parameters**. Similarly, some plots have been removed/moved to the supplement and figures of optimal CSC outputs are shown in the results alongside the new case study.

*c. Consider to shorten the text and densify the information on tile sizes, patches and different model configurations. For readers with no background in machine learning all those parameters might be confusing. Probably, a table including all those different*

*parameters and the corresponding accuracies could help for a condensed and better overview.* We have followed your suggestions as noted above and created the table you suggest (Table 1). Information on tile sizes, patch sizes and image bands are no longer in the results section but have been condensed into section 2.6.

*d. The authors put a lot of effort into comparing different model configurations (tile and patch size) which is highlighted in several figures and graphs. In my opinion, it would be worth to merge some of the figures to shorten the manuscript. Please see the suggestions in the technical corrections below.* Again, we have reduced the number of figures and merged some as suggested. The paper is now shorter, and we have tried to shift the focus from different model configurations to a more balanced overview of both the advantages of the approach in terms of glaciological applications and the technical considerations for future research in this field.

*Technical corrections:*

*Figure 1: This is a really nice figure and helps to understand your classification approach much better. Well done!* Thank you!

*Figure 2: Nice idea to combine the class examples within this figure. Looks much nicer now.* We appreciate the positive feedback.

*Figure 6 & 7: Consider to merge those two figures into Figure 6a & 6b. It will be easier to see the differences between the cCNN and MPL approach.* Thank you for this suggestion, the figures have now been merged.

*Figures 11, 12, 13, 15, 16 & 17 demonstrate the classification results. The amount of figures might be a bit too heavy compared to the length of the paper. You could consider to merge the results into less figures or shift some results to the supplementary materials.* As noted above, the number of figures has been reduced and some have been shifted to the supplement. In the revised version of the manuscript Figures 8 and 9 now show optimal classification/calving front detection examples of CSC applied to seasonal imagery and Figure 10 represents a small case study which highlights the usefulness of our multi-class approach.

*Figure 14: You could condense the information and use one plot showing all three glaciers in the same plot with the best model configuration (RGB+NIR, Single). The remaining part could be moved to the supplementary.* We have removed this plot completely as the information is now shown in Tables 1 and 2.

*Figure 15d: It is interesting, that the model confuses mélange with glacier ice so heavily. Could you explain why this is the case?* We expect this is to do with the spectral similarity between

the classes as well as textural properties. In the Helheim training area, most of the glacier ice is highly crevassed, whereas the section that was misclassified as mélange in the Store test image (originally Figure15d, now Figure S3d) appears to be less crevassed. I.e., since the CNN had not seen many samples of less crevassed glacier ice, it was less likely to predict correctly in this case. As a result, including additional training data from the Store site improved the model's ability to predict the class correctly (Joint training). We have also suggested in the revised manuscript that including more diverse training data from more than one glacier in future work would likely aid classification.

*L371: "trains to learn". Please re-phrase.* This has been corrected.

*L486: What does "bergy" mean? Or just a typo.* This was a typo and has been corrected.

*L591: Why does the joint model provide higher classification accuracies but the single model higher accuracies for the calving front extraction? This seems to be contradictory.* F1 scores show accuracy for the entire test image so while small classification differences at the calving front may not significantly affect F1s for the image classification as a whole, differences of just a few pixels (e.g., in areas of shadow) can impact calving front detection more acutely. Thus, small changes in the way the Single and Joint models predict pixels at the calving front can lead to some variation in error between the two approaches even though the additional training data for the Joint approach improved overall classification performance.

*L644: Here you mention that the developed approach might be suitable for lake mapping but earlier it is mentioned that the approach has difficulties with classes being smaller than the tile size. Are those lakes always large enough to be captured by your model? Additionally, the class "lake" is not included in the classification or is it defined as open water enclosed by glacier ice?* We suggest that smaller scale features such as lakes could be identified by using CSC to isolate target classes (line 633). For example, if we are purely interested in supraglacial lakes, the glacier ice class could be used as a search area input for an additional model which is designed to detect lakes (this could be a simple MLP), rather than searching a whole image which may produce noisier outputs.

*Supplement:*

*1. Why did you use additional Helheim scenes for the joint training method even though the single model was trained on Helheim anyways? (see scenes with \* in Table S1)* It allows a comparison and appreciation for the addition of fine-tuning for out-of-sample sites. We did not expect Joint results for the Helheim site to show significant improvements over Single training for the exact reason you suggest, but thought it was valuable to test this anyway.

*2. Why do you provide only a confusion matrix for the single training but not the combined training approach? It would be interesting to see the performance differences to justify the necessity of a joint and single model approach. Moreover, I would assume that the most robust (spatially transferrable) model would be achieved by including several training areas from the beginning (instead of only Helheim) over different glaciers which would make the additional joint training unnecessary.* We have now included confusion matrices for both Single and Joint outputs using optimal parameters to allow comparison. We agree that the addition of more training data from more glaciers may improve the performance of the phase one CNN on out-of-sample images which we state on line 643. Nonetheless, we believe that the ability of CSC to accurately classify out-of-sample images when trained only on data from Helheim (F1s over 90%) and produce calving front errors comparable to manual delineation/previous deep learning methods shows the benefits of the workflow.

*Please don't be discouraged by the length of my review. I know that a lot of work went into the revised version and the provided comments might require some further effort. Nevertheless, my comments are mostly suggestions and not a must hopefully helping to improve your manuscript.* Thank you for your suggestions, they have been valuable, and we think they have helped improve the manuscript.

**Reviewer 3**

*The authors have made significant improvements to the manuscript, and have resolved many of the previously addressed issues. I believe the manuscript provides a valuable addition to the literature by providing a novel approach for delineating calving fronts only using tile labels and including multiple surface classes.* We thank the reviewer for their helpful comments which have helped to improve the clarity of the paper.

*My remaining comments are with regards to improved clarity of the paper:*

*Section 2.1: it should be emphasized that the output of Phase 1 is not used as the input of Phase 2. Rather, the output of phase is the training LABEL of the next phase. Both phases use the original image as the input. This can also be clarified in Figure 1 by adding an arrow from the original image to the input of phase 2.* This has been clarified in section 2.1 and Figure 1 has also been adapted as suggested.

*Lastly, the authors previously noted in the response letter that quantifying the average uncertainty of manual delineations is beyond the scope of the work, while noting that there is "Clearly, there is a 3-5 pixel error in manual digitising." That is exactly my point. Including an approximate (but justified!) uncertainty range for the corresponding manual digitising provides*

*much better context for the baseline "expectation" of the uncertainty of results with the given images and sites.* We have included an estimate of manual digitisation error on line 407 in the revised manuscript and agree that it provides a good baseline expectation for what automated techniques can achieve.